**Impacts of Cloud Microphysics Parameterizations on Simulated Aerosol-Cloud-Interactions**
**for Deep Convective Clouds over Houston**
Yuwei Zhang[1, 2], Jiwen Fan[2, *], Zhanqing Li[1], Daniel Rosenfeld[3]
[1]Department of Atmospheric and Oceanic Science, University of Maryland, College Park, MD,
USA
[2]Atmospheric Sciences and Global Change Division, Pacific Northwest National Laboratory,
Richland, WA, USA
[3] Institute of Earth Sciences, The Hebrew University of Jerusalem, Jerusalem, Israel
* *Correspondence to*: Jiwen Fan (jiwen.fan@pnnl.gov)

**Abstract**

Aerosol-cloud interactions remain largely uncertain in predicting their impacts on weather and climate. Cloud microphysics parameterization is one of the factors leading to the large uncertainty. Here we investigate the impacts of anthropogenic aerosols on the convective intensity and precipitation of a thunderstorm occurring on 19 June 2013 over Houston with the Chemistry version of Weather Research and Forecast model (WRF-Chem) using the Morrison two-moment bulk scheme and spectral-bin microphysics (SBM) scheme. We find that the SBM predicts a deep convective cloud agreeing better with observations in terms of reflectivity and precipitation compared with the Morrison bulk scheme that has been used in many weather and climate models. With the SBM scheme, we see a significant invigoration effect on convective intensity and precipitation by anthropogenic aerosols mainly through enhanced condensation latent heating (i.e., the warm-phase invigoration). Whereas such an effect is absent with the Morrison two-moment bulk microphysics, mainly because the saturation adjustment approach for droplet condensation and evaporation calculation removes the dependence of condensation on droplet properties and limits the ice processes by a more efficient conversion of droplets into raindrops, which leads to less cloud droplets being transported to the altitudes above the freezing level.

## 1 Introduction

Deep convective clouds (DCCs) produce copious precipitation and play important roles in the hydrological and energy cycle as well as regional and global circulation (e.g., Arakawa, 2004; Houze, 2014). DCCs and associated precipitation are determined by water vapor, vertical motion of air, and cloud microphysics that could be affected by aerosols through aerosol-radiative interactions (ARI) or aerosol-cloud interactions (ACI) or both. The cloud-mediated aerosol effects are recognized by the Intergovernmental Panel on Climate Change (IPCC) as one of the key sources of uncertainty in our knowledge of Earth's energy budget and anthropogenic climate forcing (e.g., Arakawa, 2004; Andreae et al., 2005; Haywood and Boucher, 2000; Lohmann and Feichter, 2005).

Precipitation, latent heat, and cloud radiative forcing associated with DCCs are strongly associated with cloud microphysical processes, which can be modulated by aerosols through serving as cloud condensation nuclei (CCN) and ice nuclei (IN). For aerosol-DCC interactions, a well-known theory is that increasing aerosol concentrations can suppress warm rain as a result of increased droplet number but reduced droplet size. This allows more cloud droplets to be lifted to altitudes above the freezing level, inducing stronger ice microphysical processes (e.g., droplet freezing, riming, and deposition) which release larger latent heating, thereby invigorating convective updrafts (referred to as "cold-phase invigoration,"; Khain et al. 2005; Rosenfeld et al., 2008). It is significant in the situations of warm-cloud bases (> 15°C; Fan et al., 2012b; Li et al., 2011; Rosenfeld et al., 2014; Tao and Li, 2016) and weak wind shear (Fan et al., 2009, 2012b, 2013; Li et al., 2008; Lebo et al., 2012). Grabowski and Morrison (2016; 2020) rejected this invigoration concept by arguing that the increase in the buoyancy by freezing is completely offset by the buoyancy for carrying the extra cloud water across the freezing level. However, Rosenfeld

et al. (2008) showed that the buoyancy restores and increases after the precipitation of the ice
hydrometeors that form upon freezing of the high supercooled liquid water content into large
graupel and hail.
Another theory is that increasing aerosols enhances droplet nucleation particularly secondary
nucleation after warm rain initiates, which promotes condensation because of larger integrated
droplet surface area associated with a higher number of small droplets (Fan et al., 2007, 2013,
2018; Koren at al., 2014; Lebo, 2018; Sheffield et al., 2015; Chen et al., 2020). This so-called
"warm-phase invigoration", which is manifested in a warm, humid, and clean environment under
which the addition of a large number of ultrafine aerosol particles from urban pollution leads to
stronger invigoration than the "cold-phase invigoration" (Fan et al., 2018). Grabowski and
Morrison (2020) proposed a different interpretation of the warm-phase invigoration from the
literature listed above. They argued that condensation rates only depend on updraft velocity with
the quasi-steady assumption (i.e., the true supersaturation is approximated with the equilibrium
supersaturation), therefore they interpreted that it is the lower equilibrium supersaturation in
polluted conditions that lead to a larger buoyancy, thus enhanced updraft speeds, and condensation.
Several studies showed that the quasi-steady assumption is invalidated in the conditions of low
droplet concentrations (Politovich and Cooper, 1988; Korolev and Mazin, 2003) or acceleration of
vertical velocity (Pinsky et al., 2013).
Many factors can affect whether aerosols invigorate or suppress convective intensity through
ACI, such as environmental wind shear (Fan et al., 2009; Lebo et al., 2012), relative humidity (Fan
et al., 2007; Khain et al., 2008), and Convective Available Potential  Energy (Lebo et al., 2012;
Morrison, 2012; Storer et al., 2010). Meteorological buffering effects were also found for aerosol
effects on convective clouds over a large region and long-time (over a few days and weeks)
simulations (Stevens and Feingold, 2009; van den Heever et al., 2011). Dagan et al. (2018) showed
that the lifetimes of cloud systems are mostly much shorter than that and rarely reach this buffering
state. For DCCs with complicated dynamics, thermodynamics, and microphysics, aerosol impacts
are extremely complex and still remain poorly known. Confidently isolating and quantifying an
aerosol deep convective invigoration effect from observations requires very long-term
measurements: data of 10 years are still not enough over the Southern Great Plains due to the large
variability of meteorological conditions (Varble, 2018).

Modeling of ACI is quite dependent on cloud microphysics parameterization schemes (e.g.,

Fan et al., 2012a; Khain and Lynn, 2009; Khain et al., 2009, 2015; Lebo and Seinfeld, 2011; Lee
et al., 2018; Loftus and Cotton, 2014; Wang et al., 2013). Two-moment bulk and bin schemes have
been widely used in ACI studies (e.g., Chen et al., 2011; Fan et al., 2013; Khain et al., 2010). In
two-moment bulk schemes, hydrometeor size distributions are diagnosed from the predicted
number and mass with an assumed spectral shape (e.g., gamma function). The saturation
adjustment approach is often used for calculating condensation and evaporation, meaning
supersaturation and undersaturation with respect to water are removed in cloud within a timestep.
Some bulk schemes take the explicit supersaturation approach to allow supersaturation to evolve
(e.g., Morrison and Grabowski, 2007; 2008). In bin schemes, the size distributions of hydrometeors
are discretized by a number of size bins and predicted, which represents some aerosol-cloud
interaction processes more physically compared with bulk schemes (Fan et al., 2016; Khain et al.,
2015). Supersaturation is generally predicted in bin schemes.

Many studies have shown that bulk schemes are limited in representing certain important

microphysical processes such as aerosol activation, condensation, deposition, sedimentation, and
rain evaporation (Ekman et al., 2011; Khain et al., 2009; Lee et al. 2018; Li et al., 2009;  Milbrandt
and Yau, 2005;  Morrison, 2012; Wang et al., 2013). Though bin cloud microphysics can provide
a more rigorous numerical solution and a more robust cloud microphysics representation than
typical bulk microphysics, it is often applied in simulations for process understanding but rarely
in operational applications due to the expensive computation cost. For not introducing further
computation cost, bins schemes are also often run with a prescribed aerosol spectrum assuming a
fixed composition and a simple aerosol budget treatment without coupling with chemistry/aerosol
calculations. As a result, many aerosol life cycle processes such as aerosol nucleation, growth,
aqueous chemistry, aerosol resuspension, and below-cloud wet removal are missing or crudely
parameterized. Therefore, it is difficult to simulate the spatial and temporal variabilities of aerosol
chemical composition and size distribution. In Gao et al. (2016), we have coupled a spectral-bin
microphysics scheme (SBM; Fan et al., 2012a; Khain et al., 2004) with the Chemistry version of
Weather Research and Forecast model (WRF-Chem; Grell et al., 2005; Skamarock et al., 2008),
called WRF-Chem-SBM, to address above-mentioned limitations. In this new model, the SBM
was coupled with the Model for Simulating Aerosol Interactions and Chemistry (MOSAIC; Fast
et al., 2006; Zaveri et al., 2008). The newly coupled system was initially evaluated for warm
marine stratocumulus clouds and showed a much-improved simulation of cloud droplet number
concentration and liquid water content compared with the default Morrison two-moment bulk
scheme (Gao et al., 2016).
The Houston area in summer, where isolated convective clouds with very warm cloud-bases
often occurred in the afternoon (Yuan et al., 2008), offers (a) a combination of polluted aerosols
from the urban and industrial area of Houston with significantly low background aerosol
concentrations surrounding Houston, (b) aerosol sources that are not correlated with meteorology,
and (c) weak synoptic forcing along with strong local triggering in the form of land-sea contrasts
and sea breeze fronts. This combination allows the manifestation of potentially large aerosol
effects. In this study, we choose a sea-breezed induced DCC case occurring 19-20 June 2013 near
Houston to (1) evaluate the performances of WRF-Chem-SBM in simulating deep convective
clouds and (2) gain a better understanding of the differences in aerosol effects predicted by SBM
and the Morrison two-moment bulk scheme as well as the major factors/processes responsible for
the differences. Considering that the convective clouds over the Houston area are mainly impacted
by the aerosols produced from anthropogenic activities, we focus on the anthropogenic aerosol
effect in this study. The simulated storm case is the same as the case for the Aerosol-Cloud-
Precipitation-Cloud (ACPC) Model Intercomparison Project (Rosenfeld et al., 2014;
www.acpcinitiative.org).
**2 Case Description and Observational Data**
A local convective event near Houston, Texas on 19-20 June 2013 is selected for the study
owing to the most favorable conditions for simulating isolated convective cells. As above-
mentioned, the case is also selected for the ACPC Model Intercomparison Project
(www.acpcinitiative.org). The isolated relatively weak convective clouds started from the late
morning because of a trailing front. With increased solar radiation in the early afternoon and
strengthening of a sea breeze circulation that transports warm and humid air from the Gulf of
Mexico to Houston urban area, deep convective cells over Houston and Galveston bay areas
developed (Fig. 1). The strong convective cell observed near the Houston city was initiated around
2145 UTC (local time 16:45) and developed to its peak precipitation at 2217 UTC based on radar
observation (Fig. 1). The maximum reflectivity was more than 55 dBZ. This storm cell lasted for
about 1.5 hours.
We used the following observation data for model evaluation. Particulate matter (PM) 2.5
data provided by Texas Commission for Environmental Quality (TCEQ) at
https://www.tceq.texas.gov/agency/data/pm25.html are used to evaluate the simulated aerosols
near the surface. The data for evaluating cloud base heights and CCN number concentration at
cloud base are obtained from the Visible Infrared Imaging Radiometer Suite (VIIRS) retrievals
based on the method of Rosenfeld et al., (2016). The 2-m temperature and 10-m winds are from
the North American Land Data Assimilation System (NLDAS) with 0.125-deg resolution at
https://climatedataguide.ucar.edu/climate-data/nldas-north-american-land-data-assimilation-
system. The observed radar reflectivity is used to evaluate the simulated convective system. The
radar reflectivity is obtained from Next-Generation Weather Radar (NEXRAD) network at
https://www.ncdc.noaa.gov/data-access/radar-data/nexrad-products, with a temporal frequency of
every ~ 5 minutes and 1 km horizontal spatial resolution.
**3. Model description and experiments**
We conducted model simulations using the version of WRF-Chem based on Gao et al.
(2016) coupling with the Morrison two-moment scheme (Morrison et al., 2005; Morrison et al.,
2009; Morrison and Milbrandt, 2011) and SBM (Khain et al., 2004; Fan et al., 2012). The version
of SBM employed in this study is a fast version of the Hebrew University Cloud Model (HUCM)
described by Khain et al. (2004) with improvements from Fan et al. (2012a) and (2017). The
considered hydrometer size distributions are droplets/raindrops, cloud ice/snow, and graupel. The
graupel version is used because it is more appropriate for simulating the convective storm over the
Houston area than the hail version. SBM is currently coupled with the four-sector version of
MOSAIC (0.039-0.156, 0.156-0.624, 0.624-2.5 and 2.5-10.0 μm). As detailed in Gao et al. (2016),
the aerosol processes including aerosol activation, resuspension, and in-cloud wet-removal are also
improved. Theoretically, both aerosol and cloud processes can be more realistically simulated
particularly under the conditions of complicated aerosol compositions and aerosol spatial
heterogeneity compared with original WRF-Chem. The dynamic core of WRF-Chem-SBM is the
Advanced Research WRF model that is fully compressible and non-hydrostatic with a terrain-
following hydrostatic pressure vertical coordinate (Skamarock et al., 2008). The grid staggering is
the Arakawa C-grid. The model uses the Runge-Kutta 3rd order time integration schemes, and the
3rd and 5th order advection schemes are selected for the vertical and horizontal directions,
respectively. The positive definite option is employed for advection of moist and scalar variables.
Two nested domains with horizontal grid spacings of 2 and 0.5 km and horizontal grid points
of $450 \times 350$ and $500 \times 400$ for Domain 1 and Domain 2, respectively, are used (Fig. 2a), with 51
vertical levels up to 50 hPa which allows about 50-100 m grid spacings below 2-km altitude and
~500 m above it. The simulations for Domain 1 and Domain 2 are run separately and the Domain
1 simulations serve to provide the chemical and aerosol lateral boundary and initial conditions of
Domain 2. The chemical and aerosol lateral boundary and initial conditions for Domain 1
simulations were from a quasi-global WRF-Chem simulation at 1-degree grid spacing, and
meteorological lateral boundary and initial conditions were created from MERRA-2 at the grid
spacing of $0.5° \times 0.625°$ (Gelaro et al., 2017). Two simulations were run over Domain 1 with
anthropogenic emissions turned on and off, respectively, to provide two different aerosol scenarios
for the initial and boundary chemical and aerosol conditions for Domain 2 simulations: (1) a
polluted aerosol scenario with anthropogenic aerosols accounted which is for the real situation; (2)
an assumptive clean scenario without anthropogenic aerosols. Domain 2 is run with initial and
lateral boundary chemical and aerosols fields from Domain 1 outputs and initial and lateral
boundary meteorological conditions from MERRA-2. Note that we use the meteorology from
MERRA-2 as the initial and lateral boundary conditions for Domain 2 instead of Domain 1 outputs,
because we want to keep the initial and lateral boundary meteorological conditions the same for
all the sensitivity tests with different microphysics and aerosol setups (meteorology is different
between the two simulations over Domain 1).

The simulations in Domain 1 were initiated at 0000 UTC on 14 Jun and ended at 1200 UTC

20 June with about 5 days for the chemistry spin-up. The meteorological field was reinitialized
every 36 hours to prevent the model drifting. The dynamic time step was 6 s for Domain 1 and 3
s for Domain 2. The anthropogenic emission was from NEI-2011 emissions. The biogenic
emission came from the Model of Emissions of Gases and Aerosols from Nature (MEGAN)
product (Guenther et al., 2006). The biomass burning emission was from the Fire Inventory from
NCAR (FINN) model (Wiedinmyer et al., 2011). We used the Carbon Bond Mechanism Z
(CBMZ) gas-phase chemistry (Zaveri and Peters, 1999) and MOSAIC aerosol model with four
bins (Zaveri et al., 2008). The physics schemes other than microphysics applied in the simulation
are the Unified Noah land surface scheme (Chen and Dudhia, 2001), Mellor-Yamada-Janjic
planetary boundary layer scheme (Janjic et al., 1994), Multi-layer, Building Environment
Parameterization (BEP) urban physics scheme (Salamanca and Martilli, 2010), the RRTMG
longwave and shortwave radiation schemes (Iacono et al., 2008).

The main purpose of the simulations in Domain 1 is to provide initial and boundary chemical

and aerosol conditions for the simulations in Domain 2. To save computational cost, WRF-Chem
coupled with Morrison two-moment bulk microphysics scheme (Morrison et al., 2005) is used for
the simulations in Domain 1. Two simulations run for Domain 1 are referred to as D1_MOR_anth
in which the anthropogenic emissions are turned on and D1_MOR_noanth where the
anthropogenic emissions are turned off. Then four major experiments are carried out to simulate
the convective event near Houston over Domain 2 with two cloud microphysics schemes and two
aerosol scenarios, respectively. We refer to the simulation in which SBM is used and the
anthropogenic emissions are included using the initial and boundary chemicals and aerosols from
D1_MOR_anth, as our baseline simulation (referred to as "SBM_anth"). SBM_noanth is based on
SBM_anth but uses initial and boundary chemicals and aerosols from D1_MOR_noanth and turns
off the anthropogenic emissions, meaning that anthropogenic aerosols are not taken into account.
MOR_anth and MOR_noanth are the two corresponding simulations to SBM_anth and
SBM_noanth, respectively, using the Morrison two-moment bulk microphysics scheme. To
examine the contribution of the saturation adjustment approach for condensation and evaporation
to the simulated aerosol effects with the Morrison scheme, we further conducted two sensitivity
tests, based on MOR_anth and MOR_noanth, by replacing the saturation adjustment approach in
the Morrison scheme with the condensation and evaporation calculation based on an explicit
representation of supersaturation over a time step as described in Lebo et al. (2012). That is the
supersaturation is solved by the source and sink terms of dynamic forcing and
condensation/evaporation within a one-timestep. Note in both SBM and this modified Morrison
schemes, the supersaturation for condensation and evaporation is calculated after the advection.
These two simulations are referred to as MOR_SS_anth and MOR_SS_noanth, respectively. To
present more robust results, we carry out a small number of ensembles (three) for each case over
Domain 2 (we do not have computer time to do more ensemble runs). The three ensemble runs are
only different in the initialization time: 0000 UTC, 0600 UTC, and 1200 UTC on 19 June. All the
simulations end at 1200 UTC 20 June. The analysis results for Domain 2 simulations in this study
are based on the mean values of three ensemble runs and the ensemble spread is shown as the
shaded area in all profile figures.
We evaluate the aerosol and CCN properties simulated by D1_MOR_anth to ensure realistic
aerosol fields, which are used for the Domain 2 simulations with anthropogenic aerosols
considered. These evaluations are included in section 4.1.
From D1_MOR_anth, we see a very large spatial variability of aerosol number concentrations
(Fig. 2b). There are three regions with significantly different aerosol loadings over the domain as
shown by the black boxes in Fig. 2b: (a) the Houston urban area, (b) the rural area about 100 km
northeast to Houston, and (c) Gulf of Mexico. Aerosols over the Houston urban area are mainly
contributed by organic aerosols, which are highly related to industrial and ship channel emissions.
The rural area aerosols are mainly from sulfate and sea salt aerosol is the major contributor over
the Gulf of Mexico. This suggests that aerosol properties are extremely heterogenous in this region.
The aerosols over Houston urban area are generally about 5 and 10 times higher than the rural and
Gulf area, respectively (Fig. 2c). The size distributions show a three-mode distribution with the
largest differences from the Aitken mode (peaks at 50 nm; Fig. 2c). These ultrafine aerosol
particles are mainly contributed by anthropogenic activities (Fig. 2b, d). With the anthropogenic
emissions turned off, the simulated aerosols are much lower and have much less spatial variability
(Fig. 2d).
**4 Result**
**4.1 Model Evaluation**
We first show the evaluation of the aerosol and CCN properties simulated by
D1_MOR_anth, which runs over Domain 1, much larger than Domain 2. As described in Table 1,
there are eight PM monitoring sites from TCEQ around the Houston area. Surface PM2.5 shows
high concentrations at Houston and its downwind regions (Fig. 3). Though not exactly the same,
the values from D1_MOR_anth show a similar distribution with the observations in terms of the
surface PM2.5 averaged over 24 hours (the day before the convection near Houston). The hourly
variations of ground-level PM2.5 concentrations from both observation and D1_MOR_anth for
these sites in the day before the convective initiation is depicted in Fig. 4. Generally, the simulated
hourly pattern agrees with the observation for eight stations. D1_MOR_anth reproduces the diurnal
variations, especially the increasing trend from 1200 UTC to 1800 UTC 19 Jun prior to the
initiation of deep convective cells over Houston and Galveston bay areas.
The evaluation of the cloud base heights and CCN at cloud bases at the warm cloud stage
before transitioning to deep clouds (2000 UTC) are shown in Fig.5. Over the Houston and its
surrounding area (black box in Fig. 5), the simulated cloud base heights are about 1.5-2 km, in an
agreement with the retrieved values from VIIRS satellite, which are around 1.2-1.8 km (Fig. 5a-
b). The retrieved CCN concentrations at cloud bases vary significantly over the domain and this
spatial variability is generally captured by the model (Fig. 5c-d). For example, D1_MOR_anth
simulates some high CCN concentrations (400-800 $cm^{-3}$ with some above 1000 $cm^{-3}$) over the
Houston and around the Bay area, relatively low CCN values at the rural areas (about 200-600 $cm^{-}$
$^3$), and very low values over the Gulf of Mexico (less than 200 $cm^{-3}$), as shown in Fig. 5d. This is
consistent with the spatial variability from the retrievals (Fig. 5c). The evaluation of aerosol
properties before the initiation of Houston convective cells and CCN at the warm cloud stage
before transitioning to deep clouds provides us confidence in using the chemical and aerosol fields
from Domain 1 outputs to feed Domain 2 simulations.
Now we are evaluating near-surface temperature and winds, reflectivity and precipitation
simulated by SBM_anth and MOR_anth. Fig. 6 shows the comparisons in 2-m temperature and
10-m winds at 1800 UTC (before the convective initiation). Compared with the coarse resolution
NLDAS data, both SBM_anth and MOR_anth capture the general temperature pattern with a little
overestimation at the northeast part of the domain (mainly rural area). The modeled southerly
winds do not reach further north as the NLDAS data, possibly because of the feedback of the small-
scale features which are simulated with the high resolution to mesoscale circulations. However,
the simulation of temperature over Houston and sea breeze winds from the Gulf of Mexico to
Houston is the most important in this case. SBM_anth predicts a slightly higher temperature than
MOR_anth in the northern part of the Houston region (purple box in Fig. 6), which agrees with
NLDAS better. SBM_anth gets the similar southerly winds from the Gulf of Mexico to Houston
as shown in NLDAS, while the southerly winds from Gulf of Mexico become very weak or
disappear prior to reaching Houston in MOR_anth.

For the Houston convective cell that we focused (red box in Fig. 7a), SBM_anth simulates

it well in both location and high reflectivity value (greater than 50 dBZ) in comparison with the
NEXRAD observation (Fig. 7a-b). The simulated composite reflectivities (i.e., the column
maximum) are up to 55-60 dBZ from all three ensemble members, consistent with NEXRAD.
With the Morrison scheme, MOR_anth simulates several small convective cells near Houston with
a maximum reflectivity of 55 dBZ or less (Fig. 7c). All three ensemble members consistently show
smaller but more scattered convective cells with the Morrison scheme compared with SBM. The
contoured frequency by altitude diagram (CFAD) plots for the entire storm period show that
SBM_anth is in a better agreement with observation compared with MOR_anth, especially for the
vertical structure of the high reflectivity range (greater than 48 dBZ, black dashed lines in Fig. 8)
and echo top heights, which can reach up to 14-15 km (Fig. 8a-b). MOR_anth overestimates the
occurrence frequencies of the 35-45 dBZ range and underestimates those of the low and high
reflectivity ranges (less than 15 dBZ or larger than 50 dBZ) as well as the echo top heights (1-2
km lower than SBM_anth; Fig. 8c).

For the precipitation rates averaged over the study area (red box in Fig. 7), the observation

shows two peaks, which are captured by both SBM_anth and MOR_anth (Fig. 9a). However, the
timing for the first peak is about 30 and 60 min earlier in SBM_anth and MOR_anth than the
observation, respectively. Also, SBM_anth predicts the rain rate intensities at the two peak times
more consistent with the observations whereas MOR_anth underestimates the rain rate intensity at
the second peak time (Fig. 9a). The large precipitation rates (greater than 15 mm h$^{-1}$) in SBM_anth
has a ~1.5 times larger occurrence probability than those in MOR_anth, showing a better
agreement with the observation (Fig. 9b). The observed accumulated rain over the time period
shown in Fig. 9a is about 3.8 mm, both SBM_anth (~4.5 mm) and MOR_anth (~4.2 mm)
overestimate the accumulated precipitation due to the longer rain period compared with the
observations. Overall, the performance of SBM_anth is superior to MOR_anth in simulating the
location and intensity of the convective storm and associated precipitation.
**4.2 Simulated Aerosol Effects on Cloud and Precipitation**

Now we look at the effects of anthropogenic aerosols on the deep convective storm

simulated with SBM and Morrison microphysics schemes. Fig. 9a shows that with the SBM
scheme, anthropogenic aerosols remarkably increase the mean surface rain rates (by ~30%; from
SBM_noanth to SBM_anth), mainly because of the increased occurrence frequency (nearly
doubled) for relatively large rain rates (i.e., 10-15 mm h$^{-1}$ and >15 mm h$^{-1}$) in Fig. 9b. With the
Morrison scheme, the changes in mean precipitation and the PDF from MOR_noanth to
MOR_anth are relatively small, showing a very limited aerosol effect on precipitation. With the
SBM scheme, the increase in the updraft speeds by the anthropogenic aerosols is even more notable
than the precipitation (Fig. 10a-b). Above 5-km altitude, the occurrence frequencies of updraft
speed greater than 0.4% extend to much larger values, with 36 m s$^{-1}$ at the upper levels in
SBM_anth while only ~ 20 m s$^{-1}$ in SBM_noanth. With the Morrison scheme, the changes are not
significant by the anthropogenic aerosols (MOR_noanth vs MOR_anth in Fig. 10c-d). From
MOR_noanth to MOR_anth, there is a slight increase in updraft speed at around 9-11 km altitudes
but a slight decrease at 6-8 km altitudes. The significant invigoration of convective intensity by
anthropogenic aerosols with the SBM scheme explains the much larger occurrences of relatively
large rain rates and overall more surface precipitation due to the anthropogenic aerosol effect (Fig.
9). Note Fig. 9a shows that anthropogenic aerosols lead to an earlier start of the precipitation with
both SBM and Morrison, which reflects the faster transition of warm rain to mixed-phase
precipitation. We do see the delay of warm rain by aerosols but only about 5 min (probably due to
the humid condition of the case), which is difficult to be shown in Fig. 9a since averaged rain rate
for the analysis box is ~0.02 mm hr$^{-1}$ and the time period is very short (~10 min).

Now the question is why the anthropogenic aerosols enhance the convective intensity of

the storm with the SBM scheme while the effect is very small with the Morrison scheme. Fig. 11
shows the vertical profiles of mean updraft velocity, thermal buoyancy (from temperature and
moisture perturbation), and total latent heating rate of the top 25$^{th}$ percentile updrafts with a value
greater than 2 m s$^{-1}$ during the deep convective cloud stage. With the SBM microphysics scheme,
the increased convective intensity due to the anthropogenic aerosol effect corresponds to the
increased thermal buoyancy which is particularly notable at upper levels (~ 20%) from
SBM_noanth to SBM_anth (Fig. 11a, c). The increased thermal buoyancy can be explained by
the increased total latent heating (Fig. 11e), which is mainly from the larger condensation latent
heating (Fig. 12a). From SBM_noanth to SBM_anth, the latent heating from ice-related
microphysical processes (including deposition, drop freezing, and riming) has a relatively smaller
increase than that from condensation (about half of the increase in condensation latent heating as
shown in Fig. 12a). As shown in Fan et al., (2018), the increase in lower-level condensation latent
heating has a much larger effect on intensifying updraft intensity compared with the same amount
of increase in high-level latent heating from ice-related microphysical processes. This suggests
that the convective invigoration by the anthropogenic aerosols with the SBM scheme should be
mainly through the "warm-phase invigoration" mechanism.  Compared with the Morrison scheme,
the increase of total latent heating by the anthropogenic aerosols is almost doubled with the SBM
scheme, explaining more remarkable enhancement of thermal buoyancy and thus the convective
intensity (red lines vs blue lines in Fig. 11). From MOR_noanth to MOR_anth, there is a small
increase in both the condensation latent heating and high-level latent heating associated with ice-
related processes (blue lines in Fig. 12b). The major difference in the increase of latent heating by
the anthropogenic aerosols between SBM and Morrison microphysics schemes comes from the
condensation latent heating, with a ~20% increase with SBM but only ~ 8% with Morrison (Fig.
12). The lack of a significant increase in condensation latent heating limits the "warm-phase
invigoration", mainly responsible for the limited aerosol impacts on the convective intensity and
associated precipitation with the Morrison scheme.

To understand why the responses of condensation to the anthropogenic aerosols are

different between the SBM and Morrison schemes, we look into the process rates of drop
nucleation and condensation (Fig. 13). The calculations of aerosol activation and
condensation/evaporation in the SBM scheme are based on the Köhler theory and diffusional
growth equations in light of particle size and supersaturation, receptively. Whereas in the Morrison
scheme, the Abdul-Razzak and Ghan (2002) parameterization is used for aerosol activation and
the saturation adjustment method is applied for condensation and evaporation calculation. With
the SBM scheme, the anthropogenic aerosols increase the drop nucleation rates by a few times
over the profile (red lines in Fig. 13a), and the condensation rates are also drastically increased
(doubled between 4-6 km altitudes as shown in Fig. 13c). The enhanced condensation rate by the
anthropogenic aerosols is because much more aerosols are activated to form a larger number of
small droplets, increasing the integrated droplet surface area for condensation, as documented in
Fan et al., (2018). As a result, supersaturation is drastically lower in SBM_anth than SBM_noanth
(green lines in Fig. 13a). With the Morrison scheme, we still see a large increase in the droplet
nucleation rate (Fig. 13b). However, the condensation rates are barely increased (blue solid vs.
dashed lines in Fig. 13d). We hypothesize that the lack of response of condensation to the increased
aerosol activation with the Morrison scheme is mainly because of the saturation adjustment
calculation of the condensation and evaporation process. The approach does not allow
supersaturation in cloud and the calculation does not depend on supersaturation, thus removes the
sensitivity to the anthropogenic aerosols.
To verify our hypothesis and examine how much the saturation adjustment method is
responsible for the weak responses of condensation latent heating and convection to the added
anthropogenic aerosols, we conducted two additional sensitivity tests by replacing the
saturation adjustment approach in the Morrison scheme with the condensation and evaporation
calculation based on an explicit representation of supersaturation over a time step, as described in
Section 3. The result shows the Morrison scheme with the simple calculation of supersaturation
for condensational growth significantly changes the condensation rate (orange vs. blue lines in Fig.
13d) and a similarly large enhancement  (from MOR_SS_noanth to MOR_SS_anth in Fig. 13d) is
seen as the SBM scheme (Fig. 13c). This leads to a larger increase in condensation latent heating
(orange lines in Figure 12b) compared with the original Morrison scheme, resulting in a similarly
large increase in thermal buoyancy by the anthropogenic aerosols as with the SBM scheme(orange
lines in Fig. 11d), thus a similarly large increase in the convective intensity (orange lines in Fig.
11b). The increase of precipitation from MOR_SS_noanth to MOR_SS_anth is also similar to that
with the SBM scheme (not shown). These results verify that the saturation adjustment approach
for parameterizing condensation and evaporation is the major reason responsible for limited
aerosol effects on convective intensity and precipitation with the Morrison scheme. Past studies
also showed the limitations of the saturation adjustment approach in simulating aerosol impacts
on deep convective clouds (e.g., Fan et al., 2016; Lebo et al., 2012; Lee et al., 2018; Wang et al.,

2013).

Fig. 14 and Fig. 15 show the responses of hydrometeor mass and number to anthropogenic
aerosol effects. With the SBM scheme, the increases in mass and number of cloud droplets,
raindrops, and total ice particles (ice, snow, and graupel) by the anthropogenic aerosols are very
significant (Fig. 14-15, left), corresponding to convective invigoration. The increase of the total
ice mass is particularly significant (from 3.5 to 5.5 g kg$^{-1}$ around 10-km altitude), suggesting a
large effect of enhanced convective intensity on ice hydrometeors. However, with the Morrison
scheme, little change is seen (Fig. 14-15, right, blue lines). By replacing the saturation adjustment
with a simple calculation based on supersaturation for condensation and evaporation in the
Morrison scheme, the increases in those hydrometeor masses become as evident as those with the
SBM scheme (Fig. 14-15, right, orange lines).
Now we explain why the saturation adjustment approach leads to smaller condensational
heating than the explicit supersaturation approach in Morrison Scheme and why it leads to a
smaller sensitivity to aerosols compared with the explicit supersaturation approach. We examine
the time evolution of latent heating, updraft, and hydrometeor properties. At the warm cloud stage
at 1700 UTC, the saturation adjustment indeed produces more condensational latent heating which
leads to larger buoyancy and stronger updraft intensity compared to the explicit supersaturation
because of removing supersaturation (Fig. 16, left, blue vs. orange). By the time of 1900 UTC
when the clouds have developed into mixed-phase clouds, the saturation adjustment produces less
condensational heating and weaker convection than the explicit supersaturation approach (Fig. 16,
middle). The results remain similarly later at the deep cloud stage 2100 UTC (Fig. 16, right).

How does this change happen from 1700 to 1900 UTC? At the warm cloud stage (17:00

UTC), the saturation adjustment produces droplets with larger sizes (up to 100% larger for the
mean radius) than the explicit supersaturation because of more cloud water produced as a result of
zeroing-out supersaturation at each time step (droplet formation is similar between the two cases
as shown in Fig. 13). This results in much faster and larger warm rain, while with the explicit
supersaturation rain number and mass are absent at 1700 UTC as shown in Fig. 17d and 18d). As
a result, when evolving into the mixed-phase stage (19:00 UTC), much fewer cloud droplets are
transported to the levels above the freezing level (Fig. 17b and 18b). Whereas with the explicit
supersaturation, because of the delayed/suppressed warm rain and smaller droplets (the mean
radius is decreased from 8 to 6 μm at 3 km), much more cloud droplets are lifted to the higher
levels. Correspondingly, a few times higher total ice particle number and mass are seen compared
with the saturation adjustment (Fig. 17g and 18g) because more droplets above the freezing level
induce stronger ice processes (droplet freezing, riming, and deposition). This leads to more latent
heat release (Fig. 16e), which increases the buoyancy and convective intensity. With the explicit
supersaturation, increasing aerosols leads to a larger reduction in droplet size (up to 1 μm more in
the mean radius) than the saturation adjustment, therefore more enhanced ice microphysical
processes and the larger latent heat. Besides, the condensational heating is more enhanced by
aerosols with the explicit supersaturation (Fig. 16). Together, a much larger sensitivity to aerosols
is seen with the explicit supersaturation.

**5 Conclusions and Discussion**

We have conducted model simulations of a deep convective cloud case occurring on 19 June
2013 over the Houston area with WRF-Chem coupled with the SBM and Morrison microphysics
schemes to (1) evaluate the performance of WRF-Chem-SBM in simulating the deep convective
clouds, and (2) explore the differences in aerosol effects on the deep convective clouds produced
by the SBM and Morrison schemes and the major factors responsible for the differences.
We have evaluated the simulated aerosols, CCN, cloud base heights, reflectivity, and
precipitation. The model simulates the large spatial variability of aerosols and CCN from the Gulf
of Mexico, rural areas, to Houston city. On the bulk magnitudes, the model captures the surface
PM2.5, cloud base height, and CCN at cloud bases near the Houston reasonably well. These
realistically simulated aerosol fields were fed to higher resolution simulations (0.5 km) using the
SBM and Morrison schemes. With the SBM scheme, the model simulates a deep convective cloud
over Houston in a better agreement with the observed radar reflectivity and precipitation,
compared with using the Morrison scheme.
By excluding the anthropogenic aerosols in the simulations, the effects of anthropogenic
aerosols on the deep convective clouds and differences in aerosol effects using the two
microphysics schemes were examined. With the SBM scheme, anthropogenic aerosols notably
increase the convective intensity, enhance the peak precipitation rate over the Houston area (by ~
30%), and double the frequencies of relatively large rain rates (> 10 mm h$^{-1}$).  The enhanced
convective intensity by anthropogenic aerosols makes the simulated storm agree better with the
observed, mainly attributed to the increased condensation latent heating, indicating the "warm-
phase invigoration". In contrast, with the Morrison scheme, there is no significant anthropogenic
aerosol effect on the convective intensity and precipitation.

Sensitivity tests by replacing the saturation adjustment with the condensation and evaporation

calculation based on an explicit supersaturation approach show weaker warm clouds with smaller
cloud droplet sizes because of smaller condensational growth than the saturation adjustment which
eliminates all supersaturation. This leads to less efficient conversion of cloud droplet to rain and
allows more cloud droplets to be transported to altitudes above the freezing level at the mixed-
phase and deep cloud stages, resulting in stronger ice microphysical processes (freezing, riming,
and deposition), therefore larger latent heat release, invigorating convective updrafts. Lebo et al.
(2012) showed a similar feature that the saturation adjustment has larger total condensate mass at
the beginning but less at the later stage compared to the explicit supersaturation approach,
particularly in total ice mass. Grabowski and Morrison (2017) also showed that the saturation
adjustment affected ice processes by producing larger ice particles with larger falling velocities
compared with the explicit supersaturation approach, leading to the reduction of anvil clouds. The
increased condensation is significant for the enhanced warm clouds when saturation adjustment is
used. This is different from the points of Grabowski and Jarecka (2015) that the cloud edge
evaporation effect is more important for the nonprecipitating shallow clouds.

It is also notable that the adjusted Morrison scheme by replacing saturation adjustment with

explicit supersaturation for condensation and evaporation show the similar aerosol effects on
condensation, convective intensity, hydrometeor mass mixing ratios, and precipitation as with the
SBM scheme. Therefore, the saturation adjustment method for the condensation and evaporation
calculation is mainly responsible for the limited aerosol effects with the Morrison scheme. This is
because the saturation adjustment method does not allow for the "warm-phase invigoration",
which is different from Lebo et al. (2012) showing that the saturation adjustment artificially
enhanced condensation latent heating at low levels and limited the potential for aerosols to
invigorate convection through the "cold-phase invigoration" mechanism in their idealized
simulations of a supercell storm with the thermal bubble initiation. In this study of the
thunderstorm with WRF real-case simulations for both chemistry/aerosols and clouds, the
saturation adjustment method actually leads to a smaller condensation latent heating than the
explicit calculation with supersaturation (solid bold blue vs. solid bold orange line in Fig. 12b).
Thus, when the computational resource is not sufficient or in other situations such as the
application of SBM is not available, the Morrison scheme modified with the condensation and
evaporation calculation based on a simple representation of supersaturation can be applied to study
aerosol effects on convective clouds, especially for warm and humid cloud cases in which the
response of condensation to aerosols is particularly important.

Following Fan et al., (2018), which showed that the "warm-phase invigoration" mechanism

was manifested by ultrafine aerosol particles in the Amazon warm and humid environment with
extremely low background aerosol particles. Here we showed that in summer anthropogenic
aerosols over the Houston area may also enhance the thunderstorm intensity and precipitation
through the same mechanism by secondary nucleation of numerous ultrafine aerosol particles from
the anthropogenic sources.  But the magnitude of the effect is not as substantial as in the Amazon
environment. Possible reasons include that background aerosols are much higher over the Houston
area and air is not as humid as Amazon.

**Acknowledgments**

This study is supported by the U.S. Department of Energy Office of Science through its Early Career Award Program and a grant DE-SC0018996 and the NSF (AGS1837811). PNNL is operated for the US Department of Energy (DOE) by Battelle Memorial Institute under Contract DE-AC05-76RL01830. This research used resources of PNNL Institutional Computing (PIC), and the National Energy Research Scientific Computing Center (NERSC), a U.S. Department of Energy Office of Science User Facility operated under contract DE-AC02-05CH11231. We thank Chun Zhao at China University of Science and Technology for providing the quasi-global WRF-Chem simulation data, and Hugh Morrison at the National Center for Atmospheric Research for the Morrison code with supersaturation-forced condensation and evaporation calculation.

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

**Table 1** Descriptions of the PM2.5 Monitoring Sites over the Houston area from TCEQ

| Abbreviation | Site Descriptions | Latitude | Longitude |
|---|---|---|---|
| HA | Houston Aldine | 29.901 | -95.326 |
| HDP | Houston Deer Park 2 | 29.670 | -95.129 |
| SFP | Seabrook Friendship Park | 29.583 | -95.016 |
| CR | Conroe Relocated | 30.350 | -95.425 |
| KW | Kingwood | 30.058 | -95.190 |
| CT | Clinton | 29.734 | -95.258 |
| PP | Park Place | 29.686 | -95.294 |
| GS | Galveston 99th Street | 29.254 | -94.861 |




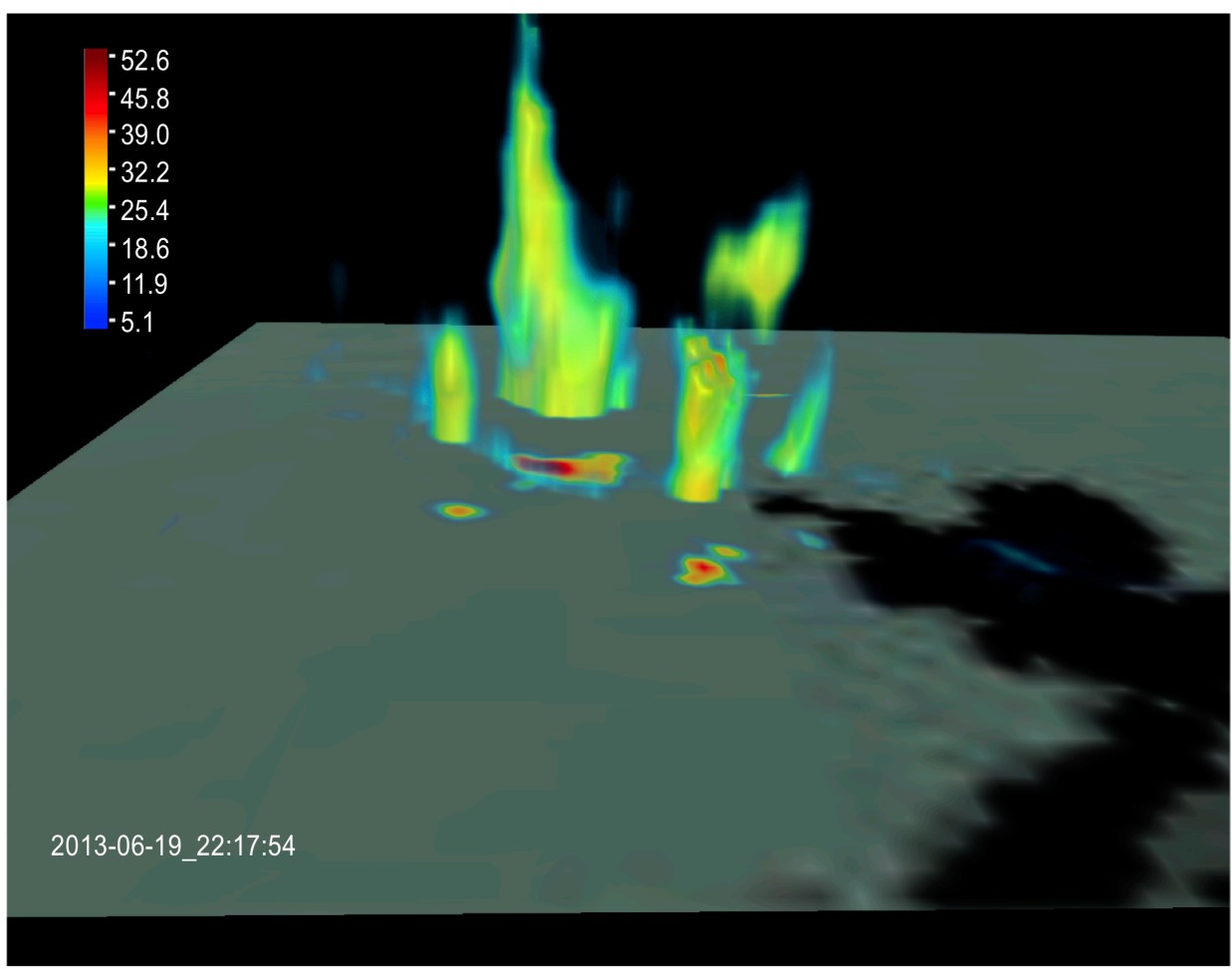


**Figure 1** 3D structure snapshot of radar reflectivity (unit: dBZ) from NEXRAD, overlaid with the
composite reflectivity shown on the surface at the time when the maximum reflectivity is observed
(2217 UTC). The dark shade shows the water body and the largest cell is in the Houston.

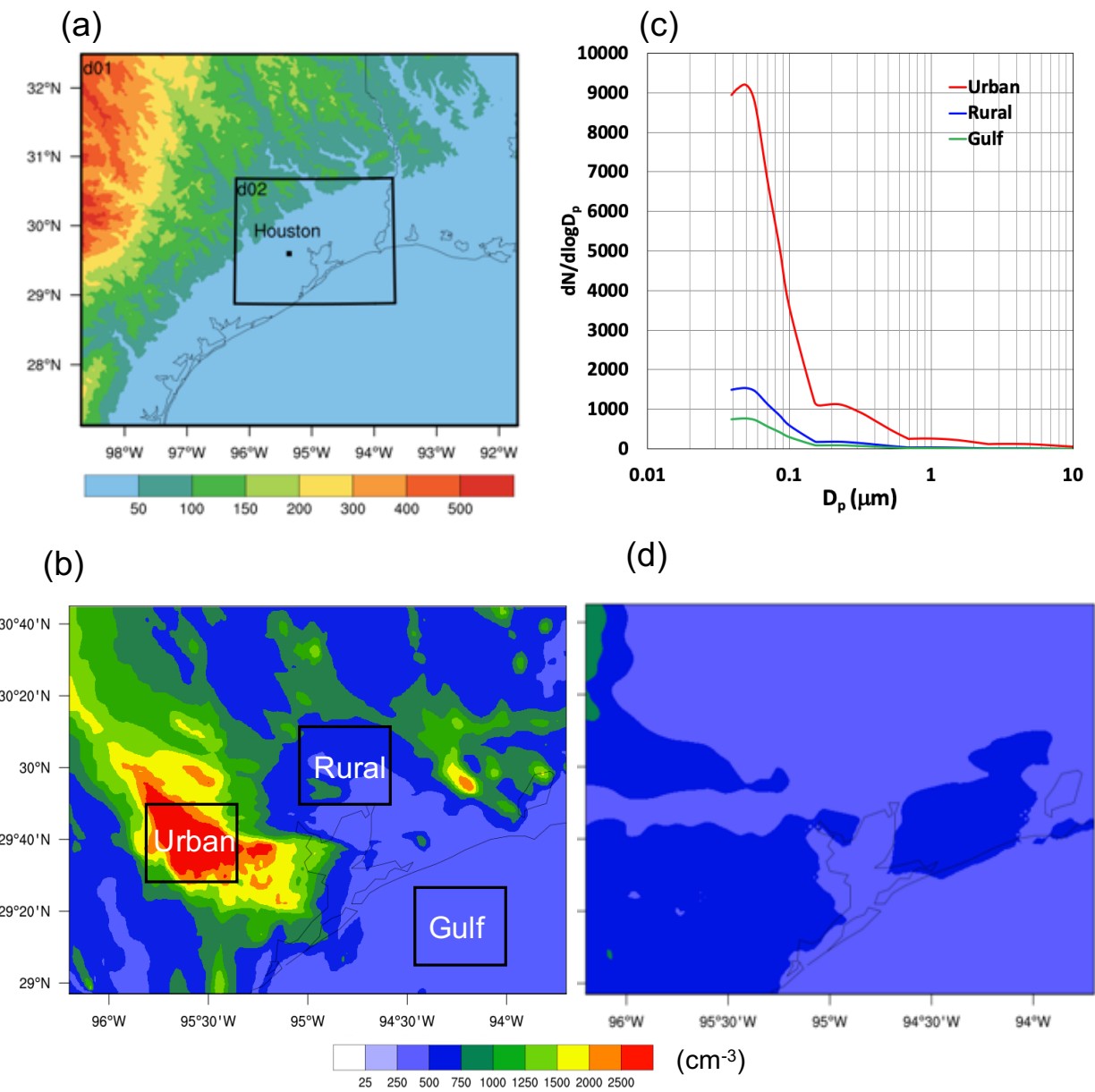


**Figure 2** (a) Simulation domains with the terrain heights (unit: m), (b) aerosol number

concentration (unit: cm$^{-3}$) from D1_MOR_anth, (c) aerosol size distributions over the urban, rural,

and Gulf of Mexico as marked by three black boxes in Fig. 2b at 1200 UTC, 19 Jun 2013 (6-hr

before the convection initiation), and (d) the same as Fig. 2b, but for D1_MOR_noanth in which

the anthropogenic aerosols are excluded.


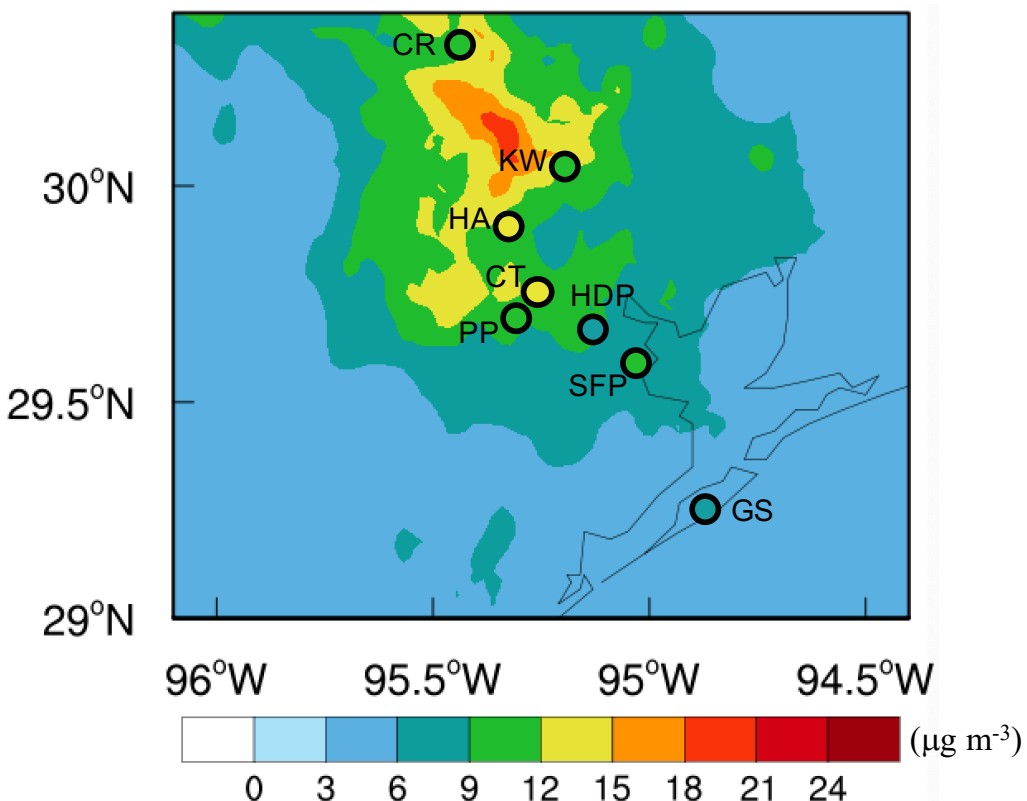


**Figure 3** Comparisons of 24-hr averaged PM2.5 mass concentrations (unit: μg m$^{-3}$) between model
simulation D1_MOR_anth (contoured) and site observation from TCEQ (colored circles) from
1800 UTC, 18 June 2013 to 1800 UTC, 19 June 2013 (1 day before the convection initiation). The
site names and other information are shown in Table 1.

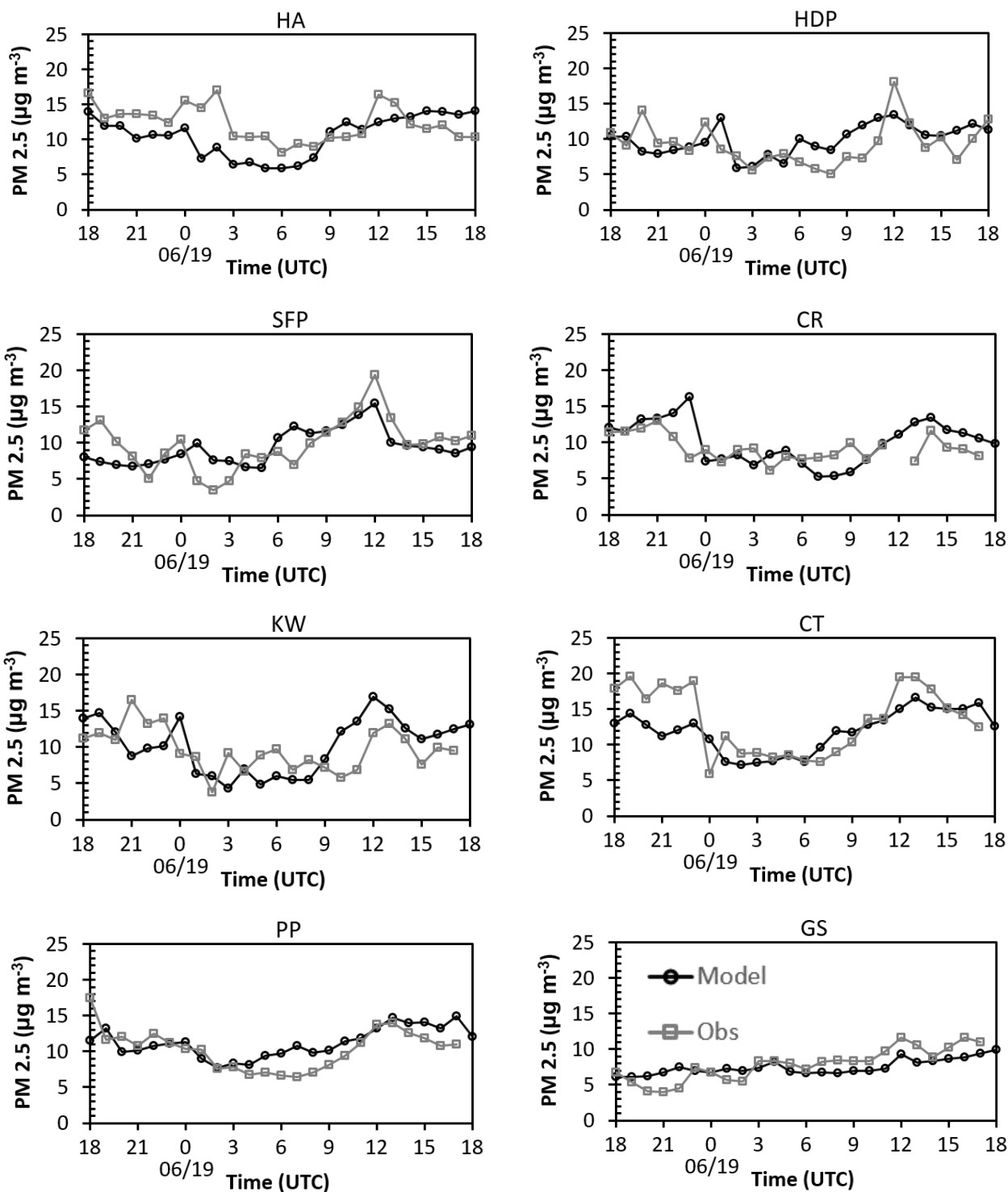


**Figure 4** Site-by-site comparisons of hourly PM2.5 mass concentrations (unit: μg m⁻³) from

D1_MOR_anth and TCEQ site observation over 24 hours from 1800 UTC, 18 June 2013 to 1800
UTC, 19 June 2013 (1 day before the convection initiation).

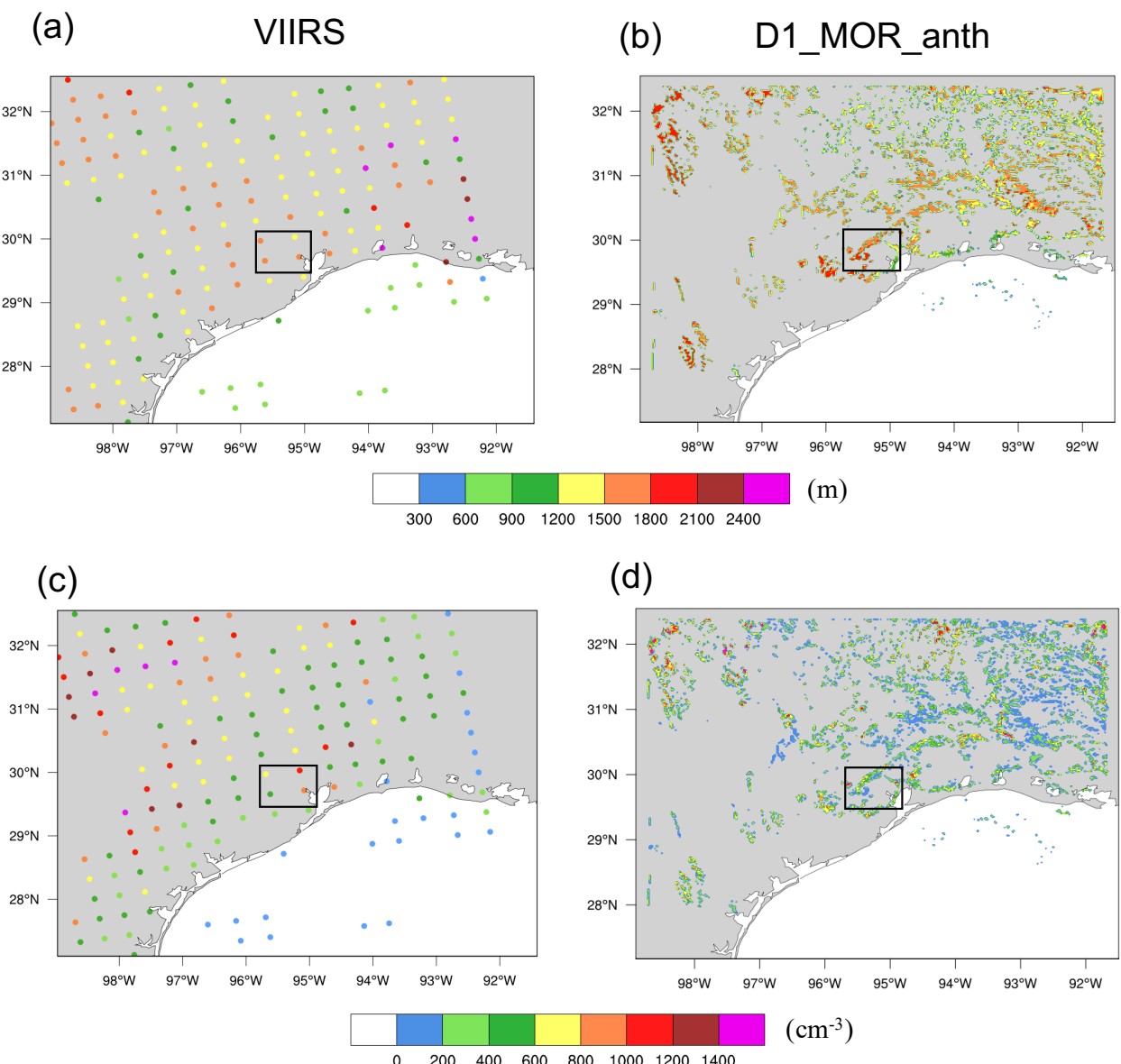


**Figure 5** Evaluation of (a,b) cloud base heights (unit: m) and (c,d) CCN number concentration at cloud base (unit: cm$^{-3}$) from VIIRS satellite (left) retrieved at 1943 UTC (Rosenfeld et al. 2016) and model simulation D1_MOR_anth (right) at 2000 UTC, 19 June 2013. The Houston area is marked as the black box. Satellite-retrieved cloud base height was calculated from the difference between reanalysis surface air temperature (from reanalysis data) and VIIRS-measured cloud base temperature (warmest cloudy pixel) divided by the dry adiabatic lapse rate, while modeled cloud base height was determined by the lowest cloud layer with cloud mass mixing ratio greater than $10^{-5}$ kg kg$^{-1}$.


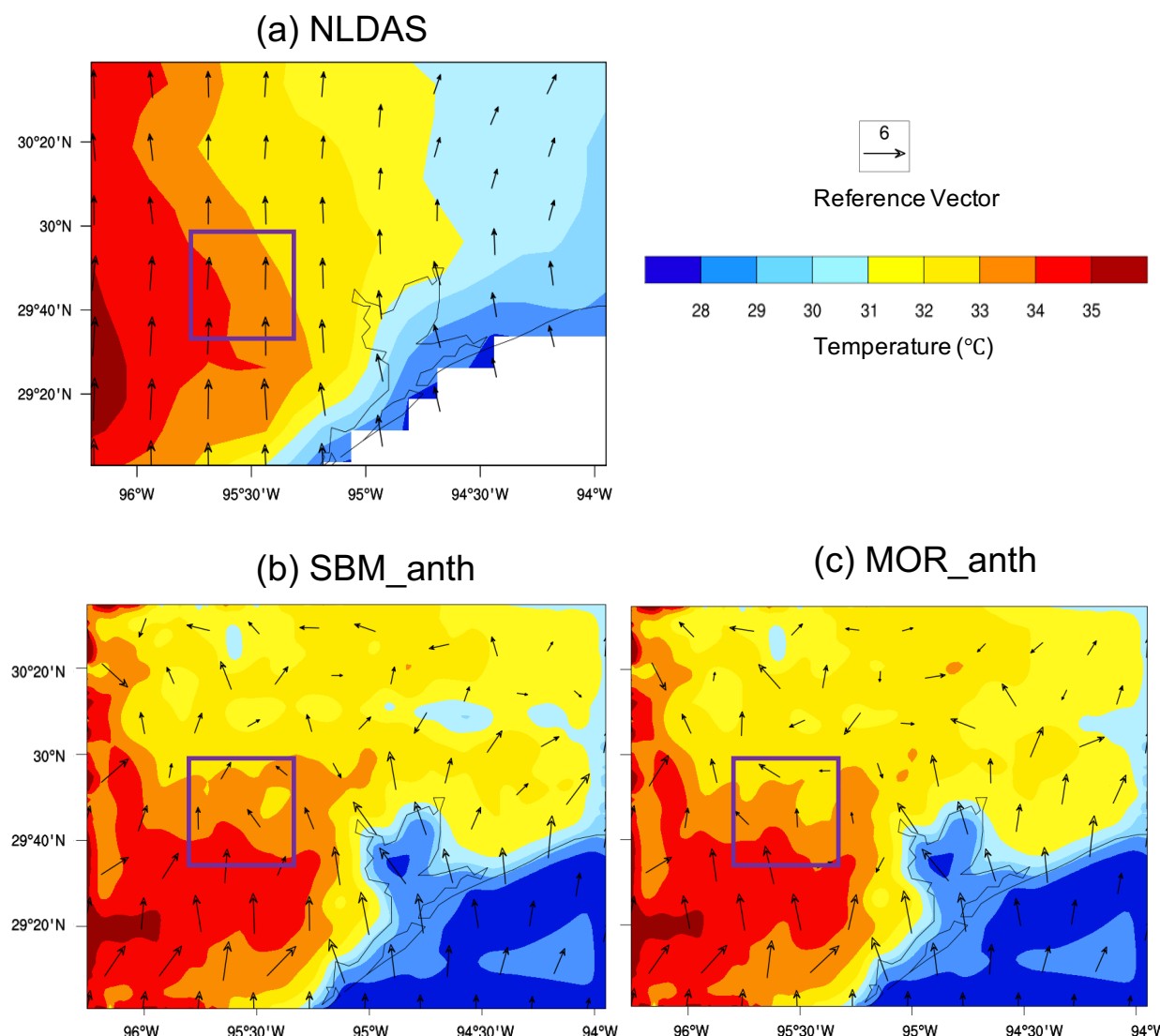


**Figure 6** 2-m Temperature (shaded; unit: °C) and 10-m winds (vectors; unit: m s$^{-1}$) from (a)

NLDAS, (b) SBM_anth and (c) MOR_anth at 1800 UTC, 19 Jun 2013. The purple box denotes

the Houston area.


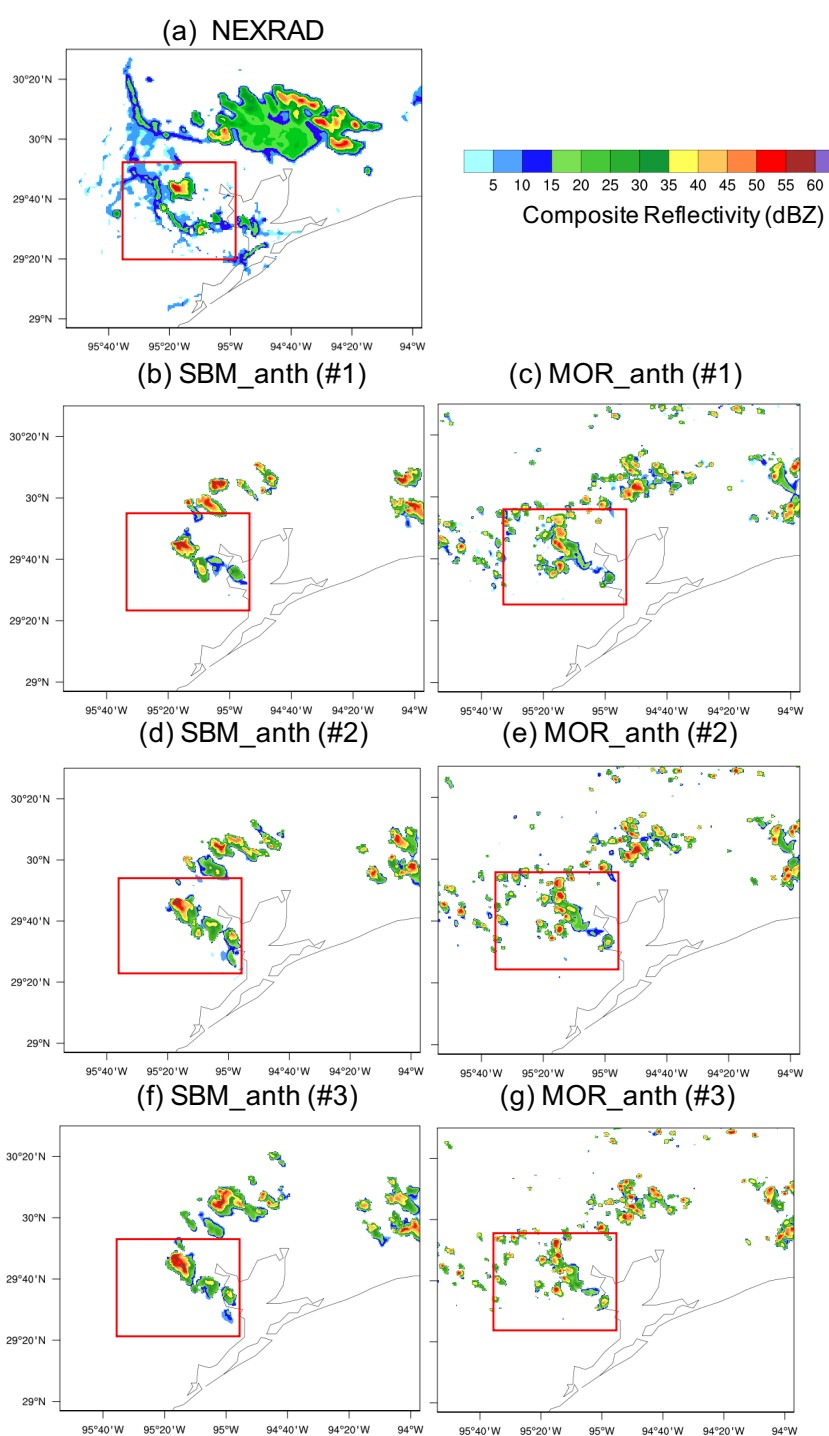


**Figure 7** Composite reflectivity (unit: dBZ) from (a) NEXRAD (2217 UTC), (b, d, f) three
ensemble runs for SBM_anth (2140 UTC) and (c, e, g) three ensemble runs for MOR_anth (2125
UTC) when maximum reflectivity in Houston is observed on 19 June 2013. The red box is the
study area for convection cells near Houston.

792

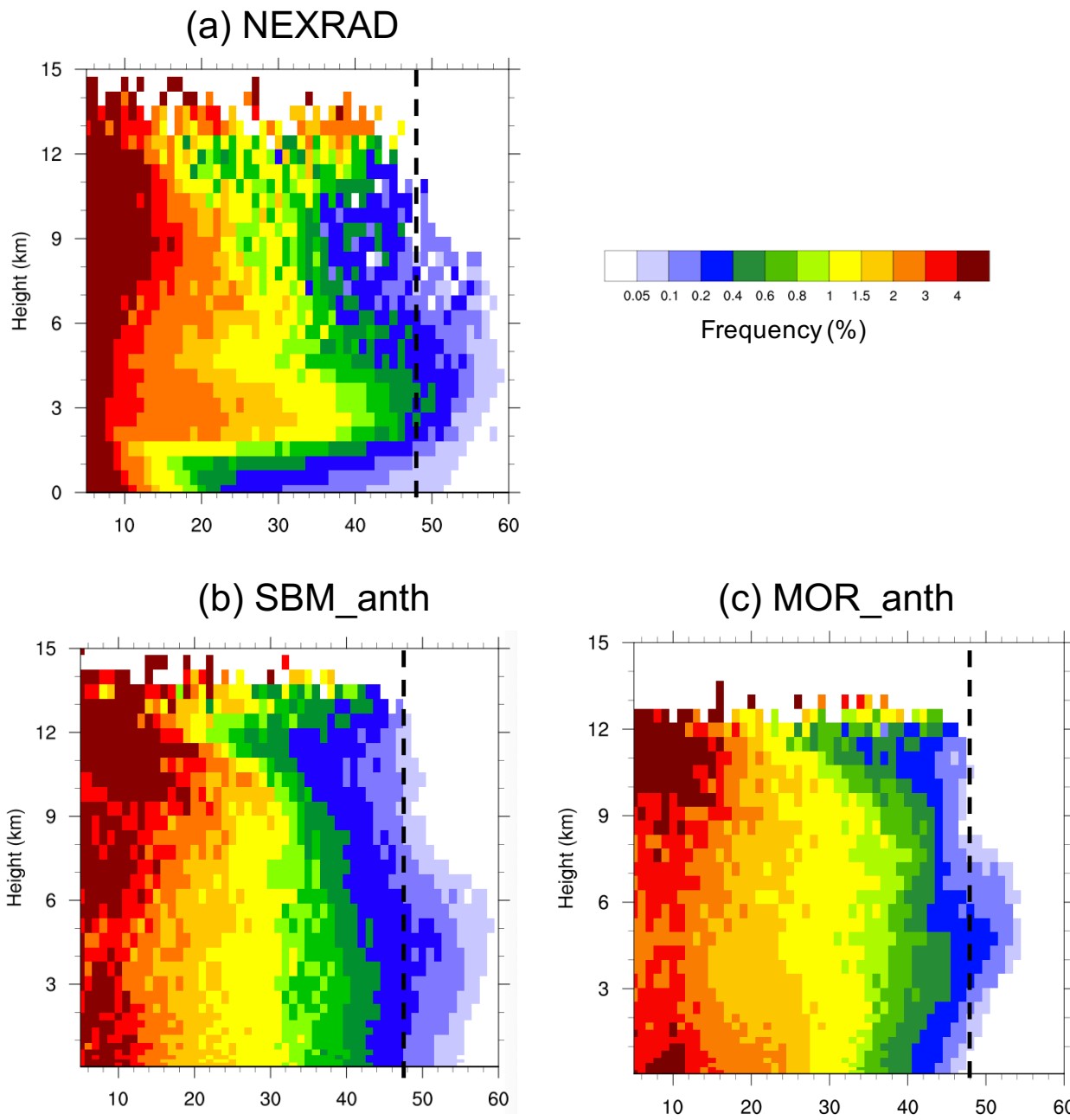

**Figure 8** The CFAD of reflectivity (unit: dBZ) for the values larger than 0 dBZ from (a) NEXRAD, (b) SBM_anth and (c) MOR_anth over the study area (red box in Fig. 7) from 1800 UTC, 19 Jun to 0000 UTC, 20 Jun 2013. The black solid lines denote the reflectivity with the value of 48 dBZ. The results are the three ensemble means.

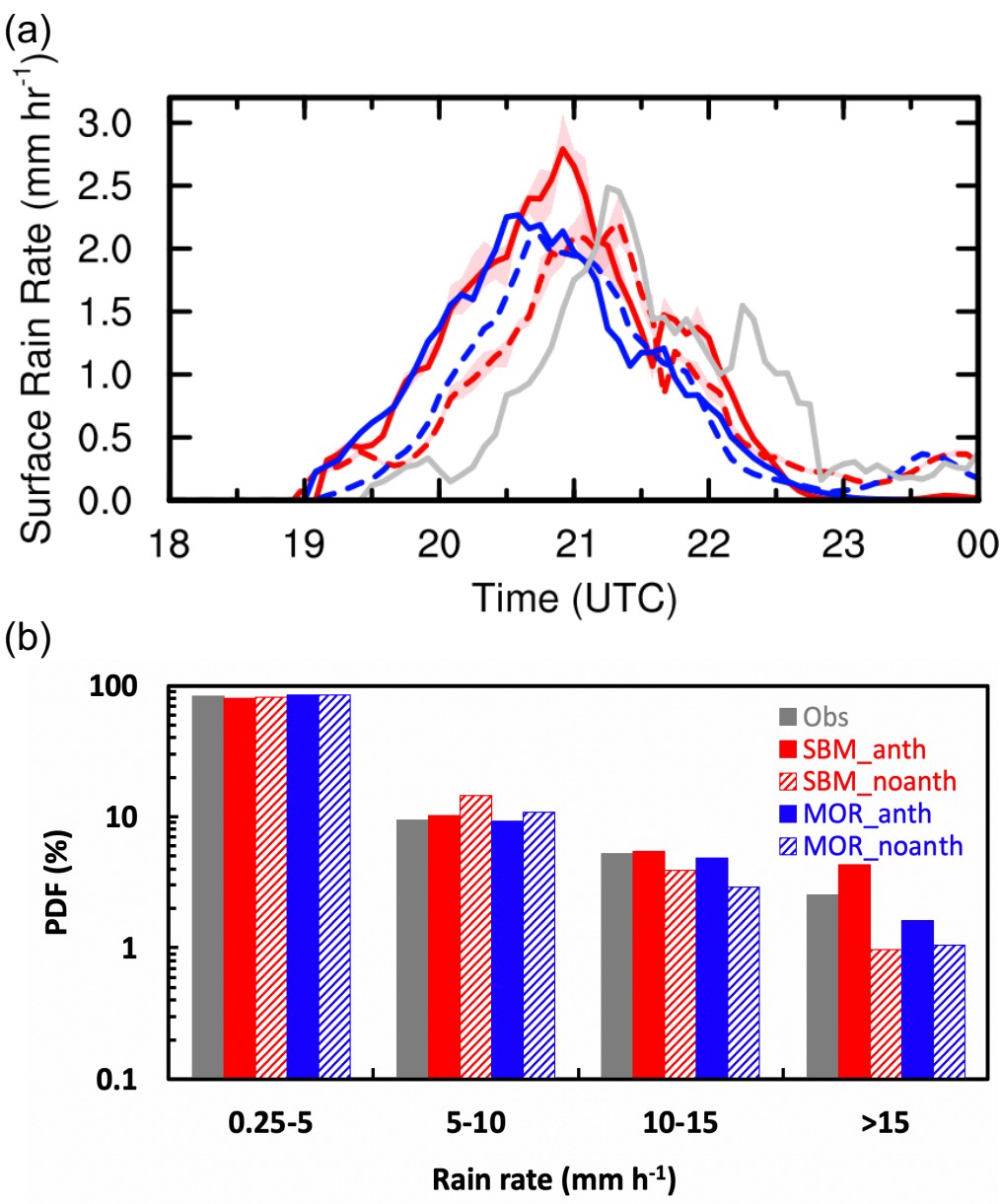

799

**Figure 9** (a) Time series of averaged surface rain rate (unit: mm h⁻¹) and (b) PDFs of rain rate for
the values larger than 0.25 mm h⁻¹ over the study area (red box in Fig. 7) from observation (grey),
SBM_anth and SBM_noanth (red), MOR_anth and MOR_noanth (blue) from 1800UTC, 19 Jun
2013 to 0000 UTC, 20 Jun 2013. The observed precipitation rate is obtained by NEXRAD
retrieved rain rate. Both observation and model data are in every 5-min frequency. The results are
the three ensemble means. The shaded areas mark the spread of the ensemble members.

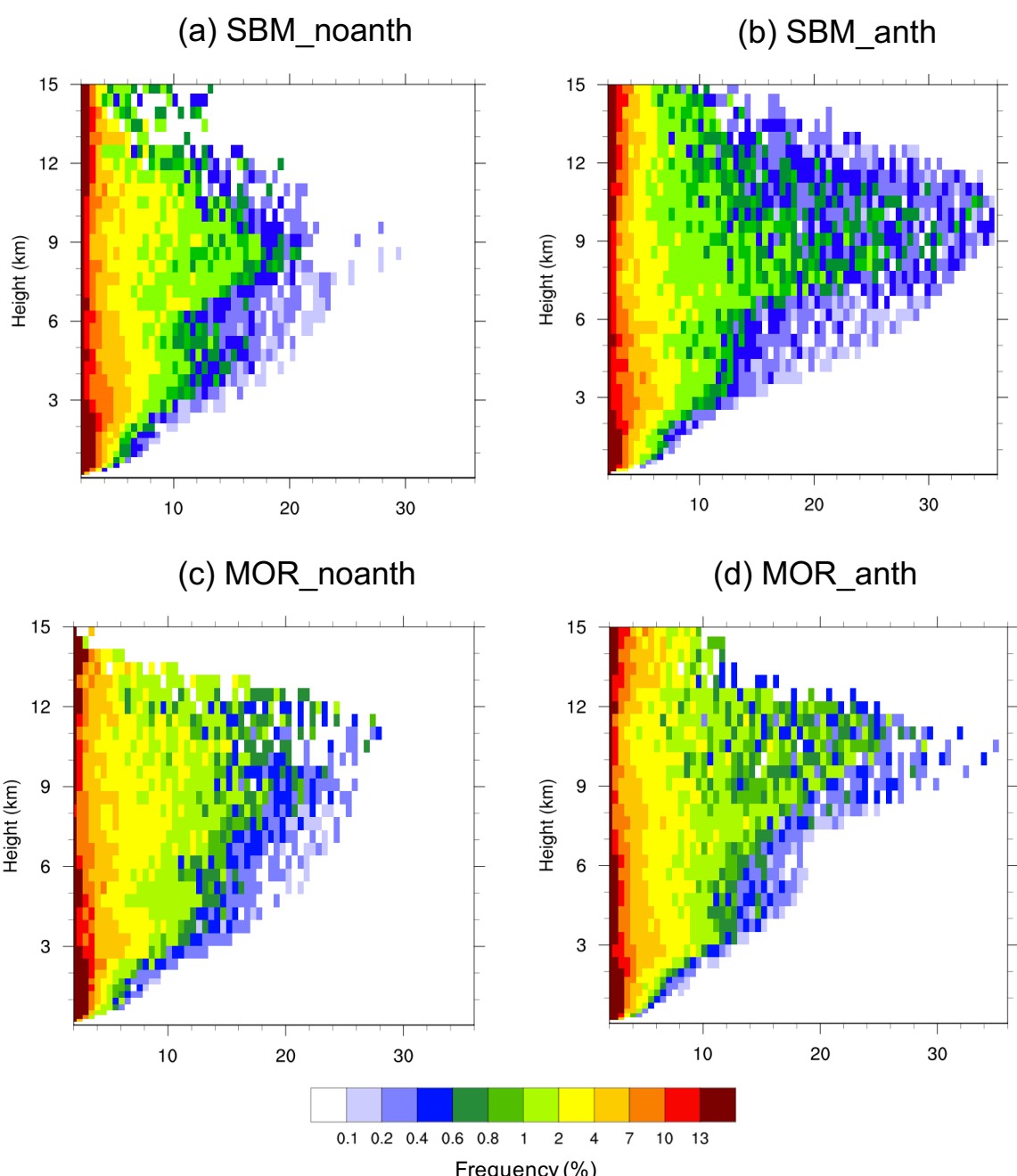

809

**Figure 10** CFADs of updraft velocity (unit: m s⁻¹) for values larger than 2 m s⁻¹ from (a) SBM_noanth, (b) SBM_anth, (c) MOR_noanth, and (d) MOR_anth over the study area (red box in Fig. 7) during the strong convection period (2000 – 2300 UTC, 19 Jun 2013). The results are the three ensemble means.

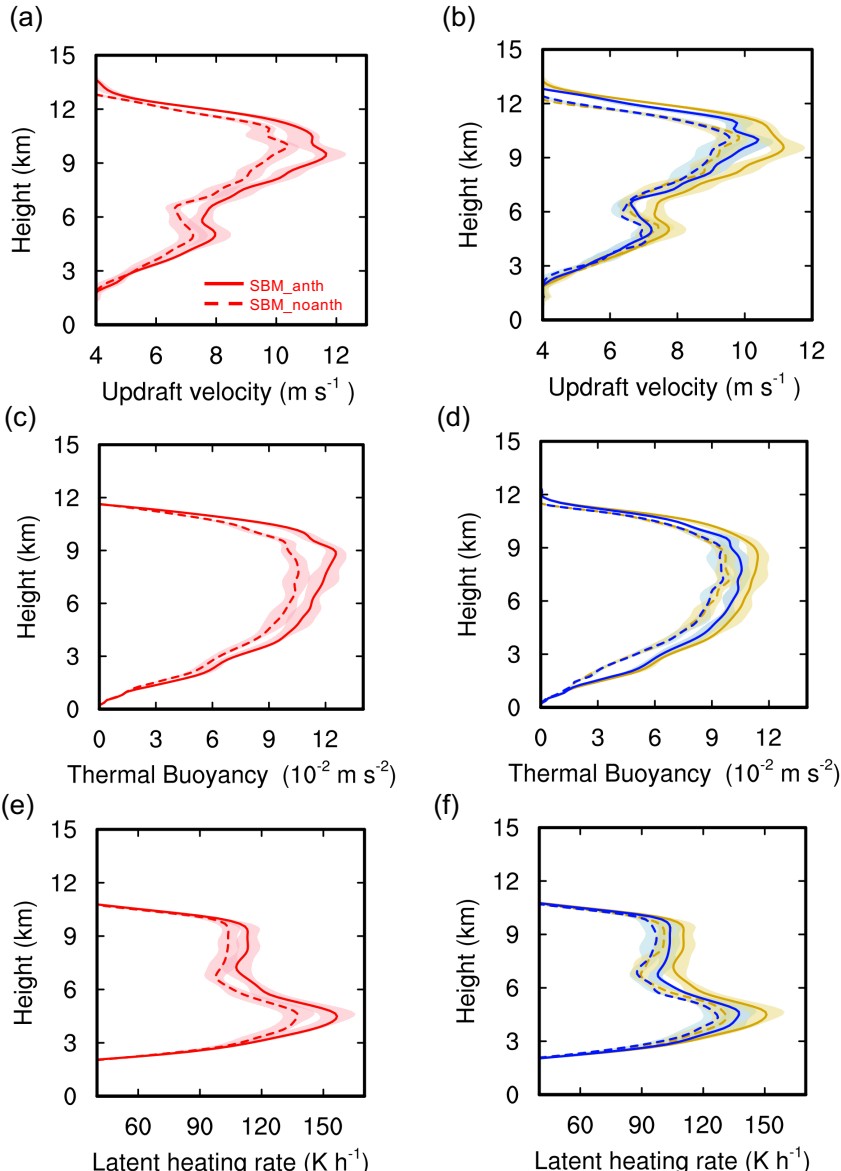


**Figure 11** Vertical profiles of (a,b) updraft velocity (unit: m s⁻¹) , (c,d) thermal buoyancy (unit:
m s⁻²) and (e,f) total latent heating rate (unit: K h⁻¹) averaged over the top 25 percentiles (i.e., from
75th to 100th) of the updrafts with velocity greater than 2 m s⁻¹ from the simulations SBM_anth
and SBM_noanth (red), MOR_anth and MOR_noanth (blue),  and MOR_SS_anth and
MOR_SS_noanth (orange) over the study area (red box in Fig. 7) during the strong convection
period (2000 – 2300 UTC, 19 Jun 2013). The results are the three ensemble means. The shaded
areas mark the spread of the ensemble members.

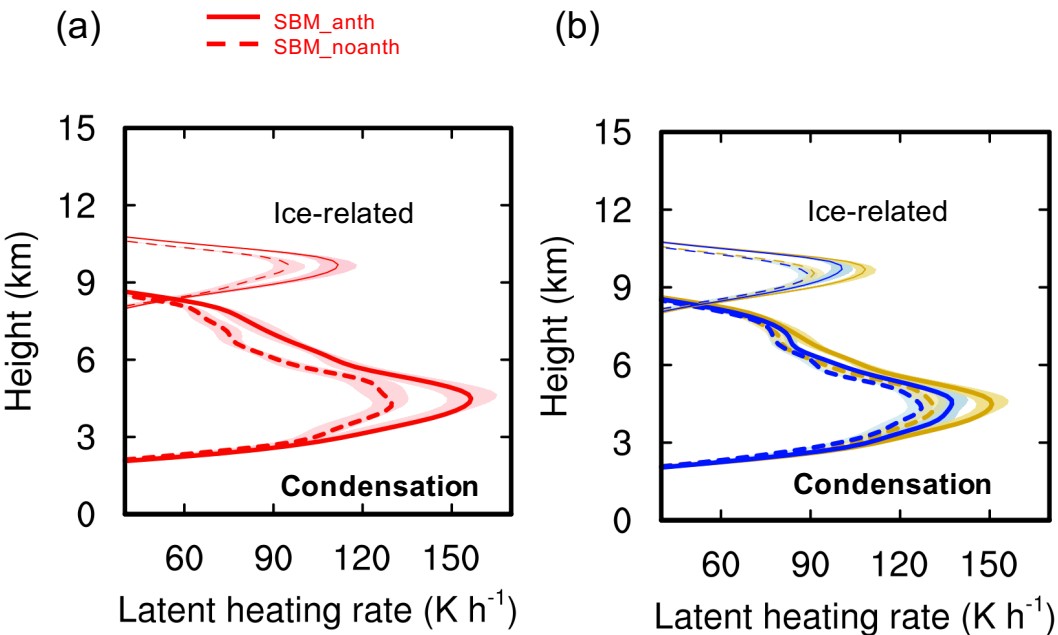


**Figure 12** Vertical profiles of condensation heating rate (thick lines below 9 km; unit: K h$^{-1}$) and
ice-related latent heating rate (thin lines above 9 km; unit: K h$^{-1}$) averaged over the top 25
percentiles (i.e., 75th to 100th) of the updrafts with velocity greater than 2 m s$^{-1}$ from the
simulations (a) SBM_anth and SBM_noanth (red), and (b) MOR_anth and MOR_noanth (blue),
and MOR_SS_anth and MOR_SS_noanth (orange) over the study area (red box in Fig. 7) during
the strong convection period (2000 – 2300 UTC, 19 Jun 2013). Data are processed in the same
way as Figure 11.

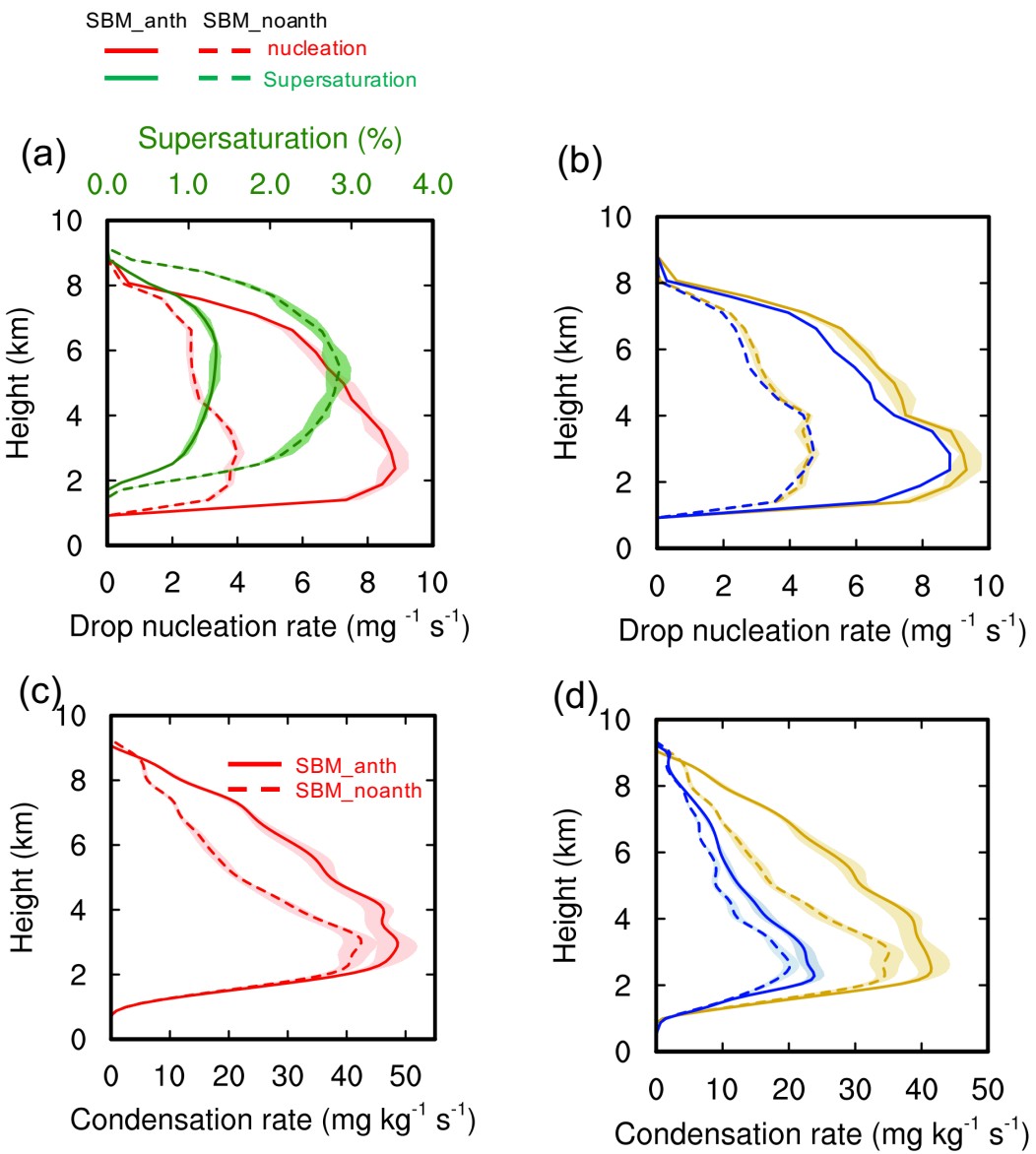


**Figure 13** Vertical profiles of (a) drop nucleation rate (red; unit: mg$^{-1}$ s$^{-1}$) and supersaturation with respect to water (green; unit: %) from SBM_anth and SBM_noanth, (b) drop nucleation rate (unit: mg$^{-1}$ s$^{-1}$) from MOR_anth and MOR_noanth (blue), and MOR_SS_anth and MOR_SS_noanth (orange), (c) condensation rate (unit: mg kg$^{-1}$ s$^{-1}$) from SBM_anth and SBM_noanth (red), and (d) the same as (c) but from MOR_anth and MOR_noanth (blue), and MOR_SS_anth and MOR_SS_noanth (orange), averaged over the top 25 percentiles (i.e., from 75[th] to 100[th]) of the updrafts with velocity greater than 2 m s$^{-1}$ over the study area (red box in Fig. 7) during the strong convection period (2000 – 2300 UTC, 19 Jun 2013). Data are processed in the same way as Figure 11.

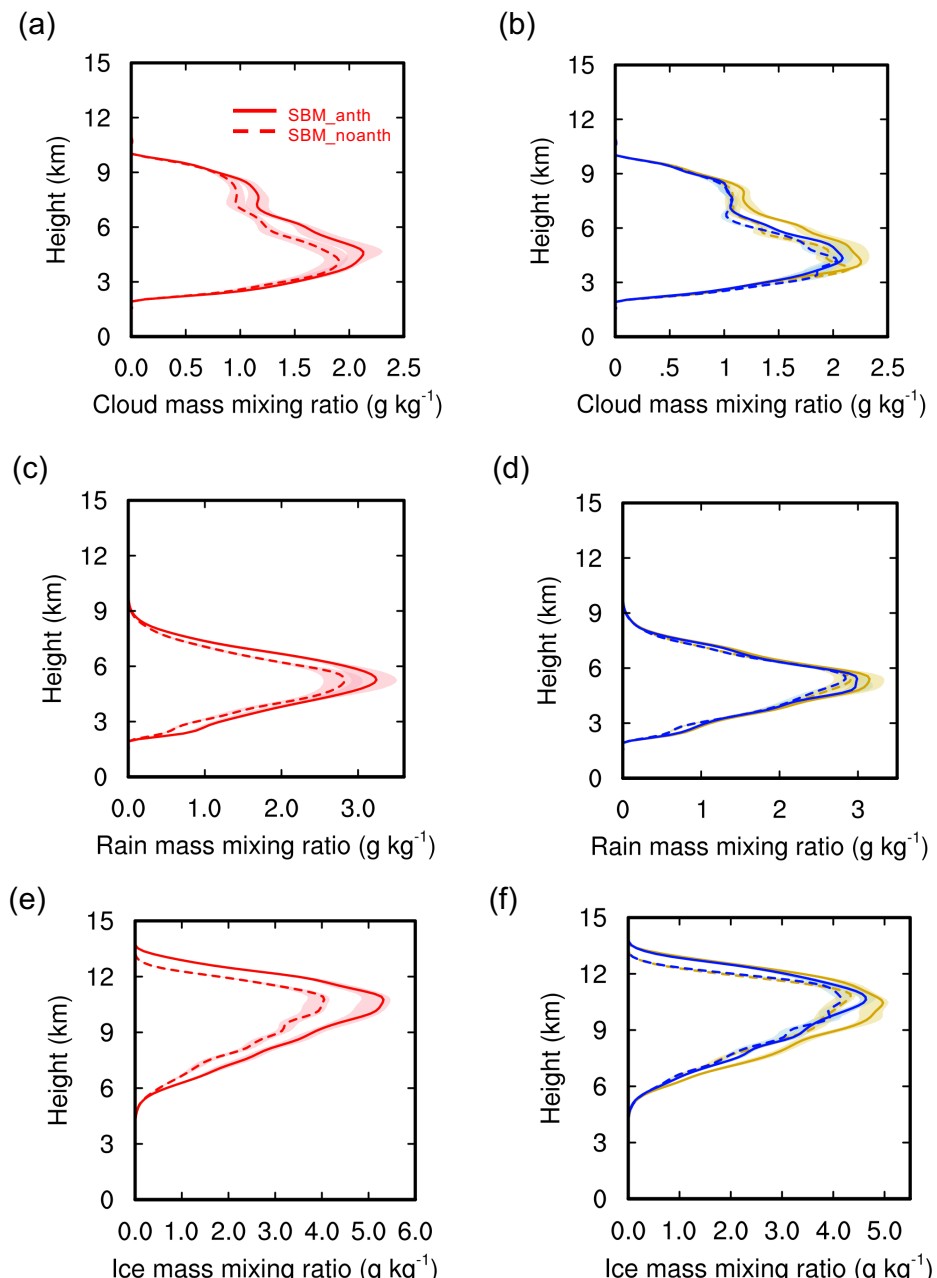


**Figure 14** Vertical profiles of (a, b) cloud droplet, (c, d) rain drop and (e, f) ice particle (including
ice, snow, and graupel) mass mixing ratios (unit: g kg$^{-1}$) averaged over the top 25 percentiles (i.e.,
75th to 100th) of the updrafts with value greater than 2 m s$^{-1}$ from the simulations SBM_anth and
SBM_noanth (red), MOR_anth and MOR_noanth (blue), and MOR_SS_anth and
MOR_SS_noanth (orange) over the study area (red box in Fig. 7) during the strong convection
period (2000 – 2300 UTC, 19 Jun 2013). Data are processed in the same way as Figure 11.

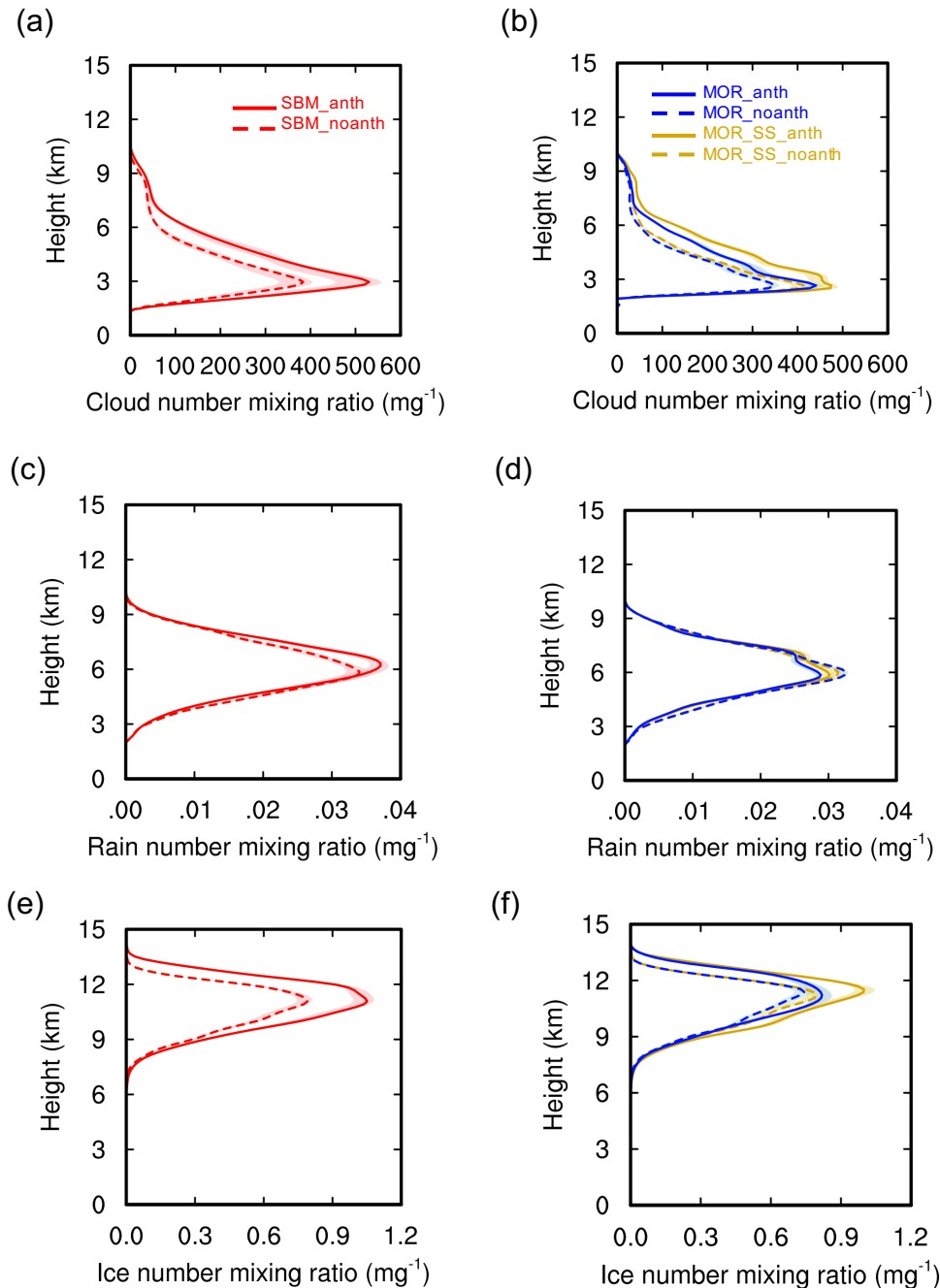


**Figure 15** Same as Figure 14, but for hydrometeor number mixing ratio.

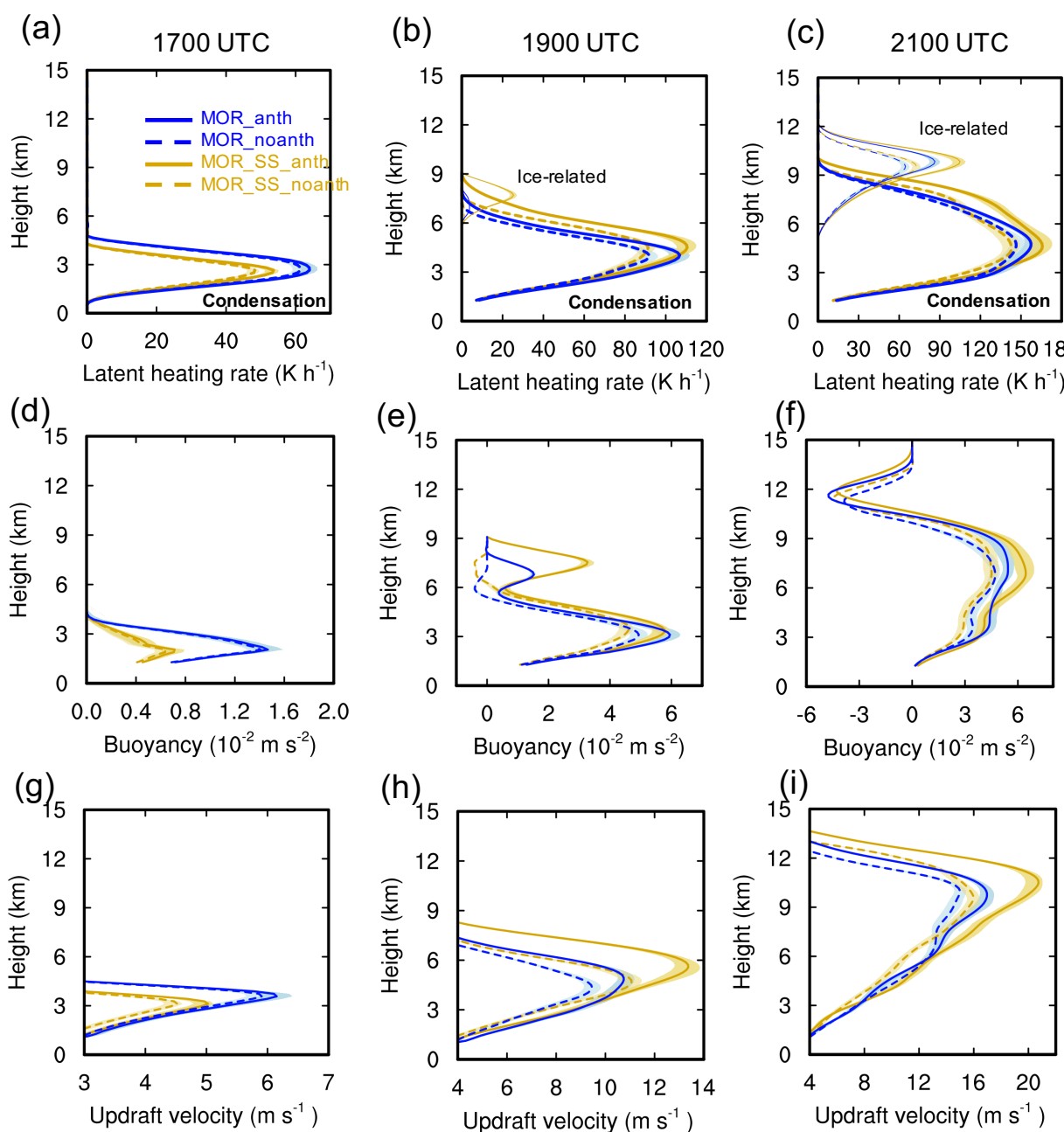


**Figure 16** Vertical profiles of (a-c) latent heating rate from condensation (weighted lines) and ice-related processes (freezing, riming, and deposition; thin lines), (d-f) total buoyancy, (g-i) updraft velocity averaged over the top 25 percentiles (i.e., 75th to 100th) of the updrafts with value greater than 2 ms$^{-1}$ from the simulations MOR_anth, MOR_noanth, MOR_SS_anth, and MOR_SS_noanth over the analysis domain as shown in the red box in Figure 7 at 1700 UTC, 1900 UTC and 2100 UTC. Data are processed in the same way as Figure 11.


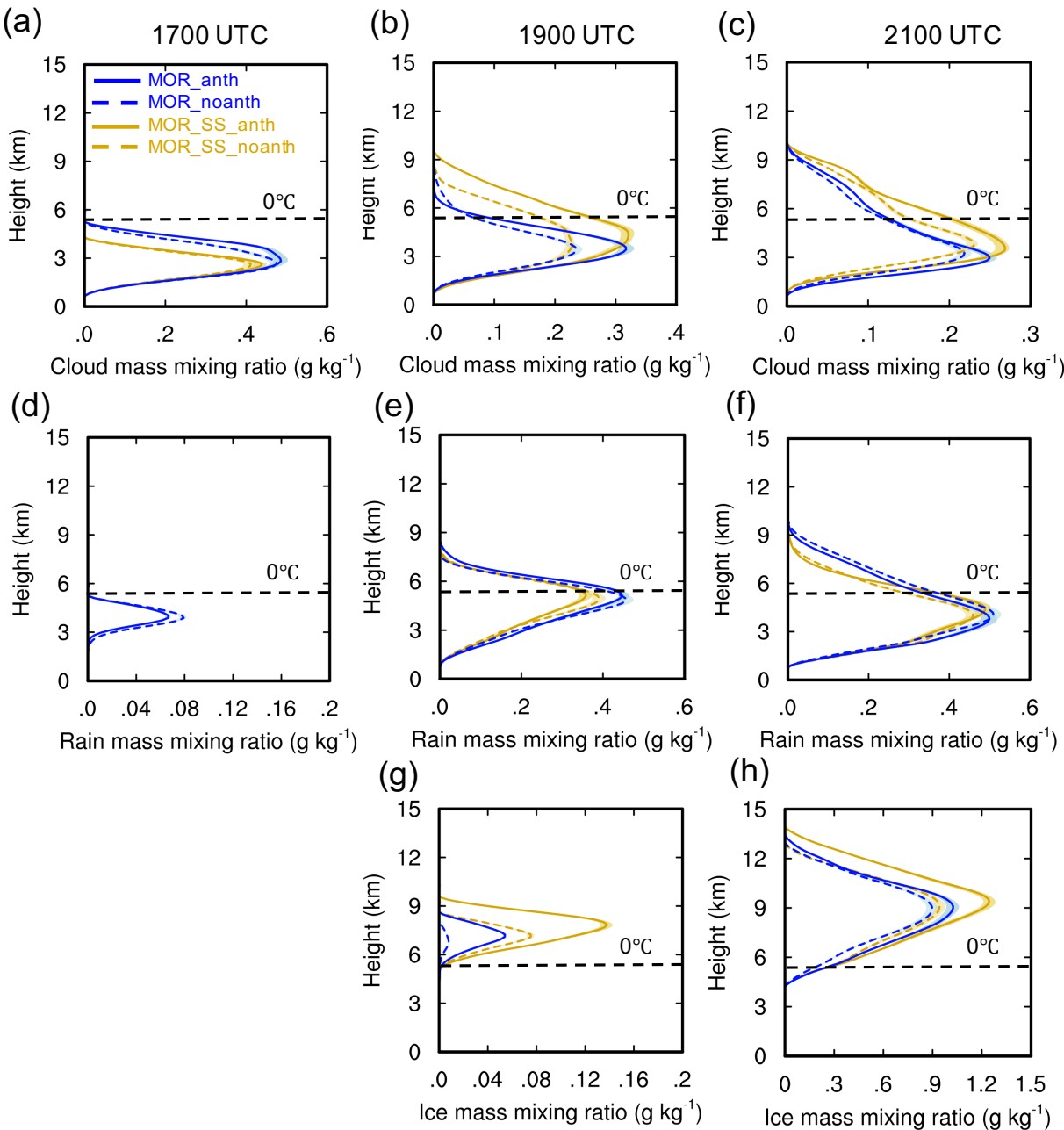


**Figure 17** Vertical profiles mass mixing ratios of (a-c) cloud droplet, (d-f) rain drop and (g-h) ice
particle (including ice, snow, and graupel) averaged over cloudy points (hydrometeor mass larger
than $10^{-5}$ kg kg$^{-1}$) from the simulations MOR_anth, MOR_noanth, MOR_SS_anth, and
MOR_SS_noanth over the analysis domain as shown in the red box in Figure 7 at 1700 UTC, 1900
UTC and 2100 UTC. Data are processed in the same way as Figure 11.


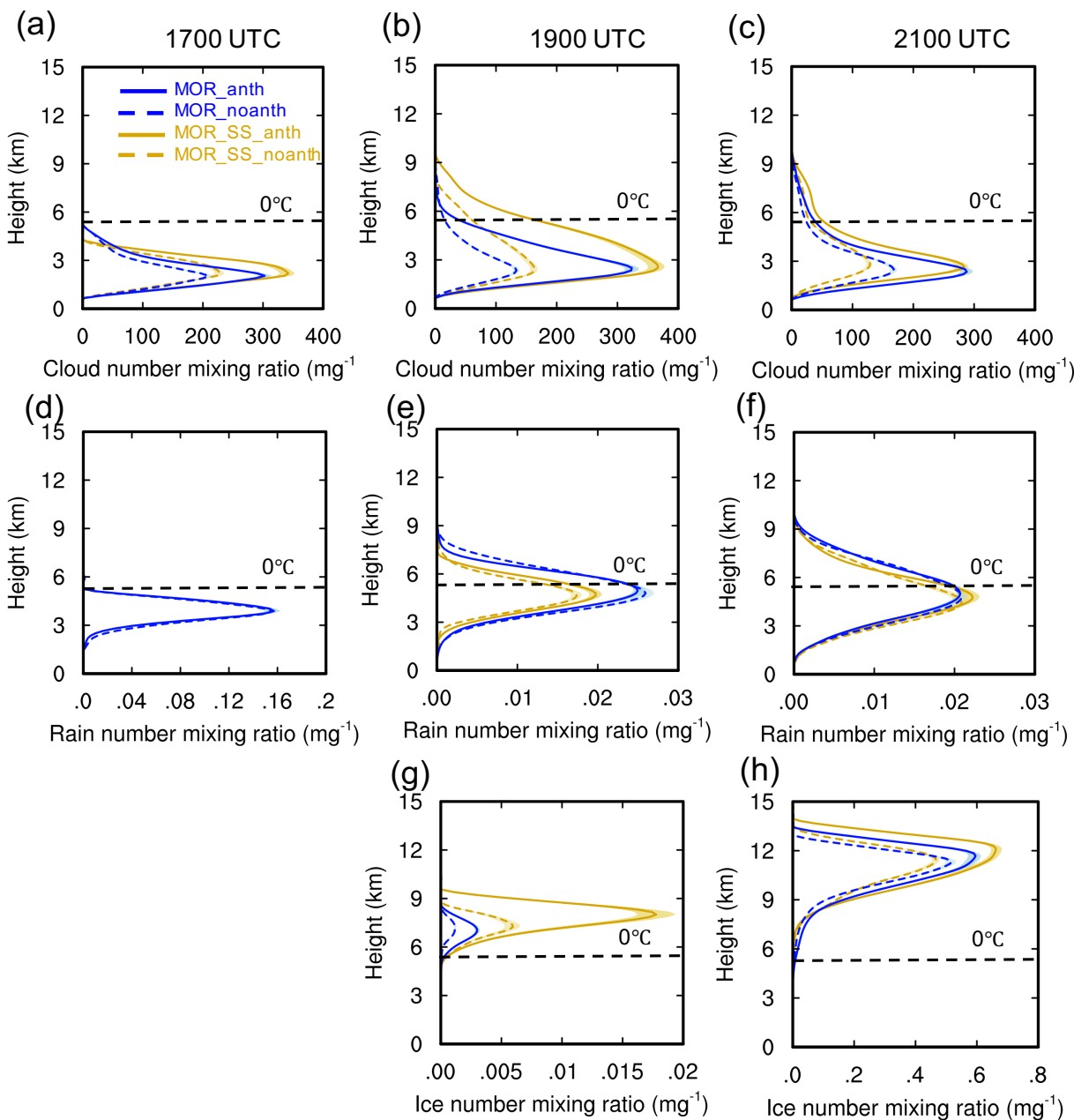


**Figure 18** Same as Figure 17, but for hydrometeor number mixing ratio.