# Peer review of "Impacts of Cloud Microphysics Parameterizations on Simulated Aerosol-Cloud-Interactions for Deep Convective Clouds over Houston"

_Atmospheric Chemistry and Physics, 2020_

## Referee Comment (RC1) · Anonymous Referee #1 · 5 Jun 2020

This is a WRF-Chem modeling study using a case of summertime convection in Houston from the ACPC Model Intercomparison Project. The focus is on the indirect aerosol effects on deep convection, using both SBM and Morrison microphysical schemes. The paper is certainly within the scope of ACP. It is well organized and clearly written, with adequate introduction and scientific review. The goals of the study, as elucidated in the first paragraph of section 5, are to (1) evaluate the performance of the WRF-Chem-SBM scheme, (2) explore the differences in aerosol effects on deep convective clouds produced by the SBM and Morrison schemes, and (3) explore the major factors responsible for the differences. The first two goals are descriptive in nature, they are fulfilled and clearly documented. However, I found the deductions made re-

garding the third goal to be questionable and poorly-supported by the data presented. The manuscript concludes that, the "warm-phase invigoration" effect is absent with the Morrison scheme, and this is "mainly due to limitations of the saturation adjustment approach for droplet condensation and evaporation calculation". While the saturation adjustment is probably the root cause, I find it unlikely that it is the DIRECT cause of the simulated sensitivities. Other processes have to be involved, and they need to be identified and properly analyzed. I'll elaborate on this in my specific comments. This flaw needs to be addressed before the manuscript is published.

Specific Comments:

There are three sets of model sensitivity tests using either realistic anthropogenic aerosol loadings or no anthropogenic aerosol: the explicit SBM scheme, the 2-moment Morrison scheme with saturation adjustment technic, and the Morrison scheme improved with a super saturation formula. The SBM scheme simulated stronger convection and more aerosol sensitivity compared with the original Morrison scheme, whereas the improved Morrison scheme produced results and sensitivities closer to the SBM results. The conclusion followed was that "...the saturation adjustment method for the condensation and evaporation calculation is mainly responsible for the limited aerosol effects with the Morrison scheme." This should be the correct conclusion, that the limitations in saturation adjustment are the root cause of the simulated differences in sensitivities. However, it cannot be the DIRECT cause. I can think of two pieces of evidence to support my assertion.

1. In the conclusion, the authors stated: "...the saturation adjustment method actually leads to a smaller condensation latent heating than the explicit calculation with supersaturation..." (L407). Fig. 12b was given to support the statement. However, saturation adjustment cannot be the direct reason for the smaller latent heating in Fig. 12b (or in any of the plots in Figs. 11~14). Figs. 11~14 only showed mean vertical profiles of various variables for the "top 25 percentiles" of the simulated updrafts. The main reason the Morrison scheme has smaller latent heating in Fig. 12b is not because

of the saturation adjustment, it is because the updrafts are weaker (Fig. 11 a, b). The dynamics already determined the differences in the latent heating, buoyancy, condensation rate, et al. shown in Figs. 11~14, not the other way around. In other words, the top 25 percentile of the updrafts are already weaker in the Morrison scheme simulation. As a result, latent heating should be weaker. Whether saturation adjustment causes this or not cannot be established by Figs. 11~ 14.

2. If saturation adjustment were the immediate/main cause of the simulated sensitivities, then the original Morrison scheme should produce stronger convection than SBM, given the same aerosol loading. This is because the saturation adjustment converts ALL supersaturation into cloud water, and thus should release the most latent heating among all schemes used. The fact that the SBM_anth case has much stronger convection than MOR_anth clearly precludes this possibility. If the authors plot Fig. 12 for the same vertical velocity (or super saturation), the Morrison scheme should have more latent heating, not less.

In conclusion, the saturation adjustment cannot be the direct cause of the simulated sensitivities. Something else must interact with it to cause these sensitivities. The authors actually observed the oddity of their conclusion in their conclusion, L401~L405. They noted that their study differs from Lebo et al. (2012). In this sense, Lebo et al. (2012) gave a feasible explanation, that the the "cold-phase invigoration" is in play together with saturation adjustment. The current case study may or may not have the same mechanism. Nevertheless, the authors need to find the missing link between the saturation adjustment, which produces the maximum possible latent heating by eliminating all super saturation, and the enhanced convection when super saturation is allowed.

Another result that puzzles me comes from Fig. 9a, where the high aerosol loading cases (SBM_anth and MOR_anth) rain earlier than the low aerosol cases. Why? The conventional wisdom is the opposite. High aerosol loading will produce more, smaller cloud droplets, reducing auto conversion and delaying surface rainfall onset. Can this

be checked and explained?

---

## Referee Comment (RC2) · Anonymous Referee #2 · 9 Jun 2020

Review of a manuscript entitled "Impacts of Cloud Microphysics Parameterizations on Simulated Aerosol-Cloud Interactions for Deep Convective Clouds over Houston" by Zhang, Fan, Li, and Rosenfeld, considered for publication in ACP, manuscript acp-2020-372.

Recommendation: not acceptable in its current form (major revision or rejection)

This manuscript presents results of numerical simulations that consider impacts of cloud microphysics parameterization on convective clouds near Houston. Overall, this is an impressive study that includes simulation and validation of aerosols that play key role in cloud dynamics and microphysics, and subsequently investigates the CCN im-

pact on convective dynamics. My main problem is with the context of this study and with the interpretation of model results. Obviously, the authors are strongly for the invigoration and I am one of those who oppose their views as scientifically unjustified. The manuscript should provide less biased view of the invigoration and needs to include additional analysis of model results as suggested in my specific comments.

Major comments:

1. The introduction needs to provide a better context for this work. A brief discussion of invigoration in the second paragraph of the introduction is misleading. It presents the authors view that is not supported by simple arguments and by other studies. For instance, the "cold-phase invigoration" as described in lines 42-45 is simply not possible because the latent heat released by freezing the cloud water carried across the melting level in the polluted case only balances the weight of the water carried upwards. So where does the invigoration come from? The explanation of the "warm-phase invigoration" is simply incorrect and it repeats the incorrect argument used in papers the authors cite. The latent heating does not depend on the droplet concentration and droplet radius as long as the updraft velocity does not change. This is strictly true when the in-cloud supersaturation is equal to the quasi-equilibrium supersaturation. The validity of the quasi-equilibrium supersaturation approximation has been argued in many studies, at least in the absence of ice (e.g., Politovich and Cooper JAS 1988). Such an incorrect interpretation is repeated in lines 334-337. I suggest the authors consult the recently accepted manuscript by Grabowski that provides a thorough discussion of the two invigorations, see section 2 there. The manuscript is available on EOR in JAS (https://journals.ametsoc.org/doi/abs/10.1175/JAS-D-20-0012.1). I also suggest the authors consult and cite a paper by Varble (JAS 2018) for a less biased discussion of the invigoration problem.

2. The discussion of bulk versus bin microphysics starting in l. 61 misses an important point: not all bulk schemes apply saturation adjustment. For instance, the scheme of Morrison and Grabowski (JAS 2007, 2008a,b) allows supersaturation to evolve. The

scheme shows a good agreement with bin microphysics in simple tests. This is important for the context of simulations described in the manuscript under review.

3. The setup of model simulations is not clear to me. I understand the motivation for applying the same boundary conditions for the inner domain in all simulations and hence using the MERRA-2 data on the inner-domain boundaries. However, how this is done with the outer domain present in not clear to me. Is it fair to say that outer domain is ran initially without the inner domain to simulate aerosol evolution and then the inner domain simulations are run without the outer domain using boundary conditions from MERRA-2 for the dynamics and thermodynamics, and applying the outer domain data for the aerosols? In other words, simulations with the two nested domains are actually never run together, correct? If my understanding is correct, then the description on p. 8 and 9 needs to change along my suggestion above. Also, it would be useful to describe in more detail the vertical grid structure. The 51 levels suggest quite a low vertical resolution.

4. The description of the simulation setup mentions 3-member ensembles. However, the ensemble information is never shown in the discussion of results. I think this is important because one may wonder to what extent a specific realization of the convection development affects the comparison. In other words, are the differences systematic or coincidental? All profiles shown in the figures should include the ensemble spread. Also, Fig. 7 should show all ensemble members and not just one realization. Specifically, is the more organized bin microphysics convection present in all ensemble members when compared to a more scattered bulk convection, or is this true only for the example shown in Fig. 7?

5. Although never mentioned in the manuscript, the vertical resolution near the cloud base is too low to properly resolve CCN activation in the bin scheme. It is well known that the vertical grid length around 10 m is needed to resolve the cloud base supersaturation maximum. Poor representation of cloud base activation affects droplet concentrations. In fact, droplet concentrations simulated by the two schemes are never

compared in the paper. This key parameter should be analyzed and presented. Should the bin scheme use parameterization of the cloud base CCN activation as the Morrison scheme?

6. Saturation adjustment and its role in the simulations. I think this aspect is poorly represented in the manuscript. First, one needs to clearly explain that saturation adjustment affects cloud buoyancy and thus simulated vertical velocity. The impact on the cloud buoyancy has been shown theoretically in Grabowski and Jarecka (JAS 2015) and discussed in the context of deep convection simulation in Grabowski and Morrison (JAS 2017). There are two aspects: 1) the increase of the vertical velocity because of the increased buoyancy (that does lead to the increased condensation), and 2) the increase of the condensation rate when the updraft is the same (this is because reducing supersaturation to zero gives more condensation). One way to separate the two effects is to show the condensation rate for a given vertical velocity (at a given height) and then repeat it for different vertical velocities. And do it separately for bin and bulk schemes. I expect that in undiluted or weakly diluted cloudy volumes the condensation rate is similar for the same vertical velocity in the two schemes and for the two aerosol conditions. I leave it to the authors to figure out what it means if my prediction turns out correct. Note that such an analysis eliminates the impact of different convection realizations and properly demonstrates the impact of the microphysics scheme on the condensation rate.

7. Saturation adjustment may also affect the way ice processes are simulated. Grabowski and Morrison (JAS 2017) document some possible impacts. This aspect begs the question about the representation of ice processes in the two schemes. I expect there are differences that are never discussed in the paper. Specifically, are ice concentrations similar between the two schemes? If there are significant differences, these have significant implications for the simulated cloud processes. As with the cloud droplet concentrations, this is never shown and discussed in the paper. 8. Related to some of the points above: How different is the supersaturation simulated

by the bin scheme from the quasi-equilibrium supersaturation below the freezing level? The quasi-equilibrium supersaturation can be derived from the local updraft velocity and droplet spectral characteristics. I expect the two are quite close in undiluted or weakly diluted cloudy volumes as suggested by other studies. If so, then please see comment 1 above.

Specific comments:

1. The abstract requires revisions after major comments above are addressed.

2. Grabowski and Jarecka (2015) show that the key impact in shallow convection simulations is the way saturation adjustment affects cloud edge evaporation (either resolved or because of the numerical diffusion). This aspect is never mentioned in the manuscript under review, but perhaps the cloud water evaporation plays some role, for instance, by driving stronger cloud-edge downdrafts when saturation adjustment is used.

3. L. 162: The grid length of the MERRA data should be mentioned here.

4. L 198: Rather than sending the reader to Lebo et al. (2012), please explain what is meant by "explicit representation of supersaturation over a time step". Is this close to the quasi-equilibrium supersaturation?

5. L. 229. I would not call the agreement shown in Fig. 3 "very good". This would imply that a color inside each circle is as in the background. This is not the case in several circles. Similar comment applies to Figs. 4 – 6. I understand the difficult task the model faces, but being honest about the simulation drawbacks would be appropriate. For instance, in Fig. 6, the temperature and wind simulations are closer to each other than to the observations.

6. What is "thermal buoyancy"? I think this is just "buoyancy", correct? Please change.

7. Fig. 8. To me, the simulations look close to each other and different than the NEXRAD picture. Is the plot for all three ensemble members? This needs to be clearly

stated.

8. Fig. 9. Again, are the plots for all ensemble members? How large is the variability among the ensemble members? I suggest to show the total accumulation in addition to the rate. Total accumulation tends to eliminate the impact of statistical fluctuations due to different flow realizations.

9. Fig. 10. Again, all ensemble members? How different are the figures for individual ensemble members? If they are much different, then more ensemble members are needed.

10. Fig. 13. Again, all ensemble members? What is the "drop nucleation rate"? Is this "CCN activation rate"? To what extent it is affected by the low vertical resolution?

11. For figures 1 -14, it is not clear how the averaging is done. For instance, if there are differences in the number of updrafts but their strength does not change, some of those profiles would change as well, correct? I think one has to clearly explain how the averaging is done to get a clear picture of the processes involved. And document the ensemble spread. As an example, see section 6 in Grabowski's manuscript (JAS 2020, Early Online Release) that discusses the incorrect interpretation of the enhanced lighting over south-east Asia shipping lines. More latent heating may simply come from a larger number of convective updrafts, not necessarily stronger updrafts.

---

## Author Comment (AC1) · 14 Aug 2020

Responses to Reviewer 1

This is a WRF-Chem modeling study using a case of summertime convection in Houston from the ACPC Model Intercomparison Project. The focus is on the indirect aerosol effects on deep convection, using both SBM and Morrison microphysical schemes. The paper is certainly within the scope of ACP. It is well organized and clearly written, with adequate introduction and scientific review. The goals of the study, as elucidated in the first paragraph of section 5, are to (1) evaluate the performance of the WRF-Chem-SBM scheme, (2) explore the differences in aerosol effects on deep convective clouds produced by the SBM and Morrison schemes, and (3) explore the major factors responsible for the differences. The first two goals are descriptive in nature, they are fulfilled and clearly documented. However, I found the deductions made regarding the third goal to be questionable and poorly-supported by the data presented. The manuscript concludes that, the "warm-phase invigoration" effect is absent with the Morrison scheme, and this is "mainly due to limitations of the saturation adjustment approach for droplet condensation and evaporation calculation". While the saturation adjustment is probably the root cause, I find it unlikely that it is the DIRECT cause of the simulated sensitivities. Other processes have to be involved, and they need to be identified and properly analyzed. I'll elaborate on this in my specific comments. This flaw needs to be addressed before the manuscript is published.

We thank the reviewer for your time and constructive comments. We have provided detailed responses point-by-point as below.

Specific Comments:
There are three sets of model sensitivity tests using either realistic anthropogenic aerosol loadings or no anthropogenic aerosol: the explicit SBM scheme, the 2-moment Morrison scheme with saturation adjustment technic, and the Morrison scheme improved with a super saturation formula. The SBM scheme simulated stronger convection and more aerosol sensitivity compared with the original Morrison scheme, whereas the improved Morrison scheme produced results and sensitivities closer to the SBM results. The conclusion followed was that ". . .the saturation adjustment method for the condensation and evaporation calculation is mainly responsible for the limited aerosol effects with the Morrison scheme." This should be the correct conclusion, that the limitations in saturation adjustment are the root cause of the simulated differences in sensitivities. However, it cannot be the DIRECT cause. I can think of two pieces of evidence to support my assertion.
1. In the conclusion, the authors stated: ". . .the saturation adjustment method actually leads to a smaller condensation latent heating than the explicit calculation with supersaturation. . ." (L407). Fig. 12b was given to support the statement. However, saturation adjustment cannot be the direct reason for the smaller latent heating in Fig. 12b (or in any of the plots in Figs. 11~14). Figs. 11~14 only showed mean vertical profiles of various variables for the "top 25 percentiles" of the simulated updrafts. The main reason the Morrison scheme has smaller latent heating in Fig. 12b is not because of the saturation adjustment, it is because the updrafts are weaker (Fig. 11 a, b). The dynamics already determined the differences in the latent heating, buoyancy, condensation rate, et al. shown in Figs. 11~14, not the other way around. In other words, the top 25 percentile of the updrafts are already weaker in the Morrison scheme simulation. As a result, latent heating should be weaker. Whether saturation adjustment causes this or not cannot be established by Figs. 11~ 14.

2. If saturation adjustment were the immediate/main cause of the simulated sensitivities, then the original Morrison scheme should produce stronger convection than SBM, given the same aerosol loading. This is because the saturation adjustment converts ALL supersaturation into cloud water, and thus should release the most latent heating among all schemes used. The fact that the SBM_anth case has much stronger convection than MOR_anth clearly precludes this possibility. If the authors plot Fig. 12 for the same vertical velocity (or super saturation), the Morrison scheme should have more latent heating, not less. In conclusion, the saturation adjustment cannot be the direct cause of the simulated sensitivities. Something else must interact with it to cause these sensitivities. The authors actually observed the oddity of their conclusion in their conclusion, L401~L405. They noted that their study differs from Lebo et al. (2012). In this sense, Lebo et al. (2012) gave a feasible explanation, that the the "cold-phase invigoration" is in play together with saturation adjustment. The current case study may or may not have the same mechanism. Nevertheless, the authors need to find the missing link between the saturation adjustment, which produces the maximum possible latent heating by eliminating all super saturation, and the enhanced convection when super saturation is allowed.

Although it is difficult to untangle the cause and effect relationship between the updraft and latent heating, the simulation sets of MOR and MOR_SS (i.e., MOR_anth vs MOR_SS_anth and MOR_noanth vs MOR_SS_noanth) allow us to attribute the differences in the response of latent heating and convective intensity to the method of saturation adjustment versus explicit treatment of supersaturation for condensation and evaporation. In Figs. r1, latent heating rates at 1700, 1900, and 2100 UTC are shown (first, second, and third columns, respectively). Note that at 1700 UTC, MOR_anth has stronger convection than MOR_SS_anth, suggesting that saturation adjustment removes all supersaturation and releases more latent heating. However, the development of deep convection is slower and weaker in MOR_anth because MOR_SS_anth has much more increase in latent heating significantly above the cloud base (one important feature of "warm-phase" invigoration) from 1700 to 1900 UTC. This suggests MOR_anth with supersaturation adjustment has limitation on "warm-phase" invigoration. After the convection in MOR_SS_anth become stronger than MOR_anth, the stronger updrafts further enhance the latent heating, suggesting a positive feedback. The saturation adjustment also shows weaker responses to different aerosol loadings with incorrect aerosol effect on diffusional growth (MOR_anth vs MOR_noanth).

[Figure]

Figure r1 Vertical profiles of (a-c) updraft velocity and (d-f) latent heating rate averaged over the top 25 percentiles (i.e., 75th to 100th) of the updrafts with value greater than 2 m s−1 from the simulations MOR_anth, MOR_noanth, MOR_SS_anth and MOR_SS_noanth over the analysis domain as shown in the red box in Figure 7 at 1700 UTC, 1900 UTC and 2100 UTC.

Another result that puzzles me comes from Fig. 9a, where the high aerosol loading cases (SBM_anth and MOR_anth) rain earlier than the low aerosol cases. Why? The conventional wisdom is the opposite. High aerosol loading will produce more, smaller cloud droplets, reducing auto conversion and delaying surface rainfall onset. Can this be checked and explained? The warm rain period is very weak (analysis box averaged rain rate at ~0.02 mm hr[-1]) and short (~10min), so the delay of precipitation is too hard to be shown from Fig. 9a. Actually, the clean cases initialize the rain about 5min earlier. The surface rain become large until the precipitation from deep clouds are formed. The time shift at ~1900 UTC from Fig. 9a may suggest that the high aerosol loading cases has faster transition from warm rain to mixed-phase precipitation.

---

## Author Comment (AC2) · 14 Aug 2020

**Responses to Reviewer 2**

This manuscript presents results of numerical simulations that consider impacts of cloud microphysics parameterization on convective clouds near Houston. Overall, this is an impressive study that includes simulation and validation of aerosols that play key role in cloud dynamics and microphysics, and subsequently investigates the CCN impact on convective dynamics. My main problem is with the context of this study and with the interpretation of model results. Obviously, the authors are strongly for the invigoration and I am one of those who oppose their views as scientifically unjustified. The manuscript should provide less biased view of the invigoration and needs to include additional analysis of model results as suggested in my specific comments.

We thank the reviewer for your time and constructive comments. We have provided detailed responses point-by-point as below.

**Major comments:**

1. The introduction needs to provide a better context for this work. A brief discussion of invigoration in the second paragraph of the introduction is misleading. It presents the authors view that is not supported by simple arguments and by other studies. For instance, the "cold-phase invigoration" as described in lines 42-45 is simply not possible because the latent heat released by freezing the cloud water carried across the melting level in the polluted case only balances the weight of the water carried upwards. So where does the invigoration come from? The explanation of the "warm-phase invigoration" is simply incorrect and it repeats the incorrect argument used in papers the authors cite. The latent heating does not depend on the droplet concentration and droplet radius as long as the updraft velocity does not change. This is strictly true when the in-cloud supersaturation is equal to the quasi-equilibrium supersaturation. The validity of the quasi-equilibrium supersaturation approximation has been argued in many studies, at least in the absence of ice (e.g., Politovich and Cooper JAS 1988). Such an incorrect interpretation is repeated in lines 334-337. I suggest the authors consult the recently accepted manuscript by Grabowski that provides a thorough discussion of the two invigorations, see section 2 there. The manuscript is available on EOR in JAS

(https://journals.ametsoc.org/doi/abs/10.1175/JAS-D-20-0012.1). I also suggest the authors consult and cite a paper by Varble (JAS 2018) for a less biased discussion of the invigoration problem.

First, for the "cold-phase invigoration", the latent heat released by freezing is not compensated by the mass of cloud water. The buoyancy is mainly contributed from the thermal term corresponding with latent heating and hydrometeor loading. The freezing process does not lead to any change in the hydrometeor loading, so drop freezing increases the buoyancy through latent heat release of fusion. Also, the ice particles after freezing may participate in other ice microphysical processes (i.e., deposition and riming) to further release latent heat.

Second, for the "warm-phase invigoration", there is limitation to use quasi-equilibrium supersaturation as an estimation of in-cloud supersaturation. Previous studies (e.g., Korolev and Mazin 2003, Pinsky et al. 2013, Politovich and Cooper 1988) showed that the quasi-steady assumption is invalidated in conditions of (a) low droplet concentrations (pristine condition) and (b) intense condensation and evaporation (e.g., cloud base and strong updrafts) due to long relaxation time (larger than a few seconds). Figure r2 shows modeled supersaturation and quasi-equilibrium supersaturation, updraft velocity, phase relaxation time and droplet number

concentration from the simulations SBM\_noanth and SBM\_anth. The calculation of  $S_{eq}$  and phase relaxation time  $\tau$  follows Eq. 3 and Eq. 4 in Pinsky et al. (2013), respectively. Seq is much higher than the true supersaturation especially above cloud base (Fig. r2a) where low droplet number concentration (Fig. r2c) and strong updrafts (Fig. r2b) are seen. The averaged  $\tau$  are around 10-15 s. The exact closed supersaturation equation (Eq. 6 in Pinsky et al. 2013) shows that condensation depends on droplet number and size, and more droplets in the polluted clouds increase condensation and decrease supersaturation. Although  $S_{eq}$  in SBM\_anth is indeed much smaller than SBM\_noanth, assuming  $S \approx S_{eq}$  would lead to some invigoration effect but the effect would be much reduced because the condensation rate in the clean condition would be significantly overestimated by  $S_{eq}$ .

Figure r2 Vertical profiles of (a) supersaturation and quasi-equilibrium supersaturation, (b) updraft velocity, (c) phase relaxation time and (d) Droplet number concentration averaged over

the updrafts with value greater than 2 m s-1 from the simulations SBM\_anth and SBM\_noanth, MOR\_SS\_anth and MOR\_SS\_noanth over the analysis domain as shown in the red box in Figure 7 during the strong convection period (2000 – 2300 UTC, 19 Jun 2013). The shaded area marks the spread of ensemble runs.

2. The discussion of bulk versus bin microphysics starting in l. 61 misses an important point: not all bulk schemes apply saturation adjustment. For instance, the scheme of Morrison and Grabowski (JAS 2007, 2008a,b) allows supersaturation to evolve. The scheme shows a good agreement with bin microphysics in simple tests. This is important for the context of simulations described in the manuscript under review.

Thank you for pointing out this. We added "Instead of using saturation adjustment, some bulk schemes allow supersaturation to evolve (e.g., Morrison and Grabowski 2007, 2008) and show a good agreement with bin microphysics in simple tests." after the discussion of saturation adjustment.

3. The setup of model simulations is not clear to me. I understand the motivation for applying the same boundary conditions for the inner domain in all simulations and hence using the MERRA-2 data on the inner-domain boundaries. However, how this is done with the outer domain present in not clear to me. Is it fair to say that outer domain is ran initially without the inner domain to simulate aerosol evolution and then the inner domain simulations are run without the outer domain using boundary conditions from MERRA-2 for the dynamics and thermodynamics, and applying the outer domain data for the aerosols? In other words, simulations with the two nested domains are actually never run together, correct? If my understanding is correct, then the description on p. 8 and 9 needs to change along my suggestion above. Also, it would be useful to describe in more detail the vertical grid structure. The 51 levels suggest quite a low vertical resolution.

Yes, you are right. Two domain simulations were run separately and an important purpose of running outer domain is to get a good estimation of aerosol fields to feed to inner domain. The 51 vertical grid levels allow 50-100m resolution below 2km and ~500m above, which is not very high resolution but not too bad.

4. The description of the simulation setup mentions 3-member ensembles. However, the ensemble information is never shown in the discussion of results. I think this is important because one may wonder to what extent a specific realization of the convection development affects the comparison. In other words, are the differences systematic or coincidental? All profiles shown in the figures should include the ensemble spread. Also, Fig. 7 should show all ensemble members and not just one realization. Specifically, is the more organized bin microphysics convection present in all ensemble members when compared to a more scattered bulk convection, or is this true only for the example shown in Fig. 7?

Line 222-225 stated that all analysis results for Domain 2 simulations in this study are the ensemble mean. As you suggested, we added shaded area for the ensemble spread for all profiles (Fig. 9a and Fig.11-14). All three ensemble members are added to Fig.7. Overall, MOR has more scattered convections then SBM.

5. Although never mentioned in the manuscript, the vertical resolution near the cloud base is too low to properly resolve CCN activation in the bin scheme. It is well known that the vertical grid

length around 10 m is needed to resolve the cloud base supersaturation maximum. Poor representation of cloud base activation affects droplet concentrations. In fact, droplet concentrations simulated by the two schemes are never compared in the paper. This key parameter should be analyzed and presented. Should the bin scheme use parameterization of the cloud base CCN activation as the Morrison scheme?

The vertical resolution near the cloud base is around 70m, which is not too bad. Increasing vertical levels would also increase a lot of computation cost for so many sensitivity tests and ensemble run. The hydrometeor number concentration is shown in Fig. r3 (will also add to the manuscript). The bin scheme has explicit supersaturation that can be used for CCN activation based on the Köhler theory, which is more realistic. For a baseline simulation of a real case, it is not good to simply change to the droplet activation parameterization as the Morrison scheme. We may try a sensitivity test by changing to the CCN activation parametrization as the Morrison scheme to isolate the responses of droplet activation and diffusional growth as a future work.

---

## Author Comment (AC3) · 14 Aug 2020

The file "Response to RC1" was posted accidentally (I was just doing a test). Please ignore it. I will post our final response very soon. Sorry for the inconvenience.

Yuwei

---

## Author Comment (AC4) · 14 Aug 2020

The file "Response to RC2" was posted accidentally (I was just doing a test). Please ignore it. I will post our final response very soon. Sorry for the inconvenience.

Yuwei
* * *

---

## Referee Comment (RC3) · Anonymous Referee #2 · 25 Aug 2020

**Rebuttal of the authors' responses to the Referee 2 comments:**

I appreciate the authors' responses to my initial comments. However, the response has fundamental flaws and thus I cannot consider the issues settled. I think the authors need to reconsider my comments about the invigoration and correct obvious flaws in their arguments. Below, I copy in blue their original responses and provide my rebuttal.

First, for the "cold-phase invigoration", the latent heat released by freezing is not compensated by the mass of cloud water. The buoyancy is mainly contributed from the thermal term corresponding with latent heating and hydrometeor loading. The freezing process does not lead to any change in the hydrometeor loading, so drop freezing increases the buoyancy through latent heat release of fusion. Also, the ice particles after freezing may participate in other ice microphysical processes (i.e., deposition and riming) to further release latent heat.

The authors clearly misunderstood my argument. The figure below illustrates the starting point for my argument:

[Figure]

The left part of the panel shows a cloudy parcel that rises through the melting (freezing) level in pristine conditions. The total liquid condensate above the freezing level is $q_c$. The right part of the figure shows situation when a similar parcel rises in polluted conditions. Because of the less efficient warm rain processes, the parcel above the freezing level carries more liquid condensate, $q_c + \delta q_c$. Freezing of $\delta q_c$ in the authors' opinion (and in many other papers) is the reason for the invigoration. However, to carry $\delta q_c$ across the freezing level requires extra buoyancy when compared to the left panel. As shown in section 2a of Grabowski and Morrison (2020), the two effects approximately balance each other. It follows the original sentences in the manuscript under review that say: "… a well-known theory is that increasing aerosol concentrations can suppress warm rain as a result of increased droplet number but reduced droplet size. This allows more cloud water to be lifted to a higher altitude wherein the freezing of this larger amount of cloud water induces larger latent heating associated with stronger ice microphysical processes, thereby invigorating convective updrafts…" are simply incorrect and require additional

explanations. For instance, the invigoration would be possible if the frozen condensate was removed through precipitation processes.

That said, I really do not think referring to invigoration is needed for this manuscript. If the authors insist, then the introduction and references to the invigoration in the text should provide a less biased discussion, for instance, as in Grabowski and Morrison JAS papers and as in Varble (JAS 2018).

Second, for the "warm-phase invigoration", there is limitation to use quasi-equilibrium supersaturation as an estimation of in-cloud supersaturation. Previous studies (e.g., Korolev and Mazin 2003, Pinsky et al. 2013, Politovich and Cooper 1988) showed that the quasi-steady assumption is invalidated in conditions of (a) low droplet concentrations (pristine condition) and (b) intense condensation and evaporation (e.g., cloud base and strong updrafts) due to long relaxation time (larger than a few seconds). Figure r2 shows modeled supersaturation and quasi-equilibrium supersaturation, updraft velocity, phase relaxation time and droplet number concentration from the simulations SBM_noanth and SBM_anth. The calculation of $S_{eq}$ and phase relaxation time $\tau$ follows Eq. 3 and Eq. 4 in Pinsky et al. (2013), respectively. Seq is much higher than the true supersaturation especially above cloud base (Fig. r2a) where low droplet number concentration (Fig. r2c) and strong updrafts (Fig. r2b) are seen. The averaged $\tau$ are around 10-15 s. The exact closed supersaturation equation (Eq. 6 in Pinsky et al. 2013) shows that condensation depends on droplet number and size, and more droplets in the polluted clouds increase condensation and decrease supersaturation. Although $S_{eq}$ in SBM_anth is indeed much smaller than SBM_noanth, assuming $S \approx S_{eq}$ would lead to some invigoration effect but the effect would be much reduced because the condensation rate in the clean condition would be significantly overestimated by $S_{eq}$.

[Figure]

Figure r2 Vertical profiles of (a) supersaturation and quasi-equilibrium supersaturation, (b) updraft velocity, (c) phase relaxation time and (d) Droplet number concentration averaged over the updrafts with value greater than 2 m s−1 from the simulations SBM_anth and SBM_noanth, MOR_SS_anth and MOR_SS_noanth over the analysis domain as shown in the red box in Figure 7 during the strong convection period (2000 – 2300 UTC, 19 Jun 2013). The shaded area marks the spread of ensemble runs.

The phase relaxation time scale and the quasi equilibrium supersaturation estimates in the authors' response above *are simply wrong*. Below I include a table from Politovich and Cooper (JAS 1988) that shows phase relaxation time scale for different combinations of droplet concentrations and radii. These values are much smaller than those shown in the authors' response.

TABLE 1. Time constant characterizing supersaturation.
(Values of $\tau = 1/(a_2 I)$ s for $p$ = 771 mb, $T$ = 4.3°C)

| Radius (μm) | Droplet concentration (cm$^{-3}$) | | | |
| --- | --- | --- | --- | --- |
| | 100 | 300 | 500 | 1000 |
| 2 | 14.1 | 4.7 | 2.8 | 1.4 |
| 3 | 8.7 | 2.9 | 1.7 | 0.87 |
| 5 | 4.9 | 1.6 | 0.98 | 0.49 |
| 10 | 2.3 | 0.77 | 0.46 | 0.23 |

The key question is why?

The explanation is relatively simple. The authors say that they use the formulas from Pinsky et al., eq. (4) therein. However, Pinsky et al. apply a simplified (and in my view incorrect) droplet growth equation that is different from the comprehensive formula used in Politovich and Cooper (JAS 1988). The key point is that one has to apply exactly the same droplet growth equation in the phase relaxation time scale calculation (and thus in the quasi-equilibrium supersaturation) as used in the numerical model. I expect Khain's bin microphysics applies a correct droplet growth formulation that is close to the one used in Politovich and Cooper (JAS 1988), and not the simplified droplet growth equation applied in Pinsky et al. The supersaturation simulated by the model can be compared to the diagnosed quasi-equilibrium supersaturation only *if exactly the same droplet growth equations are used in both*. This was the case for the relatively good agreement shown in Grabowski and Morrison (JAS 2017), at least below the freezing level, see Fig. 15 therein. In summary, the values shown in the authors' response above have to be corrected.

The above discussion requires the authors to modify their responses and revise their paper accordingly. Note that the second part of my rebuttal impacts some of my other original comments. I strongly object publication of the manuscript unless those comments are appropriately addressed.

---

## Author Response (AR1)

Responses to Reviewer 1

This is a WRF-Chem modeling study using a case of summertime convection in Houston from the ACPC Model Intercomparison Project. The focus is on the indirect aerosol effects on deep convection, using both SBM and Morrison microphysical schemes. The paper is certainly within the scope of ACP. It is well organized and clearly written, with adequate introduction and scientific review. The goals of the study, as elucidated in the first paragraph of section 5, are to (1) evaluate the performance of the WRF-Chem-SBM scheme, (2) explore the differences in aerosol effects on deep convective clouds produced by the SBM and Morrison schemes, and (3) explore the major factors responsible for the differences. The first two goals are descriptive in nature, they are fulfilled and clearly documented. However, I found the deductions made regarding the third goal to be questionable and poorly-supported by the data presented. The manuscript concludes that, the "warm-phase invigoration" effect is absent with the Morrison scheme, and this is "mainly due to limitations of the saturation adjustment approach for droplet condensation and evaporation calculation". While the saturation adjustment is probably the root cause, I find it unlikely that it is the DIRECT cause of the simulated sensitivities. Other processes have to be involved, and they need to be identified and properly analyzed. I'll elaborate on this in my specific comments. This flaw needs to be addressed before the manuscript is published.

We thank the reviewer for recognizing the importance of our study and the nice summary. The reviewer's comment about through what interactions the saturation adjustment does not lead to the convective invigoration as the explicit supersaturation approach is very constructive. We have done more analysis with three figures added (Fig. 16-18). Our detailed response is provided as below.

Specific Comments:
There are three sets of model sensitivity tests using either realistic anthropogenic aerosol loadings or no anthropogenic aerosol: the explicit SBM scheme, the 2-moment Morrison scheme with saturation adjustment technic, and the Morrison scheme improved with a super saturation formula. The SBM scheme simulated stronger convection and more aerosol sensitivity compared with the original Morrison scheme, whereas the improved Morrison scheme produced results and sensitivities closer to the SBM results. The conclusion followed was that ". . .the saturation adjustment method for the condensation and evaporation calculation is mainly responsible for the limited aerosol effects with the Morrison scheme." This should be the correct conclusion, that the limitations in saturation adjustment are the root cause of the simulated differences in sensitivities. However, it cannot be the DIRECT cause. I can think of two pieces of evidence to support my assertion.
1. In the conclusion, the authors stated: ". . .the saturation adjustment method actually leads to a smaller condensation latent heating than the explicit calculation with supersaturation. . ." (L407). Fig. 12b was given to support the statement. However, saturation adjustment cannot be the direct reason for the smaller latent heating in Fig. 12b (or in any of the plots in Figs. 11~14). Figs. 11~14 only showed mean vertical profiles of various variables for the "top 25 percentiles" of the simulated updrafts. The main reason the Morrison scheme has smaller latent heating in Fig. 12b is not because of the saturation adjustment, it is because the updrafts are weaker (Fig. 11 a, b).

The dynamics already determined the differences in the latent heating, buoyancy, condensation rate, et al. shown in Figs. 11~14, not the other way around. In other words, the top 25 percentile of the updrafts are already weaker in the Morrison scheme simulation. As a result, latent heating should be weaker. Whether saturation adjustment causes this or not cannot be established by Figs. 11~14.

2. If saturation adjustment were the immediate/main cause of the simulated sensitivities, then the original Morrison scheme should produce stronger convection than SBM, given the same aerosol loading. This is because the saturation adjustment converts ALL supersaturation into cloud water, and thus should release the most latent heating among all schemes used. The fact that the SBM_anth case has much stronger convection than MOR_anth clearly precludes this possibility. If the authors plot Fig. 12 for the same vertical velocity (or super saturation), the Morrison scheme should have more latent heating, not less. In conclusion, the saturation adjustment cannot be the direct cause of the simulated sensitivities. Something else must interact with it to cause these sensitivities. The authors actually observed the oddity of their conclusion in their conclusion, L401~L405. They noted that their study differs from Lebo et al. (2012). In this sense, Lebo et al. (2012) gave a feasible explanation, that the the "cold-phase invigoration" is in play together with saturation adjustment. The current case study may or may not have the same mechanism. Nevertheless, the authors need to find the missing link between the saturation adjustment, which produces the maximum possible latent heating by eliminating all super saturation, and the enhanced convection when super saturation is allowed.

Thanks for the constructive comments. We addressed (1) and (2) together here since both of them are for the same issue: why the saturation adjustment approach leads to smaller condensational heating than the explicit supersaturation approach in Morrison Scheme and through what interactions it did not lead to the convective invigoration as the explicit supersaturation approach did.

As added in Line 416-444, "Now we explain why the saturation adjustment approach leads to smaller condensational heating than the explicit supersaturation approach in Morrison Scheme and why it leads to a smaller sensitivity to aerosols compared with the explicit supersaturation approach. We examine the time evolution of latent heating, updraft, and hydrometeor properties. At the warm cloud stage at 1700 UTC, the saturation adjustment indeed produces more condensational latent heating which leads to larger buoyancy and stronger updraft intensity compared to the explicit supersaturation because of removing supersaturation (Fig. 16, left, blue vs. orange). By the time of 1900 UTC when the clouds have developed into mixed-phase clouds, the saturation adjustment produces less condensational heating and weaker convection than the explicit supersaturation approach (Fig. 16, middle). The results remain similarly later at the deep cloud stage 2100 UTC (Fig. 16, right).

How does this change happen from 1700 to 1900 UTC? At the warm cloud stage (17:00 UTC), the saturation adjustment produces droplets with larger sizes (up to 100% larger for the mean radius) than the explicit supersaturation because of more cloud water produced as a result of zeroing-out supersaturation at each time step (droplet formation is similar between the two cases as shown in Fig. 13). This results in much faster and larger warm rain, while with the explicit supersaturation rain number and mass are absent at 1700 UTC as shown in Fig. 17d and 18d). As a result, when evolving into the mixed-phase stage (19:00 UTC), much fewer cloud droplets are transported to the levels above the freezing level (Fig. 17b and 18b). Whereas with the explicit supersaturation, because of the delayed/suppressed warm rain and smaller droplets (the mean radius is decreased from 8 to 6 μm at 3 km), much more cloud droplets are lifted to the higher levels. Correspondingly, a few times higher total ice particle number and mass are seen compared with the saturation adjustment (Fig. 17g and 18g) because more droplets above the freezing level induce stronger ice processes (droplet freezing, riming, and deposition). This leads to more latent heat release (Fig. 16e), which increases the buoyancy and convective intensity. With the explicit supersaturation, increasing aerosols leads to a larger reduction in droplet size (up to 1 μm more in the mean radius) than the saturation adjustment, therefore more enhanced ice microphysical processes and the larger latent heat. Besides, the condensational heating is more enhanced by aerosols with the explicit supersaturation (Fig. 16). Together, a much larger sensitivity to aerosols is seen with the explicit supersaturation."

Another result that puzzles me comes from Fig. 9a, where the high aerosol loading cases (SBM_anth and MOR_anth) rain earlier than the low aerosol cases. Why? The conventional wisdom is the opposite. High aerosol loading will produce more, smaller cloud droplets, reducing auto conversion and delaying surface rainfall onset. Can this be checked and explained?

The warm rain is very weak (analysis box averaged rain rate at ~0.02 mm hr-1) and the time period is short (~10min), so the delay of warm rain is too hard to be shown from Fig. 9a. We do see the delay of warm rain by aerosols but only about 5 min (probably due to the humid condition of the case study). We have added the following clarifications to the revised manuscript in Line 335-339, "Note Fig. 9a shows that anthropogenic aerosols lead to an earlier start of the precipitation with both SBM and Morrison, which reflects the faster transition of warm rain to mixed-phase precipitation. We do see the delay of warm rain by aerosols but only about 5 min (probably due to the humid condition of the case study), which is difficult to be shown in Fig. 9a since averaged rain rate for the analysis box is ~0.02 mm hr-1 and the time period is very short (~10 min)".

Responses to Reviewer 2
This manuscript presents results of numerical simulations that consider impacts of cloud microphysics parameterization on convective clouds near Houston. Overall, this is an impressive study that includes simulation and validation of aerosols that play key role in cloud dynamics and microphysics, and subsequently investigates the CCN impact on convective dynamics. My main problem is with the context of this study and with the interpretation of model results. Obviously, the authors are strongly for the invigoration and I am one of those who oppose their views as scientifically unjustified. The manuscript should provide less biased view of the invigoration and needs to include additional analysis of model results as suggested in my specific comments.

We thank the reviewer for your time and constructive comments. Our detailed point-by-point responses are provided below. As the reviewer is one of those who oppose the convective invigoration concept, we are standing at the other side as one of those who support the concept based on our theoretical analysis and modeling studies (but we do not mean that it occurs in every case since in reality many other factors are in play). The two sides of arguments have been existing for a while, and it is not the role of this paper to resolve this issue. We have submitted a comment paper on Grabowski and Morrison (2020, 2016) to J. Atmos. Sci. to detail the theoretical analysis and modeling designs between the two arguments, which would allow both sides to further discuss there.

Major comments:
1. The introduction needs to provide a better context for this work. A brief discussion of invigoration in the second paragraph of the introduction is misleading. It presents the authors view that is not supported by simple arguments and by other studies. For instance, the "cold-phase invigoration" as described in lines 42-45 is simply not possible because the latent heat released by freezing the cloud water carried across the melting level in the polluted case only balances the weight of the water carried upwards. So where does the invigoration come from? The explanation of the "warm-phase invigoration" is simply incorrect and it repeats the incorrect argument used in papers the authors cite. The latent heating does not depend on the droplet concentration and droplet radius as long as the updraft velocity does not change. This is strictly true when the in-cloud supersaturation is equal to the quasi-equilibrium supersaturation. The validity of the quasi-equilibrium supersaturation approximation has been argued in many studies, at least in the absence of ice (e.g., Politovich and Cooper JAS 1988). Such an incorrect interpretation is repeated in lines 334-337. I suggest the authors consult the recently accepted manuscript by Grabowski that provides a thorough discussion of the two invigorations, see section 2 there. The manuscript is available on EOR in JAS (https://journals.ametsoc.org/doi/abs/10.1175/JAS-D-20-0012.1). I also suggest the authors consult and cite a paper by Varble (JAS 2018) for a less biased discussion of the invigoration problem.

We put our responses to the convective invigoration questions that the reviewer raised separately at the end of this file to avoid a distraction. Here, in the introduction, we have added text to provide different arguments existing in literature.
For the cold-phase invigoration, we have added in Line 51-56 "Grabowski and Morrison (2016; 2020) rejected this invigoration concept by arguing that the increase in the buoyancy by freezing is completely offset by the buoyancy for carrying the extra cloud water across the freezing level. However, Rosenfeld et al. (2008) showed that the buoyancy restores and increases after the precipitation of the ice hydrometeors that form upon freezing of the high supercooled liquid water content into large graupel and hail (Rosenfeld et al., 2008)".

For the warm-phase invigoration, we have added in Line 63-71 "Grabowski and Morrison (2020) proposed a different interpretation of the warm-phase invigoration from the literature listed above. They argued that condensation rates only depend on updraft velocity with the quasi-steady assumption (i.e., the true supersaturation is approximated with the equilibrium supersaturation), therefore they interpreted that it is the lower equilibrium supersaturation in polluted conditions that lead to a larger buoyancy, thus enhanced updraft speeds, and condensation. Several studies showed that the quasi-steady assumption is invalidated in the conditions of low droplet concentrations (Politovich and Cooper, 1988; Korolev and Mazin, 2003) or acceleration of vertical velocity (Pinsky et al., 2013)".

We also added text of "Meteorological buffering effects were also found for aerosol effects on convective clouds over a large region and sufficiently long-time (over a few days and weeks) simulations (Stevens and Feingold, 2009; van den Heever et al., 2011). Dagan et al. (2019) showed that the lifetimes of cloud systems are mostly much shorter than that and rarely reach this buffering state" in Line 75-79 and "Confidently isolating and quantifying an aerosol deep convective invigoration effect from observations requires very long-term measurements: data of 10 years are still not enough over the South Great Plains due to the large variability of meteorological conditions (Varble, 2018)" in Line 80-83.

2. The discussion of bulk versus bin microphysics starting in l. 61 misses an important point: not all bulk schemes apply saturation adjustment. For instance, the scheme of Morrison and Grabowski (JAS 2007, 2008a,b) allows supersaturation to evolve. The scheme shows a good agreement with bin microphysics in simple tests. This is important for the context of simulations described in the manuscript under review.

What we said is that saturation adjustment is an often-used approach in bulk scheme, so our description should have no problem. But we have added a sentence to describe the bulk schemes used the explicit supersaturation, i.e., "Some bulk schemes take the explicit supersaturation approach to allow supersaturation to evolve (e.g., Li et al., 2008; 2009a; Morrison and Grabowski 2007, 2008)." (Line 92-93).

3. The setup of model simulations is not clear to me. I understand the motivation for applying the same boundary conditions for the inner domain in all simulations and hence using the MERRA-2 data on the inner-domain boundaries. However, how this is done with the outer domain present in not clear to me. Is it fair to say that outer domain is ran initially without the inner domain to simulate aerosol evolution and then the inner domain simulations are run without the outer domain using boundary conditions from MERRA-2 for the dynamics and thermodynamics, and applying the outer domain data for the aerosols? In other words, simulations with the two nested domains are actually never run together, correct? If my understanding is correct, then the description on p. 8 and 9 needs to change along my suggestion above. Also, it would be useful to describe in more detail the vertical grid structure. The 51 levels suggest quite a low vertical resolution.

Yes, two nested domains were run separately, and the purpose of running outer domain is to get a good estimation of aerosol fields to feed to inner domain for the initial and boundary chemical and aerosol conditions. We have added a sentence in Line 179-181 to clearly state this, "The simulations for Domain 1 and Domain 2 are run separately and the Domain 1 simulations serve to provide the chemical and aerosol lateral boundary and initial conditions of Domain 2."

The 51 vertical grid levels allow 50-100m resolution below 2-km altitude and ~500 m above it (added in Line 178-179), which is not very high resolution but not too bad.

4. The description of the simulation setup mentions 3-member ensembles. However, the ensemble information is never shown in the discussion of results. I think this is important because one may wonder to what extent a specific realization of the convection development affects the comparison. In other words, are the differences systematic or coincidental? All profiles shown in the figures should include the ensemble spread. Also, Fig. 7 should show all ensemble members and not just one realization. Specifically, is the more organized bin microphysics convection present in all ensemble members when compared to a more scattered bulk convection, or is this true only for the example shown in Fig. 7?

We presented the ensemble mean results in most of the analysis results for Domain 2 simulations. As the reviewer suggested, in the revised manuscript, we added the shaded areas for the ensemble spread for all the profile figures (Fig. 9a and Fig.11-14). For the spatial distribution figure (Fig. 7), we now show the results for each ensemble member. Yes, SBM has more organized convection than MOR in all three ensemble members. This information has been added to Line 297-298, "All three ensemble members consistently show smaller but more scattered convective cells with the Morrison scheme compared with SBM".

5. Although never mentioned in the manuscript, the vertical resolution near the cloud base is too low to properly resolve CCN activation in the bin scheme. It is well known that the vertical grid length around 10 m is needed to resolve the cloud base supersaturation maximum. Poor representation of cloud base activation affects droplet concentrations. In fact, droplet concentrations simulated by the two schemes are never compared in the paper. This key parameter should be analyzed and presented. Should the bin scheme use parameterization of the cloud base CCN activation as the Morrison scheme?

The droplet nucleation rate (i.e., aerosol activation rate) was indeed shown in Figure 13. The droplet nucleation rates simulated by SBM is comparable with the parameterization used in Morrison scheme, as shown in Figure 13. In this response letter, we also further showed the spatial distribution of droplet number concentration at cloud base: droplet number concentration at cloud bases in SBM_anth are similar with the observation and MOR_anth in magnitudes, suggesting the SBM is doing an okay job in cloud base activation.

[Figure]

Fig. r1 CCN number concentration at cloud base from (a) VIIRS satellite retrieved at 1943 UTC (Rosenfeld et al. 2016) and model simulation (b) SBM_anth, (c) MOR_anth at 2000 UTC, 19 June 2013.

6. Saturation adjustment and its role in the simulations. I think this aspect is poorly represented in the manuscript. First, one needs to clearly explain that saturation adjustment affects cloud buoyancy and thus simulated vertical velocity. The impact on the cloud buoyancy has been shown theoretically in Grabowski and Jarecka (JAS 2015) and discussed in the context of deep convection simulation in Grabowski and Morrison (JAS 2017). There are two aspects: 1) the increase of the vertical velocity because of the increased buoyancy (that does lead to the increased condensation), and 2) the increase of the condensation rate when the updraft is the same (this is because reducing supersaturation to zero gives more condensation). One way to separate the two effects is to show the condensation rate for a given vertical velocity (at a given height) and then repeat it for different vertical velocities. And do it separately for bin and bulk schemes. I expect that in undiluted or weakly diluted cloudy volumes the condensation rate is similar for the same vertical velocity in the two schemes and for the two aerosol conditions. I leave it to the authors to figure out what it means if my prediction turns out correct. Note that such an analysis eliminates the impact of different convection realizations and properly demonstrates the impact of the microphysics scheme on the condensation rate.

For the role of saturation adjustment, we have added more analysis as shown in Fig. 16-18 and two paragraphs (Line 416-444) to the revised manuscript, also to address a comment from Reviewer #1.

"Now we explain why the saturation adjustment approach leads to smaller condensational heating than the explicit supersaturation approach in Morrison Scheme and why it leads to a smaller sensitivity to aerosols compared with the explicit supersaturation approach. We examine the time evolution of latent heating, updraft, and hydrometeor properties. At the warm cloud stage at 1700 UTC, the saturation adjustment indeed produces more condensational latent heating which leads to larger buoyancy and stronger updraft intensity compared to the explicit supersaturation because of removing supersaturation (Fig. 16, left, blue vs. orange). By the time of 1900 UTC when the clouds have developed into mixed-phase clouds, the saturation adjustment produces less condensational heating and weaker convection than the explicit supersaturation approach (Fig. 16, middle). The results remain similarly later at the deep cloud stage 2100 UTC (Fig. 16, right).

How does this change happen from 1700 to 1900 UTC? At the warm cloud stage (17:00 UTC), the saturation adjustment produces droplets with larger sizes (up to 100% larger for the mean radius) than the explicit supersaturation because of more cloud water produced as a result of zeroing-out supersaturation at each time step (droplet formation is similar between the two cases as shown in Fig. 13). This results in much faster and larger warm rain, while with the explicit supersaturation rain number and mass are absent at 1700 UTC as shown in Fig. 17d and 18d). As a result, when evolving into the mixed-phase stage (19:00 UTC), much fewer cloud droplets are transported to the levels above the freezing level (Fig. 17b and 18b). Whereas with the explicit supersaturation, because of the delayed/suppressed warm rain and smaller droplets (the mean radius is decreased from 8 to 6 µm at 3 km), much more cloud droplets are lifted to the higher levels. Correspondingly, a few times higher total ice particle number and mass are seen compared with the saturation adjustment (Fig. 17g and 18g) because more droplets above the freezing level induce stronger ice processes (droplet freezing, riming, and deposition). This leads to more latent heat release (Fig. 16e), which increases the buoyancy and convective intensity. With the explicit supersaturation, increasing aerosols leads to a larger reduction in droplet size (up to 1 µm more in the mean radius) than the saturation adjustment, therefore more enhanced ice microphysical processes and the larger latent heat. Besides, the condensational heating is more enhanced by aerosols with the explicit supersaturation (Fig. 16). Together, a much larger sensitivity to aerosols is seen with the explicit supersaturation".

To satisfy the reviewer's curiosity about the relationship of condensation rate and vertical velocity, we plot their relationships in the simulations with the two schemes and for the two aerosol conditions at two different heights over the period 16-18 UTC where the warm cloud dominated (Fig. r2 and r3). For the same updraft velocity, the Morrison scheme with the saturation adjustment predicted larger condensation rates compared with SBM as expected because reducing supersaturation to zero gives more condensation (Fig. r2-r3, left vs right). The larger condensation rate leads to larger buoyancy and therefore strong updraft velocity as shown in Fig. 17. With the anthropogenic aerosols added, the condensation rate is not changed much with the saturation adjustment at both altitudes (right panels in Fig. r2-r3) because the approach removes the dependence of condensation on droplet properties. However, in the bin scheme, we find that SBM_anth tends to have larger condensation rates for the same updrafts than SBM_noanth (Fig. r3, a vs c) above cloud base where the increase of cloud droplet number is significant (Fig. 15a). This clearly shows that higher droplet number has larger condensation rate for the same vertical velocity, which is different from what the reviewer predicted because the reviewer's argument is that that the condensation rate is only dependent on updraft, not droplet properties, which is true for saturation adjustment approach, not for the explicit calculation in SBM.

[Figure]

Figure r2 The relationship between condensation rate and updraft velocity at 1.7 km (near cloud base) for SBM_anth, SBM_noanth, MOR_anth and MOR_noanth at warm cloud stage (1600 – 1800 UTC).

[Figure]

Figure r3 Same as Figure r2, but for 3 km altitude.

7. Saturation adjustment may also affect the way ice processes are simulated. Grabowski and Morrison (JAS 2017) document some possible impacts. This aspect begs the question about the representation of ice processes in the two schemes. I expect there are differences that are never discussed in the paper. Specifically, are ice concentrations similar between the two schemes? If there are significant differences, these have significant implications for the simulated cloud processes. As with the cloud droplet concentrations, this is never shown and discussed in the paper.

See our response to comment #6. The saturation adjustment weakens the ice processes due to less droplets remaining for being lifted above freezing level as a result of efficient conversion from cloud droplet to rain because of larger condensational growth. We also add the findings of Grabowski and Morrison (JAS 2017) in the discussion session: "Grabowski and Morrison (2017) also showed that the saturation adjustment affected ice processes by producing larger ice particles with larger falling velocities compared with the explicit supersaturation approach, leading to the reduction of anvil clouds." (Line 477-479).

We have added a figure for the hydrometeor number concentrations (Figure 15) corresponding to the mass mixing ratios shown in Figure 14.

8. Related to some of the points above: How different is the supersaturation simulated by the bin scheme from the quasi-equilibrium supersaturation below the freezing level? The quasi-equilibrium supersaturation can be derived from the local updraft velocity and droplet spectral characteristics. I expect the two are quite close in undiluted or weakly diluted cloudy volumes as suggested by other studies. If so, then please see comment 1 above.

The quasi-equilibrium supersaturation is much larger than simulated supersaturation between 3-5 km with more than 10 s relaxation time, which is mainly due to low droplet number. Please see Fig. r5 for more details.

Specific comments:
1. The abstract requires revisions after major comments above are addressed.

A new key point has been added to the abstract. That is "Whereas such an effect is absent with the Morrison two-moment bulk microphysics, mainly because the saturation adjustment approach for droplet condensation and evaporation calculation removes the dependence of condensation on droplet properties and limits the ice processes by a more efficient conversion of droplets into raindrops, which leads to less cloud droplets being transported to the altitudes above the freezing level" (Line 25-29).

2. Grabowski and Jarecka (2015) show that the key impact in shallow convection simulations is the way saturation adjustment affects cloud edge evaporation (either resolved or because of the numerical diffusion). This aspect is never mentioned in the manuscript under review, but perhaps the cloud water evaporation plays some role, for instance, by driving stronger cloud-edge downdrafts when saturation adjustment is used.

We added a sentence to the discussion part: "The increased condensation is significant for the enhanced warm clouds when saturation adjustment is used. This is different from the points of Grabowski and Jarecka (2015) that the cloud edge evaporation effect is more important for the nonprecipitating shallow clouds" (Line 479-482).

3. L. 162: The grid length of the MERRA data should be mentioned here.

The MERRA data is at the resolution of $0.5° \times 0.625°$. This information was indeed included, and now it is at Line 183-184, "meteorological lateral boundary and initial conditions were created from MERRA-2 at the resolution of $0.5° \times 0.625°$ (Gelaro et al., 2017)."

4. L 198: Rather than sending the reader to Lebo et al. (2012), please explain what is meant by "explicit representation of supersaturation over a time step". Is this close to the quasi-equilibrium supersaturation?

We have added a sentence to describe it after that sentence since we do not think it is needed to copy the equation from Lebo et al. 2012 and put there, i.e., "That is the supersaturation is solved by the source and sink terms of dynamic forcing and condensation/evaporation within an one-timestep" (Line 226-228).

5. L. 229. I would not call the agreement shown in Fig. 3 "very good". This would imply that a color inside each circle is as in the background. This is not the case in several circles. Similar comment applies to Figs. 4 – 6. I understand the difficult task the model faces, but being honest about the simulation drawbacks would be appropriate. For instance, in Fig. 6, the temperature and wind simulations are closer to each other than to the observations.

We have added text to point out the simulation drawbacks. At Line 258-260, "Though not exactly the same, the values from D1_MOR_anth show a similar distribution with the observations in terms of the surface PM2.5 averaged over 24 hours (the day before the convection near Houston)." Also at Line 281-285: "Compared with the coarse resolution NLDAS data, both SBM_anth and MOR_anth capture the general temperature pattern with a little overestimation at the northeast part of the domain (mainly rural area) . The modeled southerly winds do not reach further north as the NLDAS data, possibly because of the feedback of the small-scale features which are simulated with the high resolution to mesoscale circulations."

6. What is "thermal buoyancy"? I think this is just "buoyancy", correct? Please change.

Thermal buoyancy is the buoyancy contributed from temperature and moisture perturbation. We have added a note about this. Buoyancy can be attributed to temperature and moisture perturbation and condensate loading. The net buoyancy is the sum of thermal buoyancy and condensate loading.

7. Fig. 8. To me, the simulations look close to each other and different than the NEXRAD picture. Is the plot for all three ensemble members? This needs to be clearly stated.

Yes, this is for all three ensemble members. This information is now added to the figure captions. The major differences between the two simulations are at the low (<12 dBZ) and high large reflectivity (> 48 dBZ).

8. Fig. 9. Again, are the plots for all ensemble members? How large is the variability among the ensemble members? I suggest to show the total accumulation in addition to the rate. Total accumulation tends to eliminate the impact of statistical fluctuations due to different flow realizations.

Yes, this is for all three ensemble members. The shaded area shows the ensemble spread. We have also added the information of accumulated precipitation: "The observed accumulated rain over the time period shown in Fig. 9a is about 3.8 mm, both SBM_anth (~4.5 mm) and MOR_anth (~4.2 mm) overestimate the accumulated precipitation due to the longer rain period compared with the observations" (Line 313-316).

9. Fig. 10. Again, all ensemble members? How different are the figures for individual ensemble members? If they are much different, then more ensemble members are needed.

Yes, this is for all three ensemble members. The differences between the individual ensemble members is not very much. And also considering the expensive computation cost, we decide to keep at the current three members.

10. Fig. 13. Again, all ensemble members? What is the "drop nucleation rate"? Is this "CCN activation rate"? To what extent it is affected by the low vertical resolution?

Yes, this is for all three ensemble members. Droplet nucleation rate is also named as the CCN activation rate. The activate rates from SBM are shown ok. See our reply to the major comment #5

11. For figures 1 -14, it is not clear how the averaging is done. For instance, if there are differences in the number of updrafts but their strength does not change, some of those profiles would change as well, correct? I think one has to clearly explain how the averaging is done to get a clear picture of the processes involved. And document the ensemble spread. As an example, see section 6 in Grabowski's manuscript (JAS 2020, Early Online Release) that discusses the incorrect interpretation of the enhanced lighting over south-east Asia shipping lines. More latent heating may simply come from a larger number of convective updrafts, not necessarily stronger updrafts.

The average is done only over the grid points satisfying the thresholds described in each figure caption, meaning other grid points failed to meet the thresholds are not accounted for the average. We made this clear in the figure capture. The ensemble spread is marked as shaded areas for all profiles.
In our case the updraft speeds are indeed stronger, not because of more updrafts, as seen from the PDF figure (Figure 10).

**Responses to the reviewer' questions about the cold-phase and warm-phase invigoration by aerosols**.

The reviewer's comments:
It presents the authors view that is not supported by simple arguments and by other studies. For instance, the "cold-phase invigoration" as described in lines 42-45 is simply not possible because the latent heat released by freezing the cloud water carried across the melting level in the polluted case only balances the weight of the water carried upwards. So where does the invigoration come from? The explanation of the "warm-phase invigoration" is simply incorrect and it repeats the incorrect argument used in papers the authors cite. The latent heating does not depend on the droplet concentration and droplet radius as long as the updraft velocity does not change. This is strictly true when the in-cloud supersaturation is equal to the quasi-equilibrium supersaturation. The validity of the quasi-equilibrium supersaturation approximation has been argued in many studies, at least in the absence of ice (e.g., Politovich and Cooper JAS 1988). Such an incorrect interpretation is repeated in lines 334-337. I suggest the authors consult the recently accepted manuscript by Grabowski that provides a thorough discussion of the two invigorations, see section 2 there. The manuscript is available on EOR in JAS (https://journals.ametsoc.org/doi/abs/10.1175/JAS-D-20-0012.1). I also suggest the authors consult and cite a paper by Varble (JAS 2018) for a less biased discussion of the invigoration problem.

Here are the follow-on comments from the reviewer on the cold-phase and warm-phase invigoration.

I appreciate the authors' responses to my initial comments. However, the response has fundamental flaws and thus I cannot consider the issues settled. I think the authors need to reconsider my comments about the invigoration and correct obvious flaws in their arguments.

About the cold-phase invigoration:
The authors clearly misunderstood my argument. The figure below illustrates the starting point for my argument:

The left part of the panel shows a cloudy parcel that rises through the melting (freezing) level in pristine conditions. The total liquid condensate above the freezing level is $q_c$. The right part of the figure shows situation when a similar parcel rises in polluted conditions. Because of the less efficient warm rain processes, the parcel above the freezing level carries more liquid condensate, $q_c + \delta q_c$. Freezing of $\delta q_c$ in the authors' opinion (and in many other papers) is the reason for the invigoration. However, to carry $\delta q_c$ across the freezing level requires extra buoyancy when compared to the left panel. As shown in section 2a of Grabowski and Morrison (2020), the two effects approximately balance each other. It follows the original sentences in the manuscript under review that say: "… a well-known theory is that increasing aerosol concentrations can suppress warm rain as a result of increased droplet number but reduced droplet size. This allows more cloud water to be lifted to a higher altitude wherein the freezing of this larger amount of cloud water induces larger latent heating associated with stronger ice microphysical processes, thereby invigorating convective updrafts…" are simply incorrect and require additional explanations. For instance, the invigoration would be possible if the frozen condensate was removed through precipitation processes.

That said, I really do not think referring to invigoration is needed for this manuscript. If the authors insist, then the introduction and references to the invigoration in the text should provide a less biased discussion, for instance, as in Grabowski and Morrison JAS papers and as in Varble (JAS 2018).

About the warm-phase invigoration:
The phase relaxation time scale and the quasi equilibrium supersaturation estimates in the authors' response above are simply wrong. Below I include a table from Politovich and Cooper (JAS 1988) that shows phase relaxation time scale for different combinations of droplet concentrations and radii. These values are much smaller than those shown in the authors' response.

TABLE 1. Time constant characterizing supersaturation. (Values of $\tau = 1/(a_2 I)$ s for $p = 771$ mb, $T = 4.3°C$)

| Radius ($\mu$m) | Droplet concentration (cm$^{-3}$) | | | |
| --- | --- | --- | --- | --- |
| | 100 | 300 | 500 | 1000 |
| 2 | 14.1 | 4.7 | 2.8 | 1.4 |
| 3 | 8.7 | 2.9 | 1.7 | 0.87 |
| 5 | 4.9 | 1.6 | 0.98 | 0.49 |
| 10 | 2.3 | 0.77 | 0.46 | 0.23 |

The key question is why?
The explanation is relatively simple. The authors say that they use the formulas from Pinsky et al., eq. (4) therein. However, Pinsky et al. apply a simplified (and in my view incorrect) droplet growth equation that is different from the comprehensive formula used in Politovich and Cooper (JAS 1988). The key point is that one has to apply exactly the same droplet growth equation in the phase relaxation time scale calculation (and thus in the quasi-equilibrium supersaturation) as used in the numerical model. I expect Khain's bin microphysics applies a correct droplet growth formulation that is close to the one used in Politovich and Cooper (JAS 1988), and not the simplified droplet growth equation applied in Pinsky et al. The supersaturation simulated by the model can be compared to the diagnosed quasi-equilibrium supersaturation only if exactly the same droplet growth equations are used in both. This was the case for the relatively good agreement shown in Grabowski and Morrison (JAS 2017), at least below the freezing level, see Fig. 15 therein. In summary, the values shown in the authors' response above have to be

Corrected.
The above discussion requires the authors to modify their responses and revise their paper accordingly. Note that the second part of my rebuttal impacts some of my other original comments. I strongly object publication of the manuscript unless those comments are appropriately addressed

As we noted earlier, the two sides of arguments have been existing for a while, and it should not be the role of this paper to debate and resolve this issue. Here we only provided our key points, the detailed review and comment paper was submitted to J. Atmos. Sci., which would allow both sides to discuss and debate there. The bulk of the above comments are chiefly the expression of the reviewer's view on the aerosol invigoration effect, rather any substantial objection to the scientific importance and the findings of this study.

For the "cold-phase invigoration", the reviewer's argument "the increase in the buoyancy by freezing is completely offset by the buoyancy for carrying the extra cloud water across the freezing level" has several issues:
(1) Droplet ascending and then freezing are subsequent at different locations; also, the two processes can take at very different time scales (freezing is instant but ascending could take much longer time). How do they compensate each other at different time scale and locations? Responses of a complex non-linear dynamical system in deep convective clouds strongly depend on duration and location of the forcing.
(2) In the process of ascending in updrafts, droplets will grow through condensation, and the changes in latent heating and condensate loading from this are not considered in this argument.
(3) The argument neglected the subsequent enhanced riming and deposition resulting from more ice particles formed from enhanced droplet freezing. This leads to (a) a further increase in latent heating and (b) a reduction in condensate loading because more graupel and hail form due to increased supercooled liquid content and precipitate. Rosenfeld et al. (2008) indeed considered the possible compensation between the extra condensate loading and the extra latent heat of freezing. They showed (in line d of their Fig. 3) that the total buoyancy excesses and invigoration occurs after the ice hydrometeors are unloaded (i.e., precipitated) from the cloud parcel. The unloading is quite efficient in the case of rich supercooled liquid water content where large particles like graupel and hail can form. Many modeling studies have showed that the latent heat released from deposition and riming is much larger (at least an order of magnitude) than freezing. The increase in latent heating by aerosols is mainly from the increase in deposition and riming at the high-levels. Overall, the buoyancy increase via latent heat release exceeds the buoyancy decrease resulting from the increase in condensate loading, leading to a positive net buoyancy (e.g., Fan et al. 2012a, 2018; Tao and Li, 2016; Lebo et al. 2012; Chen et al. 2020).

For the "warm-phase invigoration", our interpretation of the mechanism is the enhanced condensation by larger droplet nucleation in the polluted conditions releases more latent heat, enhance buoyancy this updraft intensity. This is consistent with many literature studies (e.g., Khain et al. 2012; Igel et al. 2015; Sheffield et al. 2015; Chen et al. 2017; Fan et al. 2018; Lebo 2018). The reviewer argued that condensation rates only depend on updraft velocity with the quasi-steady assumption (i.e., true supersaturation is approximated with equilibrium supersaturation), therefore they interpreted that it is the lower equilibrium supersaturation in polluted conditions that lead to a larger buoyancy, thus enhanced updraft speeds and condensation.

This quasi-steady assumption is neither physically justified for the strong updrafts of deep convective clouds nor is it suitable for studying aerosol effects on deep convective clouds which requires the exact solution of supersaturation. Previous studies (e.g., Politovich and Cooper 1988; Korolev and Mazin 2003, Pinsky et al. 2013) showed that the quasi-steady assumption is invalidated in conditions of (a) low droplet concentrations (pristine condition) and (b) intense condensation and evaporation (e.g., cloud base and strong updrafts) due to long relaxation time (larger than a few seconds). Note that both Politovich and Cooper (1988) and Korolev and Mazin (2003) evaluated the phase relaxation time under the assumption of the constant drop radius, which is not as accurate as Pinsky et al. (2013) that used the accurate equation for supersaturation. However, the reviewer mistakenly thought that Politovich and Cooper (1988) used an accurate droplet growth equation but Pinsky et al. (2013) used a simplified one. So we followed Eq. 3 and Eq. 4 in Pinsky et al. (2013) to calculate Seq and phase relaxation time $\tau$, respectively. Fig. r4 shows the calculated phase relaxation time as a function of droplet number and radius from SBM_anth. The values we got are quite consistent with the Table 1 of Politovich and Cooper (1988) for droplet number concentrations (Nc) of 100, 300, and 500 cm-3, which proves that our calculation has no problem. The reviewer said our values "are simply wrong" in his follow-on comment which were from the same calculation except we showed the mean value for the updrafts with a velocity greater than 2 m s−1 (Fig. r5). In these relatively strong updrafts, the phase relaxation time is long (Fig. r5c) because of low Nc (Fig. r5d) due to fast conversion of droplets to rain. Fig. r4 showed that most of the updrafts have Nc of 5-20 cm-3. The averaged $\tau$ for the updrafts greater than 2 m s−1 are around 10-15 s (with large values exceed 60 s). Above the cloud base, the quasi-equilibrium supersaturation is much higher than the true supersaturation (Fig. r5a) where low droplet number concentration (Fig. r5c) and strong updrafts (Fig. r5b) are seen. Therefore, it is clear that the short phase relaxation time (a few seconds) is only true near cloud base with large droplet number concentrations (~ hundreds cm-3) and weak updrafts. However, in relatively strong updrafts, the droplet number above the cloud base is much reduced (~ tens cm-3 in this case) due to fast conversion of droplets to rain, thus the phase relaxation time is much longer (> 10 s and even over 60 s) and the assumption of S=Seq is not valid any more. This is particularly true for the pristine case (SBM_noanth), we can see the Seq can be much higher than the true supersaturation (Fig. r2a), so assuming S=Seq would lead to a large bias in condensation and evaporation in the pristine case.

Therefore, appropriately simulating aerosol effects on deep convective clouds requires an exact supersaturation calculation (Eq. 6 in Pinsky et al. 2013), in which the condensation depends on droplet number and size, and more droplets in the polluted clouds increase condensation and decrease supersaturation, which clearly showed our interpretation is physically solid.  In addition, as shown in our responses to the comment #6,  we see the polluted case (SBM_anth) has larger condensation rates for the same updrafts than the pristine case (SBM_noanth) (Fig. r3, a vs c) above cloud base where the increase of cloud droplet number is significant (Fig. 15a). This clearly shows that higher droplet number leads to larger condensation rate for the same vertical velocity, which rebuts the reviewer' argument that condensate rates are similar under the same vertical velocity.

[Figure]

Figure r4 (a) Relationship between phase relaxation time and droplet radius for different droplet number concentrations from the simulations SBM_anth. (b) is the same as (a), except zooming in for droplet number concentrations of 100 cm$^{-3}$(black), 300 cm$^{-3}$(blue), 500 cm$^{-3}$(red).

[Figure]

Figure r5 Vertical profiles of (a) supersaturation and quasi-equilibrium supersaturation, (b) updraft velocity, (c) phase relaxation time and (d) droplet number concentration averaged over the updraft velocity 2 
[revised manuscript text omitted]

---

## Referee Report (RR1)

Final review of the revised manuscript "Impacts of Cloud Microphysics Parameterizations on Simulated Aerosol-Cloud Interactions for Deep Convective Clouds over Houston" by Zhang, Fan, Li, and Rosenfeld, considered for publication in ACP, manuscript acp-2020-372.

Recommendation: accept after minor revisions

I have to stress that my recommendation is NOT based on the satisfactory revisions (the revisions are relatively minor), but because of my expectation that the peer review process has to stop at some point. I strongly feel that the authors' interpretation of the invigoration problem is biased and this affects the discussion of past studies and current model results. For instance, on lines 52/53 the authors say that Grabowski and Morrison (2016, 2020) "reject […] invigoration concept". This is not true: Grabowski and Morrison argue that the simplistic view of the invigoration is difficult to understand based on the buoyancy below and above the freezing level in situation without and with pollution. On lines 81/84, the authors refer to Varble (2018) arguing that Varble's study suggests that 10 years of SGP data is not sufficient to separate meteorological factors from the impact of aerosol. This is not true: Varble shows that meteorological conditions over SGP explain observed changes in convection without referring to aerosols, and that the 10-year study the authors implicitly refer to is severely flawed. It is impossible for me to point all places where the authors are biased. I thus suggest that the paper is published, perhaps with some final changes following my comments below. That said, I have to stress – as I did before – that the manuscript does not need to refer to the invigoration. The difference between saturation-adjustment in the Morrison bulk scheme and saturation-prediction in the Khain bin scheme is sufficient to explain the differences. The strength of the manuscript is in a direct prediction of CCN formation and removal, and how these can be linked to cloud processes.

The only more important and still unanswered point made in a couple of my previous comments is that the differences in the representation of ice processes in the two schemes, and how they are linked to warm-rain process, need to be clearly explained. For instance, is the ice initiation the same in both schemes? A short paragraph for each scheme should be sufficient.

Another general comment is that English needs some improvements throughout the text. Some places are listed below.

Specific comments:

1. The last sentence of the abstract is unclear. First, saturation adjustment provides the maximum buoyancy, so I do not understand why "saturation adjustment […] limits the enhancement in condensation…". I would think the opposite should be true. The (2) seemingly refers to ice processes, but it is unclear why saturation adjustment affects conversion of droplets into raindrops, and how these are related to ice processes.

2. L. 195. "The meteorological fields were reinitialized…"

3. L. 225/227. The sentence: "…the supersaturation is solved by the source and sink in terms of dynamic forcing and condensation/evaporation within a one-timestep" needs to be rewritten. I commented on the subsequent sentence in the previous round ("…the supersaturation for condensation and evaporation is calculated after the advection."), see my point 2 in the previous review. Please revise.

4. L. 252 should be "Results".

5. Discussion of Figs. 7 an 8. I am not sure if I see differences between left and right panels in Fig.7. These are just different convection realizations to me. Can one apply an objective analysis to prove that? In Fig. 8, it is difficult for me to see which model simulation is closer to the observation with the exception of the echo top, higher in bin microphysics and in agreement with the radar plot.

6. Fig. 9. The authors ignored my suggestion to plot rain accumulation as more meaningful and less dependent on the specific realization of the flow.

7. Section 4.2, lines 405-456, the interpretation of results related to the impact of pollution on the supersaturation and particle growth between the two microphysics schemes. As I stated previously (and above), a brief description of ice microphysics between the two schemes, and the key differences, would provide a proper context for this discussion.

8. L. 481: "condenses all supersaturation". Perhaps "removes"?

---

## Author Response (AR2)

Responses to Reviewer 1

I appreciate the authors' effort of adding Figs. 16 - 18, comparing the original Morrison scheme with modified Morrison scheme (MOR_SS). The conclusion, that enhanced ice processes similar to the "cold-phase invigoration" are responsible for enhanced invigoration simulated in MOR_SS, is plausible. With these new analysis, the authors then added several lines to the end of the original abstract, which unfortunately made its logic confusing. The abstract reads: "With the SBM scheme, we see a significant invigoration effect on convective intensity and precipitation by anthropogenic aerosols mainly through enhanced condensation latent heating (i.e., the warm-phase invigoration). Whereas such an effect is absent with the Morrison two-moment bulk microphysics, mainly because the saturation adjustment approach for droplet condensation and evaporation calculation removes the dependence of condensation on droplet properties and limits the ice processes by a more efficient conversion of droplets into raindrops, which leads to less cloud droplets being transported to the altitudes above the freezing level." My interpretation of this passage is: SBM scheme simulated significant invigoration. The mechanism is mainly through enhanced condensation (enhanced warm-phase invigoration). Morrison scheme produced less invigoration. The reason is related to reduced ice processes (reduced cold-phase invigoration?). To paraphrase based on Figs. 16 - 18, the MOR_SS scheme produced significant invigoration due to enhanced ice processes? Here both SBM and MOR_SS are compared to the original Morrison scheme. But SBM have more invigoration due to warm-phase processes. And MOR_SS, with similar invigoration as SBM, is due to cold-phase invigoration? How can this inference be? Shouldn't the invigoration mechanism be the same for both the SBM and the MOR_SS?

I maintain that the manuscript tackles an important, and difficult problem. It has solid and comprehensive analysis and qualifies for ACP publication. Revisions are needed in the abstract and conclusions, especially the abstract, to avoid ambiguity and keep focus. I recommend publication after minor revision.

We thank the reviewer for the careful review. The enhanced cold-phase processes by changing the saturation adjustment to the explicit supersaturation was only for explanation of the larger convective intensity seen in the explicit supersaturation in the polluted case (anthropogenic aerosols considered; i.e., MOR_anth vs MOR_SS_anth). For the aerosol effect, from the clean (without anthropogenic aerosols) to the polluted (with anthropogenic aerosols) cases (i.e., from MOR_noanth to MOR_anth vs from MOR_SS_noanth to MOR_SS_anth), there is a large condensation increase with the explicit supersaturation because the approach makes the condensation calculation dependent of droplet properties/aerosols. Sorry we forgot to emphasize this point, which caused the confusion. Basically, for the aerosol effects, the saturation adjustment approach limits the enhancement in (1) condensation latent heating by removing the dependence of condensation on droplet properties and (2) the ice-related processes by a more efficient conversion of droplets into raindrops that leads to less cloud droplets being transported to the altitudes above the freezing level. With the explicit supersaturation, the enhanced condensation heating should play a more significant role in convective invigoration than that from enhanced ice processes, which is consistent with the point with SBM. We have made changes to the abstract, the discussion of the sensitivity tests with the explicit supersaturation, and conclusion to make the point consistent (the changed text in in red in the tracked-change version).

P.s., in the revised version, we have moved the figures about the detailed analysis of the differences produced by the saturation adjustment and the explicit supersaturation at different convective stages to the supplemental materials in order to keep the paper neater and not have an excessive number of figures. The discussion is till kept in the main paper.

Responses to Reviewer 2
The revised manuscript has been improved. In particular, the invigoration (that is really only tangentially related to this study) is now better presented as still a controversial subject requiring additional studies. First, let me note that not all of my points have been addressed. In particular, I suggested that the two schemes applied in the simulations, and specifically their ice components, need to be explained in more detail (see my previous point 7). The additional material added in the revision provides some confusion to me, see below. I would either expand the analysis or remove part of that material to create no confusion, see specific comments below. This material is really not needed for the main thrust of the paper, and it should be suitable for a future manuscript when more analysis is done. Perhaps the most significant comment at this stage is that in my opinion the authors provide a bias interpretation of some of the figures. I present specific examples below. Also, some figures require adjustments as detailed below.

I should also mention in passing, that I am not fully satisfied with our previous exchanges, the quasi-equilibrium supersaturation discussion in particular. However, since this aspect is not part of the manuscript, I will put this issue aside, perhaps with the exception of the comment 2 below.

Thank the reviewer for the further comments. The additional analysis and discussion added in the previous round was to explain how the saturation adjustment approach and the explicit supersaturation produced the different results on convection, which was comment from another reviewer. The analysis also helps to explain why the saturation adjustment approach shows limited aerosol effect. We agree that such in-depth analysis has too much detail, which makes the paper have too many figures and seems too long. Thus, we have moved the figures to the supplemental materials in the revised manuscript. In addition, we have reorganized the text and revised the discussion to be clearer.
For the specific comments below, please refer to our point-by-point response. For some results, we think the reviewer misread our plots, and misunderstood what processes were changed in the explicit supersaturation experiments (droplet/ice nucleation was not changed to using the explicit supersaturation).

Specific comments:

1. The first sentence of section 2 is a repetition of the text a couple lines above, at the end of section 1. Please remove either one.

We have now modified the first sentence of section 2 to be different.

2. L. 228-229: What do you mean: "…the supersaturation for condensation and evaporation are calculated after the advection"? Do I understand that the model advects temperature and water vapor, and then combined the two to create supersaturation that is subsequently applied in microphysical calculations? If so, this is simply wrong as supersaturation calculated this way has no physical meaning and it depends on the time step applied in the advection (the longer the time step the larger the supersaturation calculated this way). If this is how WRF model works, this is a shame. There are different methods to do this correctly. See, for instance, the seminal paper by Hall (JAS 1980, p. 2486; see discussion in section 3e). My suggestion is to remove this sentence and not to open the Pandora box. But this is something to think about for the future.

We removed that sentence as suggested. But we would like to clarify that in our WRF simulations with both the explicit supersaturation of Morrison and SBM, the supersaturation is solved semi-analytically based on both the forcing from advection and the microphysics processes. In other words, before being applied to calculate condensation and evaporation, the supersaturation is derived by solving an equation system taking into account the advection and diffusional growth simultaneously over a dynamic time step. In SBM, this analytical calculation of supersaturation changes is done even during the microphysical time step (which a few times shorter than the dynamic time step) and the advection part is calculated from the temperature and water vapor differences between the current and previous time steps and is evenly distributed over the microphysical time steps. These method further reliefs the high supersaturation values and numerical droplet spectrum broadening.

3. Figure 3: as in my previous comments, there are more points with different colors than in agreement with the model. The phrase "though not exactly the same" in l. 258 in a stretch. My suggestion is to remove this figure and only include Fig. 4, maybe with the time averages. But if the goal is to show the spatial pattern, then please say that the model represents the spatial pattern, but has problems with specific measurement sites, not surprisingly I would think.

Yes, the purpose of Fig. 3 is to show the spatial pattern, so we prefer to keep it. We have modified the text as Line 256-259, "D1_MOR_anth shows a similar spatial pattern with the observations in terms of the surface PM2.5 averaged over 24 hours (the day before the convection near Houston), although with a difficulty to reproduce the values for some sites."

4. Figure 4. The comparison between model and satellite estimates is difficult. Maybe scatterplots (i.e., one versus the other, without geographic information) would be better?

It is difficult to do the scatterplot for one by one comparison since the simulated clouds may have shifts with the location of cloudy points from the satellite estimation. Fig. 5 can also show the cloud distribution over the domain and the spatial variability.

5. Figure 6. Models are close to each other and they both miss to represent E-W temperature gradient apparent in the observations. Why?

This could be related to many reasons such as the initial and boundary conditions, soil moisture conditions, surface and PBL parameterizations. It is not relevant to microphysics parameterization since both schemes have the same problem. This should not matter much to our study it is at the northern part of the domain and far from the Houston region that we studied.

6. Figure 7 is really nice, but its discussion in the text is not. First, it is great to see that all model realizations feature a cluster of convective cells in the red box (as in observations), and some convection NE to the red box. Is the position of the cluster inside the red box related to the aerosol gradient between the Houston and its vicinity? I am also not sure if I see the differences the text emphasizes. To me, model results are simply slightly different realizations of the same convective situations. If anything, the SBM realizations look a little more "diffused" to me.

There is no significant aerosol concentration gradient in the red box. For the Houston cell (29.6N-29.8N, 95.1W-95.3W) observed by NEXRAD, SBM gets the similar location, size and high reflectivity value. However, it is clear that Morrison has several smaller and scattered cells with weaker reflectivity.

7. Figure 9 and its discussion. First, I prefer to consider rain accumulation, not intensity, as more instructive. The upper panel has to include line definitions. Models start earlier than observations, solid lines before dashed (I do not know which one is which). To me, all lines are simply different realizations. Bottom panel: the anth and noanth reverses in both schemes between small rates (noanth larger) and high rates (anth larger). This should be mentioned. I suggest to use linear scale (different for different rate ranges) as the linear scale shows much better the difference. The way the figure is plotted now (rate rather than accumulation in the upper panel and log scale in the lower panel) is suboptimal. The text emphasizes differences ("remarkable" in l. 321) that I do not see. The same for the difference between SBM and MOR: the lower panel shows similar trend and I do not see what the text says in lines 321-325.

We updated Fig.9a about the legend. The linear scale is difficult to show the differences for the large precipitation rates, which has a couple of orders of magnitude lower frequency than the light precipitation rates. The accumulated precipitation was already discussed, i.e., "The observed accumulated rain over the time period shown in Fig. 9a is about 3.8 mm, both SBM_anth (~ 4.5 mm) and MOR_anth (~ 4.2 mm) overestimate the accumulated precipitation due to the longer rain period compared with the observations." (Line 313-316). We have added the discussion of similarity in aerosol effect on precipitation rates between SBM and MORR at Lines 325-331 "Both SBM and Morrison schemes show higher occurrences of large precipitation rates (> 10 mm h$^{-1}$) and lower occurrence of small precipitation rates (< 10 mm h$^{-1}$) due to anthropogenic aerosols (Fig. 9b), but the effect is larger with SBM. For the accumulated precipitation, the anthropogenic aerosols lead to a ~ 0.5 mm increase over the storm period with the SBM scheme, while only a ~ 0.2 mm increase with the Morrison scheme.". This point has also been added to Conclusion.

8. Figure 10. The impact of pollution is remarkably similar in both models. I do not see much invigoration below freezing levels (that may come from different supersaturation treatment in SBM and MOR) and there is some aloft. But where it does come from for MOR? Maybe because of difference in ice processes? But we do not know how ice processes differ in the two models. This comment is also important to understand the impact of supersaturation prediction in MOR simulations.

The reviewer may have misunderstood the figure. Figure 10b is so different from Fig 10a. There is almost no occurrence of w > 20 m s$^{-1}$ without anthropogenic aerosols in Fig. 10a but the occurrences of w of 20-36 m s$^{-1}$ are significant with anthropogenic aerosols considered in Fig. 10b. This is drastic, meaning aerosol effect is so significant with SBM. The increase of w below the freezing level is also very clear (the maximum updraft speed with the occurrence frequencies greater than 0.1% increase from 20 m s$^{-1}$ to 26 m s$^{-1}$). However, with the Morrison scheme (Fig. 10c-d), the anthropogenic aerosol effect is not much, and there is a little increase of w aloft, and it would come from the enhanced ice processes by aerosols.

9. Figure11. Left and right panels (SBM and MOR) are remarkably similar. NB, I commented before that the term "thermal buoyancy" has to be replaced by "buoyancy"; "thermal" is confusing because buoyancy has to include temperature and all water variables. The difference between MOR_noanth and MOR_anth has to do with different flow realizations, at least below the freezing level (again, what about ice representation?). There is no other explanation unless one argues about the feedback from rain processes. The MOR_SS results are unexpected as it shows the opposite impact of including finite supersaturation on the mean updraft, buoyancy, and latent heating. The rest of the manuscript tries to explain this surprising result.

Thermal buoyancy is more directly related to latent heating than the total buoyancy. Anyway, we now showed the total buoyancy now in the revised figure.
The remarkably similar results are between SBM (red lines) and MOR_SS (the test with explicit supersaturation; golden lines), but not between SBM and MOR (blue lines). We do not have an idea where the reviewer's argument "The MOR_SS results are unexpected as it shows the opposite impact of including finite supersaturation on the mean updraft, buoyancy, and latent heating" comes from, since the figure clearly showed MOR_SS did not give any opposite results compared with MOR or even SBM. MOR give a small increase in updraft, buoyancy, and latent heating by aerosol effect and MOR_SS just enlarges the differences and showed a larger increase. The reason for the larger aerosol effect with the explicit supersaturation was discussed in the paragraph after this. Here, in short, the saturation adjustment approach for droplet condensation and evaporation calculation limits the enhancement in (1) condensational latent heat by removing the dependence of condensation on droplet properties and (2) ice-related processes by a more efficient conversion of droplets into raindrops, which leads to less cloud droplets being transported to the altitudes above the freezing level.

10. To understand the effects of supersaturation prediction in MOR_SS simulations, one needs to know more details on how the prediction is linked to the microphysics. For instance, how does the change affect the droplet activation and ice processes? Grabowski and Morrison (JAS 2017, p. 2247) document some impacts on ice processes when saturation adjustment is replaced with saturation prediction. Do you still use Ghan et al. droplet activation parameterization when predicting supersaturation? How ice initiation is affected? Fig. 13 shows that saturation prediction has small impact on droplet activation (mass-wise), but large and unexpected impact on the condensation rate. How is the latter possible? I think an attempt is made to explain this in the rest of the paper, but I suggest to remove that part and MOR_SS simulations altogether as only tangentially related to main results of this study.

The reviewer had a misunderstanding about how the predicted supersaturation was used in MOR_SS. We clearly stated in Line 223-225 "by replacing the saturation adjustment approach in the Morrison scheme with the condensation and evaporation calculation based on an explicit representation of supersaturation over a time step as described in Lebo et al. (2012)." Thus, except droplet condensation and evaporation, all other processes including droplet activation and ice nucleation process between MOR and MOR_SS are exactly the same. This is to look at the impact of the saturation adjustment. That is also why a small impact on droplet activation in Fig. 13b is seen with the explicit supersaturation. As clarified above, the additional analysis and discussion were added to explain why different treatments of condensation/evaporation give different aerosol effects as seen in Fig. 13 (a comment from another reviewer). We agree that such in-depth analysis has too much detail, which makes the paper have too many figures and seem too long. Thus we have moved the figures to the supplemental materials in the revised manuscript. In addition, we have reorganized and revised the discussion to be clearer.

11. Figure 13: what is "condensation rate" in lower panels? Is that water vapor to cloud water, or water vapor to either water or ice? Please clarify.

We clarified in Line 383-384: "the condensation rates (i.e., the rate of gain in cloud water due to water vapor condensation)."

12. I have to admit that I did not read and study figures beyond 13, but I find similarities between simulations applying the SBS and MOR schemes remarkable. This should be emphasized in the paper.

Again, SBM and MOR are not similar in aerosol effects. What's similar is between SBM and MOR_SS. The only similarity is the structure of vertical profiles of updraft and hydrometeor properties, which would be mainly determined by the meteorological conditions. We have added a sentence about this in both main text and conclusion. Here is the sentence in the conclusion "Indeed, both schemes show similar vertical structures of convective intensity and hydrometeor properties, with a weaker updraft intensity with the Morrison scheme at high altitudes in the case with anthropogenic aerosols considered." (Line 474-476)

13. In view of my comments above, I do not agree with many statements in the concluding section 5, especially those related to cold and warm invigoration. I feel the section does not properly reflect results presented in the figures and is biased towards the authors' view rather than what model results show. This has to change before the manuscript is accepted.

As clearly shown in our responses above, the reviewer had some misunderstandings about the results shown in our figures. Our conclusions are strictly consistent with the results shown from the figures. We also modified some wordings and added some points pointed out by the reviewers (mainly about the similar aspects between SBM and MORR) in the revised conclusions, such as lines 474-479 and 496-499.

[revised manuscript text omitted]

---

## Author Response (AR3)

Responses to Reviewer 2
I have to stress that my recommendation is NOT based on the satisfactory revisions (the revisions are relatively minor), but because of my expectation that the peer review process has to stop at some point. I strongly feel that the authors' interpretation of the invigoration problem is biased and this affects the discussion of past studies and current model results. For instance, on lines 52/53 the authors say that Grabowski and Morrison (2016, 2020) "reject […] invigoration concept". This is not true: Grabowski and Morrison argue that the simplistic view of the invigoration is difficult to understand based on the buoyancy below and above the freezing level in situation without and with pollution. On lines 81/84, the authors refer to Varble (2018) arguing that Varble's study suggests that 10 years of SGP data is not sufficient to separate meteorological factors from the impact of aerosol. This is not true: Varble shows that meteorological conditions over SGP explain observed changes in convection without referring to aerosols, and that the 10-year study the authors implicitly refer to is severely flawed. It is impossible for me to point all places where the authors are biased. I thus suggest that the paper is published, perhaps with some final changes following my comments below. That said, I have to stress – as I did before – that the manuscript does not need to refer to the invigoration. The difference between saturation-adjustment in the Morrison bulk scheme and saturation-prediction in the Khain bin scheme is sufficient to explain the differences. The strength of the manuscript is in a direct prediction of CCN formation and removal, and how these can be linked to cloud processes.

We are glad that the reviewer now softened their arguments in Grabowski and Morrison (2016, 2020). Here are the original sentences from the abstract and conclusion of those two papers:
"There is no impact on convective dynamics above the freezing level and thus no convective invigoration" (abstract of GM16); "In summary, the invigoration of deep convection in polluted environments, either resulting from increased total CCN concentrations as suggested in Rosenfeld et al. (2008) or from addition of small CCN to the pristine environment as in Fan et al. (2018), is not supported by theoretical arguments and cloud simulations presented here" (Conclusion of GM20).
So our statement is the exact message that the authored delivered in their papers. Anyway we reworded the sentences at Line 51-52: "Grabowski and Morrison (2016; 2020) argued this invigoration does not exist because the increase in the buoyancy by freezing is completely offset by the buoyancy for carrying the extra cloud water across the freezing level."
About Varble (2018), the paper clearly stated "It will be shown that for the Atmospheric Radiation Measurement (ARM) Southern Great Plains (SGP) site in north-central Oklahoma, 14 years of observations are not sufficient for concluding that increased aerosol concentrations cause an increase in cloud-top height and decrease in CTT…".
We would like to point out to the reviewer that Varble 2018 used the CAPE from the convective storm period, not the pre-convection environment CAPE. That was a problem for the analysis since the CAPE from convective period already accounted for the microphysics feedback to dynamics. Even with that, the study did not rebut the invigoration. It stated clearly "These results do not draw into question the validity of the well-reasoned aerosol convective invigoration hypothesis for certain environmental conditions. Rather, they highlight the need for more careful, detailed, and strategic observational analyses…".

The only more important and still unanswered point made in a couple of my previous comments is that the differences in the representation of ice processes in the two schemes, and how they are linked to warm-rain process, need to be clearly explained. For instance, is the ice initiation the same in both schemes? A short paragraph for each scheme should be sufficient.

As explained in the previous responses, SBM and Morrison are completely two different realizations of cloud microphysics so they are different in the way of treating every microphysical processes. Since our results showed that the main reason responsible for the different aerosol effect on the convective storm is the treatment of condensation and evaporation, we do not think laying out all the other model differences are important and it would take long text which would distract the paper. But, we added a couple of sentences to mention the differences in ice processes:
"The SBM and Morrison schemes are two completely different representations of cloud microphysics so they are different in many aspects including major microphysical processes such as aerosol activation, condensation/evaporation, collisions, and ice nucleation and ice growth through riming and aggregation. Details are read from Khain et al. (2004; 2015), Morrsion et al. (2015) and Gao et al. (2016)".

Another general comment is that English needs some improvements throughout the text. Some places are listed below.
We did a careful review to improve the language now. For the specific comments below, see our response one-by-one.

Specific comments:
1. The last sentence of the abstract is unclear. First, saturation adjustment provides the maximum buoyancy, so I do not understand why "saturation adjustment […] limits the enhancement in condensation…". I would think the opposite should be true. The (2) seemingly refers to ice processes, but it is unclear why saturation adjustment affects conversion of droplets into raindrops, and how these are related to ice processes.

Here what we said is that the saturation adjustment limited the aerosol effect, which does not conflict with saturation adjustment provides the maximum buoyancy. The saturation adjustment limits the enhancement in condensation latent heat from clean to polluted conditions because it removes the dependence of condensation on droplets/aerosols.
About why saturation adjustment affects conversion of droplets into raindrops and how these are related to ice processes, the reviewer missed the part of the analysis of this study, which is detailed at Line 410-433. In short, the saturation adjustment leads to larger droplet sizes in the warm cloud stage than the explicit supersaturation (i.e., MOR_anth vs MOR_SS_anth), leading to more efficient conversion of droplets into raindrops, which leaves less droplets available for being transported to the altitudes above the freezing level at the mixed-phase and deep cloud stages, thus weaker ice microphysical processes are seen. We have reworded the sentence to make it easier to understand, "…mainly because the saturation adjustment approach for droplet condensation and evaporation calculation limits the enhancement by aerosols in (1) condensation latent heat by removing the dependence of condensation on droplets/aerosols and (2) ice-related processes because the approach leads to stronger warm rain and weaker ice processes than the explicit supersaturation approach".

2. L. 195. "The meteorological fields were reinitialized…"

Changed.

3. L. 225/227. The sentence: "…the supersaturation is solved by the source and sink in terms of dynamic forcing and condensation/evaporation within a one-timestep" needs to be rewritten. I commented on the subsequent sentence in the previous round ("…the supersaturation for condensation and evaporation is calculated after the advection."), see my point 2 in the previous review. Please revise.

Modified as suggested: "That is, the supersaturation is solved semi-analytically based on both the forcing from advection and the microphysics processes."

4. L. 252 should be "Results".

Changed.

5. Discussion of Figs. 7 an 8. I am not sure if I see differences between left and right panels in Fig.7. These are just different convection realizations to me. Can one apply an objective analysis to prove that? In Fig. 8, it is difficult for me to see which model simulation is closer to the observation with the exception of the echo top, higher in bin microphysics and in agreement with the radar plot.

The objective analysis/description was already shown in Line 310-324. In Fig.7, SBM_anth simulates the storm closer to the observation in both location and high reflectivity value (greater than 55 dBZ) than MOR_anth. In Fig.8, SBM_anth is in a better agreement with observation in terms of vertical structure of the high reflectivity range (> 48 dBZ) and echo top heights. MOR_anth did not simulate any occurrence of reflectivity larger than 55 dBZ.

6. Fig. 9. The authors ignored my suggestion to plot rain accumulation as more meaningful and less dependent on the specific realization of the flow.

As responded previously, we described the rain accumulation with text (Line 322-324; Line 338-340) since no figure is necessary for such a quantity given we have so many figures already.

7. Section 4.2, lines 405-456, the interpretation of results related to the impact of pollution on the supersaturation and particle growth between the two microphysics schemes. As I stated previously (and above), a brief description of ice microphysics between the two schemes, and the key differences, would provide a proper context for this discussion.

As also clarified in our previous response, the discussion for this part is based on the results from MOR and MOR_SS. There is no difference in ice microphysics between them and the test is only for the effect of different treatment of condensation/evaporation.

8. L. 481: "condenses all supersaturation". Perhaps "removes"?

"Condenses" is changed to "removes".

**Impacts of Cloud Microphysics Parameterizations on Simulated Aerosol-Cloud-Interactions**

**for Deep Convective Clouds over Houston**

Yuwei Zhang[1, 2], Jiwen Fan[2, *], Zhanqing Li[1], Daniel Rosenfeld[3]

[1]Department of Atmospheric and Oceanic Science, University of Maryland, College Park, MD,

USA

[2]Atmospheric Sciences and Global Change Division, Pacific Northwest National Laboratory,

Richland, WA, USA

[3] Institute of Earth Sciences, The Hebrew University of Jerusalem, Jerusalem, Israel

* *Correspondence to*: Jiwen Fan (jiwen.fan@pnnl.gov)

**Abstract**

Aerosol-cloud interactions remain largely uncertain in predicting their impacts on weather and climate. Cloud microphysics parameterization is one of the factors leading to large uncertainty. Here we investigate the impacts of anthropogenic aerosols on the convective intensity and precipitation of a thunderstorm occurring on 19 June 2013 over Houston with the Chemistry version of Weather Research and Forecast model (WRF-Chem) using the Morrison two-moment bulk scheme and spectral-bin microphysics (SBM) scheme. We find that the SBM predicts a deep convective cloud agreeing better with observations in terms of reflectivity and precipitation compared with the Morrison bulk scheme that has been used in many weather and climate models. With the SBM scheme, we see a significant invigoration effect on convective intensity and precipitation by anthropogenic aerosols mainly through enhanced condensation latent heating. Whereas such an effect is absent with the Morrison two-moment bulk microphysics, mainly because the saturation adjustment approach for droplet condensation and evaporation calculation limits the enhancement by aerosols in (1) condensation latent heat by removing the dependence of condensation on droplets/aerosols and (2) ice-related processes because the approach leads to stronger warm rain and weaker ice processes than the explicit supersaturation approach.

[revised manuscript text omitted]